# Lithium intercalation into bilayer graphene

Kemeng Ji [1,2], Jiuhui Han[1,2], Akihiko Hirata[1], Takeshi Fujita [1], Yuhao Shen[1,3], Shoucong Ning[4], Pan Liu [1], Hamzeh Kashani[1,2], Yuan Tian[1,2], Yoshikazu Ito[5,6], Jun-ichi Fujita[5] & Yutaka Oyama[2]

The real capacity of graphene and the lithium-storage process in graphite are two currently perplexing problems in the field of lithium ion batteries. Here we demonstrate a three-dimensional bilayer graphene foam with few defects and a predominant Bernal stacking configuration, and systematically investigate its lithium-storage capacity, process, kinetics, and resistances. We clarify that lithium atoms can be stored only in the graphene interlayer and propose the first ever planar lithium-intercalation model for graphenic carbons. Corroborated by theoretical calculations, various physiochemical characterizations of the staged lithium bilayer graphene products further reveal the regular lithium-intercalation phenomena and thus fully illustrate this elementary lithium storage pattern of two-dimension. These findings not only make the commercial graphite the first electrode with clear lithium-storage process, but also guide the development of graphene materials in lithium ion batteries.

[1] WPI Advanced Institute for Materials Research (AIMR), Tohoku University, Sendai 980-8577, Japan. [2] Department of Materials Science, Graduate School of Engineering, Tohoku University, Sendai 980-8579, Japan. [3] Key Laboratory of Polar Materials and Devices, East China Normal University, 200062 Shanghai, China. [4] Department of Mechanical and Aerospace Engineering, School of Engineering, Hong Kong University of Science and Technology, Hong Kong SAR 999077, China. [5] Institute of Applied Physics, Graduate School of Pure and Applied Sciences, University of Tsukuba, Tsukuba 305-8573, Japan. [6] PRESTO, Japan Science and Technology Agency, Saitama 332-0012, Japan. Correspondence and requests for materials should be addressed to K.J. (email: jkm728@foxmail.com) or to Y.I. (email: ito.yoshikazu.ga@u.tsukuba.ac.jp) or to Y.O. (email: oyama@material.tohoku.ac.jp)

Despite its limited capacity (maximum of 372 mAh g$^{-1}$ by forming the so-called LiC$_6$ intercalation compound[1]), graphite has many excellent properties and therefore has been regarded as the state-of-the-art anode material in rechargeable lithium ion batteries (LIBs)[2]. Before its commercial application in 1990[2], intensive studies had been performed to determine its related Li-storage process/mechanism[3,4]. Examples include the classical pleated-layer model by Daumas and Hérold in 1969, and the earlier Rüdorff model in 1965[5–7]. Recently, in the wake of the discovery and development of graphene (i.e., "monolayer graphite") and the numerous reports of enhanced energy-storage performances of graphene-modified electrode materials, many researchers believe that the capacity of LIB can be significantly enhanced through simply replacing the traditional graphite anode by this ultimate carbon material that has metal-level conductivity, large surface area (2630 m$^2$ g$^{-1}$ in theory), and possibly two exposed sides to adsorb Li atoms (namely forming the expected Li$_2$C$_6$ stoichiometry with a doubled capacity of 744 mAh g$^{-1}$)[2,8–10]. However, despite the considerable investment in money and time, this target has not been achieved by using pure graphene materials. Even the Li-storage mechanism (or the storage locations) on graphene is still up for debate both experimentally and theoretically[9–12]. For instance, based on density functional theory (DFT) calculations[1,11–14] and the different electrochemical behavior of monolayer graphene supported by Cu foil from those of multilayer graphenes/graphite for LIBs[9,11,15], it has been proposed that Li atoms cannot be adsorbed onto pristine monolayer graphene, instead they only intercalate into the graphene interlayer or the interspace between graphene and substrate through edge planes or high-order defects (e.g., divacancies). The understanding and resolution of this problem are rather essential for the development of graphene materials in LIBs.

Taking the number of graphene sheet layers (denoted as $n$) into account, the considerable difference between the deduced theoretical capacities assuming the above opposite viewpoints (in particular with $n \le 5$, Supplementary Fig. 1a) should make it easy to know which Li-storage situation is possible for pure graphene ($n = 1$). However, basic requirements for graphene in LIBs, such as a sufficient monolithic mass and absence of extrinsic interference from the as-applied substrates (e.g., planar Cu or Ni foil[9,11,15], Supplementary Fig. 1b, c) make it impractical to use pure 2D graphene sheet with a negligible mass (density: 0.77 mg m$^{-2}$) to address this issue.

In this study, through developing a high-temperature-switched chemical vapor deposition (CVD) route[8,16,17] (Fig. 1a and Supplementary Fig. 2), we have successfully synthesized a bilayer graphene foam dominated by large enough basal plane of graphenic carbon with few defects and high electronic quality. The foam is ca. 30 μm thick after removing the Ni template and

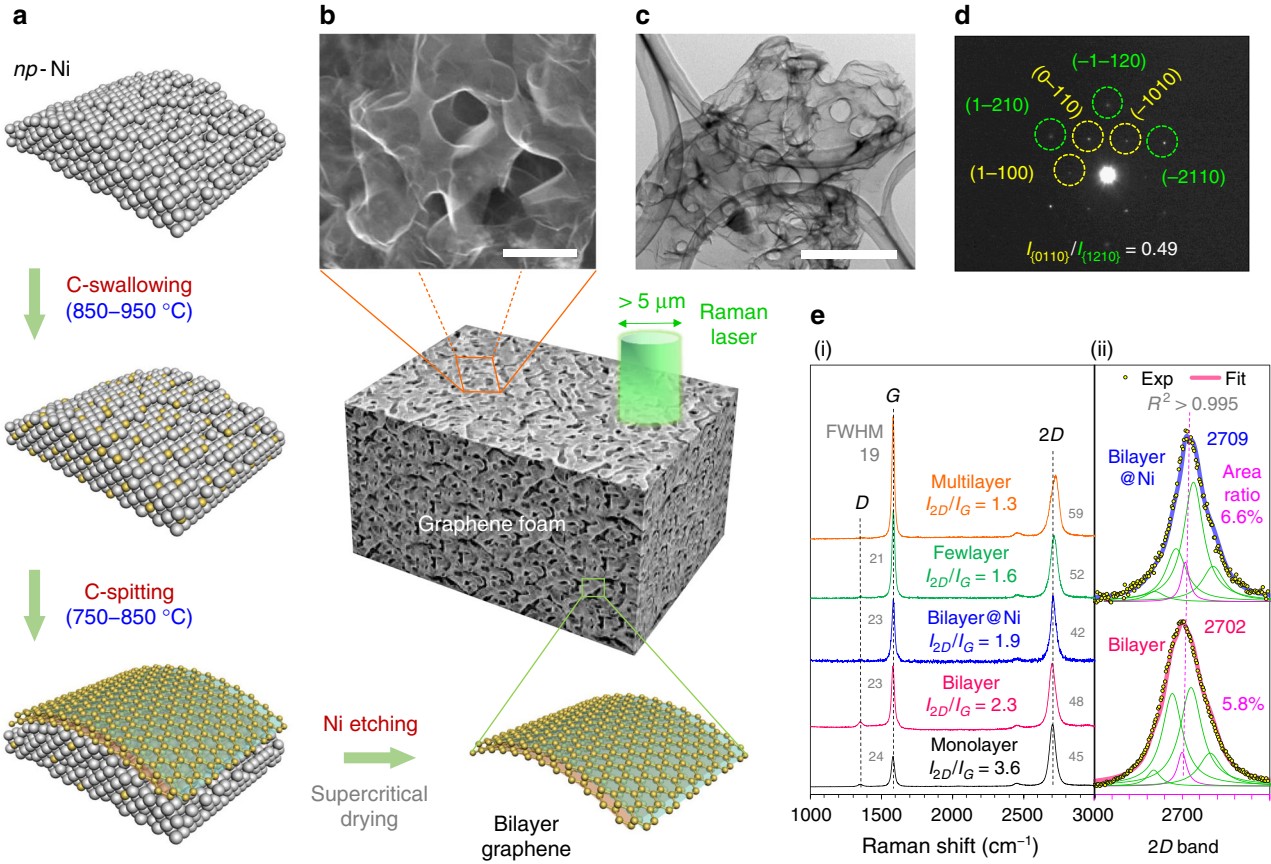

**Fig. 1** Preparation and identification of bilayer graphene with 3D porous morphology. **a** Schematic of high-temperature-switched CVD process by using 3D nanoporous nickel (np-Ni) substrate. Only a small area of the ligament is presented in illustration. **b, c** Scanning electron microscopy (SEM; scale bar, 500 nm) and transmission electron microscopy (TEM; scale bar, 1 μm) images of the freestanding bilayer graphene exfoliated from Ni foam. **d** SAED pattern in the flat region of the bilayer graphene foam. The Miller–Bravais indices (hkil) are used to label the typical sixfold Bragg reflections. **e** Typical Raman spectra captured from 3D porous graphene foam, which may possess different numbers of graphene layers in the local areas. All the $I_{2D}/I_G$ ratios and the values of 2D and G bandwidth (presented as full width at half maximum-FWHM) are shown in (i). The Lorentzian fitting analyses of the 2D bands of the two bilayer-featured Raman spectra are shown in (ii). For either fit, the four green peaks refer to the four components of the 2D band in Bernal stacking bilayer graphene when the single pink peak represents the contribution of monolayer or misoriented/incommensurate regions

supercritical drying to prevent the collapse of the 3D porous structure and thus the restacking of graphene sheets (Fig. 1b–e and Supplementary Fig. 3). Such a 3D porous monolith as a model system not only meets the basic test requirements, but also possesses the well-known resistance against the ubiquitous aggregation and restacking of graphene sheets[17,18]. Consequently, by systematically studying various physiochemical phenomena of Li storage in it, we have clarified the elementary storage pattern of Li atoms in the graphene interlayer to solve the above problem.

## Results

**Quality assessment of the 3D bilayer graphene foam.** As shown in Fig. 1b, c and Supplementary Fig. 3a–c, the as-synthesized bilayer graphene foam well duplicated the interconnected porous structure of the Ni template[17] (Supplementary Fig. 2d, residual Ni < 0.08 at.%) with pore size between 500 nm and 1 μm. The selected-area electron diffraction (SAED) patterns (Fig. 1d and Supplementary Fig. 3d, e) further indicate that this crystalline carbon material should be bilayer graphene[18–20]. In particular, the intensity ratio of the {0110} to {1210} peaks in the sixfold symmetric pattern (corresponding to ca. 0.212 and 0.123 nm in the in-plane lattice spacings, respectively) was measured to be about 0.49, in good agreement with the computational and experimental values of $I_{\{0110\}}/I_{\{1210\}} \approx 0.4$–0.5 for the Bernal (AB) stacking bilayer graphene[19–23]. By distinguishing the height, width, Lorentzian fits, intensity, and chemical-shift features of the 2D and G bands, Raman spectroscopy can provide more definitive identification for graphene materials with diverse numbers of layers regardless of the macroscopic appearance[17,21,23–27], as displayed in Fig. 1e(i) for this kind of porous graphene materials. Nevertheless, it is impractical to use current analytical techniques to exactly distinguish the mono-, few-, and multilayer fractions in this sponge-like 3D graphene (Fig. 1a). Hence, a large number of Raman spectra (with the laser spot diameter of >5 μm, namely covering at least $4 \times 10^8$ $C_6$-ring units with an area of 0.052 $nm^2$) were randomly collected from different locations on the surface and cross-section of the as-prepared samples (before and after removing the Ni substrate). In this way, the bilayer-dominated region in our target material was estimated to be over 90%[17,20,24,25]. Moreover, the specialized Lorentzian fitting analyses of the 2D bands in Fig. 1e(ii)[23,26], together with the matched FWHM and $I_{2D}/I_G$ values (i.e., peak area ratios)[27], can help confirm the predominant Bernal stacking configuration of this CVD-grown bilayer graphene foam. This is also in line with subsequent X-ray diffraction (XRD) results showing an interplanar spacing $d_{002}$ of ~3.35 Å (ca. 3.58 Å at the AA stacking mode (ref. [28]—this study reports that, "the equilibrium interplanar distance in graphite changes from 3.42 Å with AB stacking to 3.58 Å for AA stacking," and that "in the first-stage Li-GICs, the AA stacking sequence has lower total energy than the AB," for which the interplanar distance of $LiC_6$ may vary at 3.62–3.74 Å)). The weak D band from the Ni-free bilayer graphene indicates the mere existence of some intrinsic structural defects originating from the edge planes, grain boundaries, vacancies, or geometrical curvatures of the graphene sheets[1,2,4,11,12,17]. These defective sites on graphene are essential in coordinating the 3D nanoporosity[17] as well as suppressing the aforementioned restacking effect[18,29]. Combining the preparation mechanism and the above characterization results, we presume that such bilayer graphene foam can help elucidate the Li-storage mechanism in graphene-based electrodes.

**Determination of Li-storage capacity and Li-bilayer graphene phases.** Using Li metal as the counter negative electrode and the mixed electrolyte of $LiPF_6$ in ethylene carbonate/dimethyl

carbonate in R2032-type coin cells, the common half-cell configuration[2] was adopted for the electrochemical characterizations of the freestanding bilayer graphene foam. For graphite-based LIBs, it is well known that (1) the insertion/deinsertion potential of $Li^+$ ions is always below 0.3 V (vs. $Li^+/Li$)[2,11,30,31], a range that contributes to the overwhelming capacity of graphite (close to 372 mAh $g^{-1}$), and that (2) the inevitable solid-electrolyte interphase (SEI) is usually generated at above 0.5 V during the initial charging–discharging cycles[30,32–34]. Both the highly reversible galvanostatic charge–discharge (GCD) measurements (Fig. 2a inset and Supplementary Fig. 4) and cyclic voltammograms (CVs, Supplementary Figs 5 and 6) demonstrated that this bilayer graphene foam and graphite electrode share similar potential plateaus/redox peaks[31]. However, its maximum GCD capacity only reached ~180 mAh $g^{-1}$ at 50–0.2 A $g^{-1}$ (Fig. 2a), suggesting that the interlayer should be the sole space for bilayer graphene to store Li atoms to generate stoichiometric $LiC_{12}$, rather than $Li_3C_{12}$ at 558 mAh $g^{-1}$ (Supplementary Fig. 1a). The increased but still low capacities of the few- and multilayer samples (Supplementary Fig. 7a, b) confirmed this observation, and so did the CV-based charge quantities (ca. 6000–7000 C $mol^{-1}$ at 0.10 −0.002 mV $s^{-1}$) that are close to the theoretical value of $LiC_{12}$ at 8040 C $mol^{-1}$ (Supplementary Fig. 7c), in good agreement with previous DFT predictions[1,11–14].

Further, analysis was carried out in terms of the CVs at low scan rates between 0.25 and 0.001 V (vs. $Li^+/Li$) to gain deep insight into the Li-intercalation behavior in bilayer graphene. Clearly, there are seven intrinsic and quasi-reversible redox peaks in each CV profile (the cathodic peaks are denoted as C1–5 and CI–II on behalf of the Li-intercalation reactions of $Li^+ + e^- + LiC_x \rightarrow LiC_y$, and the five main anodic ones as A1–5 in Fig. 2b and Supplementary Fig. 6a), characteristic of the continuous quasi-equilibrium transformations from the Li-free bilayer graphene to the lithiated graphene phases ($LiC_x$, denoted as P1–5, and PI and PII). Compared to those reported for graphite[30,35,36], other than the emergence of one new pair of redox peaks (namely C1/A1 at the low cutoff potential; Supplementary Fig. 8a, b), the relative integral areas of these peaks (corresponding to the amount of charge transfer) seemed to change for different samples (Supplementary Fig. 8c), suggesting possible influences from the graphene quality, which depends on the specific preparations[4,5,32]. Despite the variability, the ever-present intensity changes in various bilayer samples, together with the later kinetics analysis (Fig. 2d, e), show that the C1/A1 pair should originate from the final Li-plating behavior at the defective sites on the graphene sheets. In other words, the ultimate Li-intercalation product with $LiC_{12}$ stoichiometry still corresponds to the PI phase like the case of graphite[30] rather than the P1 phase. Thus, an approximate Li-intercalation/deintercalation process for bilayer graphene could be obtained (Fig. 2b and Supplementary Fig. 7d): $C \leftrightarrows LiC_{122} \leftrightarrows C_{42.7}LiC_{42.7} \leftrightarrows C_{14}LiC_{14} \leftrightarrows C_6LiC_6$, analogous to that estimated for graphite electrodes ($C \leftrightarrows LiC_{72} \leftrightarrows LiC_{36-27} \leftrightarrows LiC_{12} \leftrightarrows LiC_6$[3,30,31,37]), especially considering the staged distribution densities of Li atoms on the $X$–$Y$ plane (Supplementary Fig. 9a, b). That is to say, even without the long-distance $Z$-axis diffusion of $Li^+$ ions in graphite and the possible influence from the neighboring layers[4,5,7,11,14,32], bilayer graphene still exhibits the same Li-storage process as graphite, indicating that the Daumas–Hérold domain model[5,7] should be more suitable for describing the $Z$-axis Li-storage behavior in graphite. Moreover, the planar distribution of these staged Li atoms (Fig. 2c and Supplementary Fig. 9c–e) may indicate that, except for the PI generation, any incoming $Li^+$ ions can enter the centroid of three adjacent Li atoms in the former phase without disturbing their initial locations too much. In other words, the whole Li-storage process needs to go through four-time fractal-way

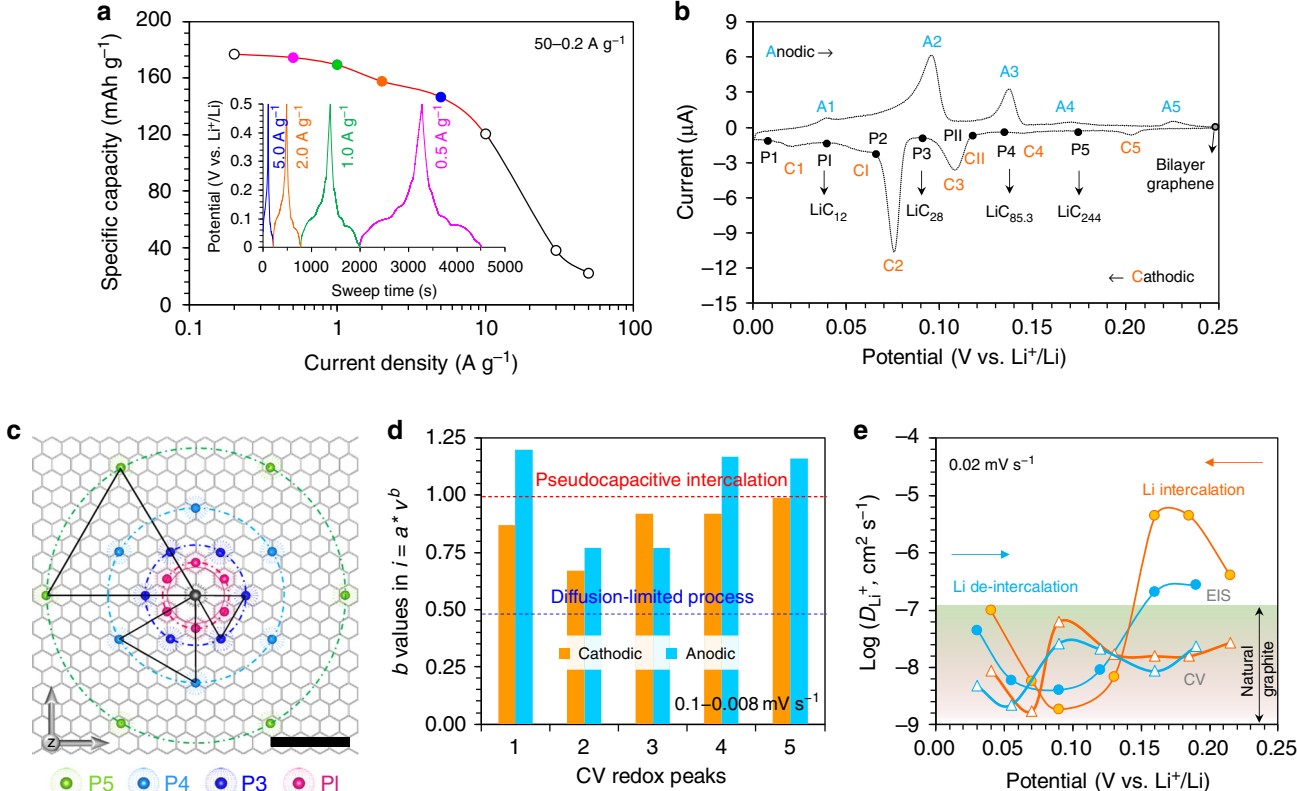

**Fig. 2** Electrochemical characterizations of the bilayer graphene electrodes in LIBs. **a** Specific capacities measured by the GCD test (illustrated in the inset). **b** Definition of the redox peaks observed from the CV curves and the corresponding $LiC_x$ phases. **c** Possible in-plane distribution of Li atoms in the main Li-graphene phases. For simplicity, each phase was represented by seven Li atoms on one graphene network as a reference (scale bar, 1 nm), and the equilateral triangles indicate their spatial relationships in two adjacent phases. **d** Kinetics analysis of the $b$ values for the cathodic and anodic peak currents at CV sweep rates from 0.1 to 0.008 mV s$^{-1}$. **e** Apparent chemical diffusion coefficients of Li$^+$ ions between two successive Li-graphene phases based on the EIS (●) and CV (△) methods

Li intercalations (Supplementary Fig. 9d), and the graphene interlayer just plays as a molecular machine to precisely control each Li-intercalation step.

**Kinetics analysis of the Li-storage process.** According to the power-law relation ($i = av^b$) between the CV peak-current response ($i$) and the sweep rate ($v$), the pseudocapacitive intercalation for Li$^+$ ions, namely the surface-controlled behavior with $b \approx 1.0$ at the initial C5–C3 and unique C1 stages, tended to become slow solid-state diffusion (like the case in a battery with $b \approx 0.5$) for the PI generation (Fig. 2d and Supplementary Fig. 6e, f)[38–40]. The apparent diffusion coefficients of Li$^+$ ions at various stages ($D_{Li^+}$, Fig. 2e), ca. $10^{-9}$–$10^{-7}$ cm$^2$ s$^{-1}$ by the well-known CV method[40] and $10^{-9}$–$10^{-5}$ cm$^2$ s$^{-1}$ by the Warburg impedance method (Supplementary Fig. 10)[35,40–43] (ref. [43]— using an "open" on-chip electrochemical cell to exclude the electrolyte's influence, the study measured the Li diffusion coefficient into bilayer graphene flake with locally varying Li densities, and provided real-time evidence that the Li intercalation/diffusion only occurs at the graphene interlayer), further verified this rate-determining step. These values and their trends are also similar to those of natural graphite electrodes ($D_{Li^+} \approx 10^{-9}$ $-10^{-7}$ cm$^2$ s$^{-1}$)[32,35,41–45]. Obviously, it is the planar distribution density of Li atoms, not the number of graphene layers, that intrinsically determines the intercalation kinetics of Li-graphene/graphite products, in line with the above-mentioned 2D model (Fig. 2c). Therefore, for either bilayer graphene or thick graphite, the interlayer should just provide a "specially restricted space"[7,44,46,47] to store Li atoms regardless of the number of graphene layers[9] or material morphology, fundamentally illustrating and complementing the Daumas–Hérold model[5,7] from the 2D perspective. By the way, in addition to the intrinsic limits in capacity and mass transfer, the high conductivity of graphene material also cannot endow LIBs with superior power compared to graphite, due to their similar high system resistances dominated by the same SEI (Supplementary Fig. 11)[42].

**Structural information of Li-bilayer graphene phases.** To verify the 2D model, we further investigated the main phases by multiple in situ and ex situ techniques. It is important to emphasize that, with respect to the angstrom-sized $LiC_x$ units and the tiny $C_6$-ring unit (refer to the hard sphere model for $LiC_6$ of graphite in ref. [7] and the Discussion section), all the characterization results are statistical and thus can reflect the macroscopic features and general phenomena. Firstly, the highly reversible in situ Raman spectra[9] for this graphene electrode (Supplementary Fig. 12) displayed no noticeable variation in the defect-related $D$ band, in contrast to $G$ band, which kept changing according to the Li concentration. This clearly illustrates the high stability of both the original defective and defect-free regions on the graphene sheets, removing concerns about the possible impact of regenerative defects on the Li-storage performances (refer to Supplementary Fig. 1d)[1,11,32]. Then, the combined characterization results using the scanning electron microscopy (SEM), ex situ Raman spectroscopy, transmission electron microscopy (TEM), and electron energy loss spectroscopy (EELS) techniques

(Supplementary Figs 13–15) show that, all the lithiated graphene electrodes (even those oxidized in air) preserved the foam and bilayered architecture (irrespective of the stacking order) without Li-dendrite or graphene restacking problems. This ulteriorly demonstrates the merit of this 3D porous morphology and the protection of the internal Li layer (denoted as $R$) by the outer graphene sheets in the C$R$C-stacking configuration[1,8,31]. Furthermore, according to the simultaneous XRD patterns of the typical bilayer and multilayer electrode samples (Supplementary Fig. 16a–c), in the wake of Li intercalation (along with the transition from Li$^+$ (radius: 0.76 Å) to Li atom (radius: 1.52 Å))[7,28], the $d_{002}$ value (~3.35 Å, similar for graphites possessing the predominant AB stacking configuration[7,31,45]) seemed to increase slightly in the foregoing P5−P3 phases of A$R$B stacking (<0.02 Å), only relying on a small population of Li atoms. However, after the unstable P2 phase[48] with a theoretically transitional stacking configuration from A$R$B to A$R$A[28] (Fig. 2b and Supplementary Figs 8 and 16), it appeared to become 3.64 Å for the bilayer PI sample (namely the A$R$A-stacking $C_6LiC_6$, Supplementary Fig. 16b). This was smaller than the multilayer (~3.70 Å, similar to that for $LiC_6$ of graphite electrode[28,48], Supplementary Fig. 16c), but larger than that of the pristine AA stacking graphite (~3.58 Å)[28]. These inequalities likely result from the varied electrostatic repulsions between neighboring graphene layers bonding with the interbedded Li atoms of different volume concentrations (namely Li/C atom ratios, Supplementary Fig. 16d), which can account for the above slight $d_{002}$ increase at the initial A$R$B mode, too. The overall increasing trend of this average spacing upon Li intercalation is consistent with literature reports for graphites[3,4,10,32,41,45,48] and was further shown by the SAED results for Li-bilayered products (from <3.42 Å at P4 to >3.50 Å at P1, Supplementary Fig. 17a, d)[28]. The in situ Raman study (see its detailed analysis in the supplementary information) and the following ex situ XPS results also favor these observations and discussions. That is to say, Li intercalation into the interlayers of various graphenic carbons not only follows similar mechanisms in the 2$D$ plane (Fig. 2c), but also along the $Z$-axis (Supplementary Fig. 16d), further supporting the Daumas–Hérold model[5,7].

Specially, we investigated the normal-incidence SAED patterns[19,46] of four lithiated bilayer graphene phases (i.e., P4, P3, PI, and P1 in Fig. 3 and Supplementary Fig. 17). Except P3 (to be explained later), all the other samples showed recognizable {0110} and {1210} peaks with the typical sixfold symmetry of pristine graphene[19,20,46]. It is worth mentioning that, the high-energy electron beam irradiation, even after a quite short time, can easily influence the initial positions of Li atoms in the interlayer by forcing them into the honeycomb lattices of the bottom graphene sheet (Supplementary Figs 18 and 19). In this way, both the originally uniform Li distribution and standard graphene stacking order are disturbed over some local regions or even the overall selected area, resulting in changing diffraction patterns for each sample. This irradiation effect is reflected by contradictions between several observed values/phenomena, including: total $I_{\{0110\}}/I_{\{1210\}}$ ratios and the branched diffracted intensities for the P4 phase with a low Li population (see the two insets taken along the red lines marked in Fig. 3a(i) and 3b(i) vs. the case in Fig. 3b(iii)); the emergence of paired orthohexagonal diffraction patterns with an arbitrary rotation angle but varied spot brightness (on behalf of incommensurate/misoriented stacking configuration (Fig. 3b(i, iii) and Supplementary Fig. 17); the changing $I_{\{0110\}}/I_{\{1210\}}$ ratios listed in Fig. 3a(i, iii) and 3b(i, iii) and Supplementary Fig. 17d; and the disparate in-plane lattice spacings for the Li-inlaid (with a ~3% lattice expansion[28]) and Li-free graphene sheets at the highly lithiated PI or P1 phases (Fig. 3c and Supplementary Figs 17c, d and 18). Despite the physical interference, quickly captured initial patterns successfully

showed the variation trend of the $I_{\{0110\}}/I_{\{1210\}}$ ratio during Li intercalation (from pristine 0.49 to 0.53 and finally to 1.4, Figs. 1d and 3a), which vividly displays the configuration transformation from the AB stacking mode of bilayer graphene to the theoretical A$R$A state of fully intercalated bilayer graphene/graphite[32], as evidenced by the computer simulation in Fig. 3d and Supplementary Fig. 20 and Table 1 (e.g., with the $I_{\{0110\}}/I_{\{1210\}}$ ratios at 0.44 (1.36) and 0.40 (1.17) for A–B (A–A) stacking $C_6C_6$ and $C_6LiC_6$, respectively). In particular, for the P3 phase with a large enough but still unsaturated Li concentration, the plentiful vacancies would make the initial movements of its interlayered Li atoms rather disordered (Supplementary Fig. 19a), which can drive the local-region graphene sheets to rotate arbitrarily and thus bring about various graphene stacking configurations over the selected area, as reflected by the particularly ambiguous sixfold-symmetry diffractogram of both the bilayered and multilayered P3 samples (Fig. 3a(ii) and Supplementary Fig. 17b, e). In fact, a similar situation was proclaimed before for the Li-graphite product at this stage, which was regarded to possess an indeterminate or possibly liquid-like structure (i.e., lacking in-plane ordering of Li)[3,35,44,45,48], supporting our observation and discussion here. After full irradiation, a sufficient number of Li atoms would be driven into the graphene honeycomb lattices, causing a remarkable lattice expansion at P3 (ca. 6–8% in experiment and 6.5% by the DFT simulation, Fig. 3b(ii, iv) and 3c and Supplementary Figs 17b,e and 19b). Thus, these comparative but regular extrinsic SAED patterns can further illustrate the intrinsic C$R$C configurations of the generated Li-bilayer graphene phases in the LIB.

**Chemical compositions of Li-bilayer graphene phases**. In view of the roughly equal concentrations (at 50 ± 3 at.%) and the common source of their O components (namely $Li_2CO_3$ or lithiated hydrocarbon $R'$-$CH_2OCO_2Li$ in the surface SEI film[49], Supplementary Fig. 21a, b), before analyzing the chaotic raw X-ray photoelectron spectra (XPS, analytical range diameter >10 μm), the O 1s, C 1s, and Li 1s species originating from SEI in each phase were assumed to share the same binding energies (BEs) at ca. 531.8, 289.9, and 55.0 eV (marked by blue in Fig. 4a–d and Supplementary Fig. 21c, d)[15,50], respectively. As a result, one can see very regular and parallel evolutions for almost every species along with the Li intercalation (where the practical P2 and P1 samples with exceptional and analogous profiles were inferred as mixtures, Supplementary Fig. 21), clearly implying the charge overlapping or bonding interaction between the interbedded Li atoms and the C atoms arranged on a perfectly honeycomb lattice[11,14,46]. Specifically, for the C 1s spectra (Fig. 4a, b), other than two inherent peaks derived from the (lithiated) graphene body (at ca. 284.9 and 286.6 eV, respectively), there were still three (or more) peaks with lower BE values in the lithiated products. This splitting phenomenon was considered to be associated with the various distances between the imbedded Li atom and its adjoining but nonequivalent C atoms[23,50], as confirmed by our DFT calculations here taking the A$R$B, A$R$A, and SP configurations of $C_6LiC_6$ into account (Fig. 4e). A similar splitting phenomenon occurred in the Li 1s spectra (Fig. 4c, d, f). Therein, the peaks at higher BE (around 53 eV) were assigned to the Li atoms exhibiting some ionicity (like in the A$R$A or SP mode), when the peaks at 52–51 eV were ascribed to metallic Li species (like in the A$R$B mode)[11,12,49,50]. Thus, from the point of view of chemical composition change, the combined experimental and theoretical results further reveal and verify the phase-wise stacking configurations suggested by the XRD and SAED characterizations performed at different analytical ranges. This is seen in: the maintained A$R$B structure at P5 ($C_{122}LiC_{122}$, showing

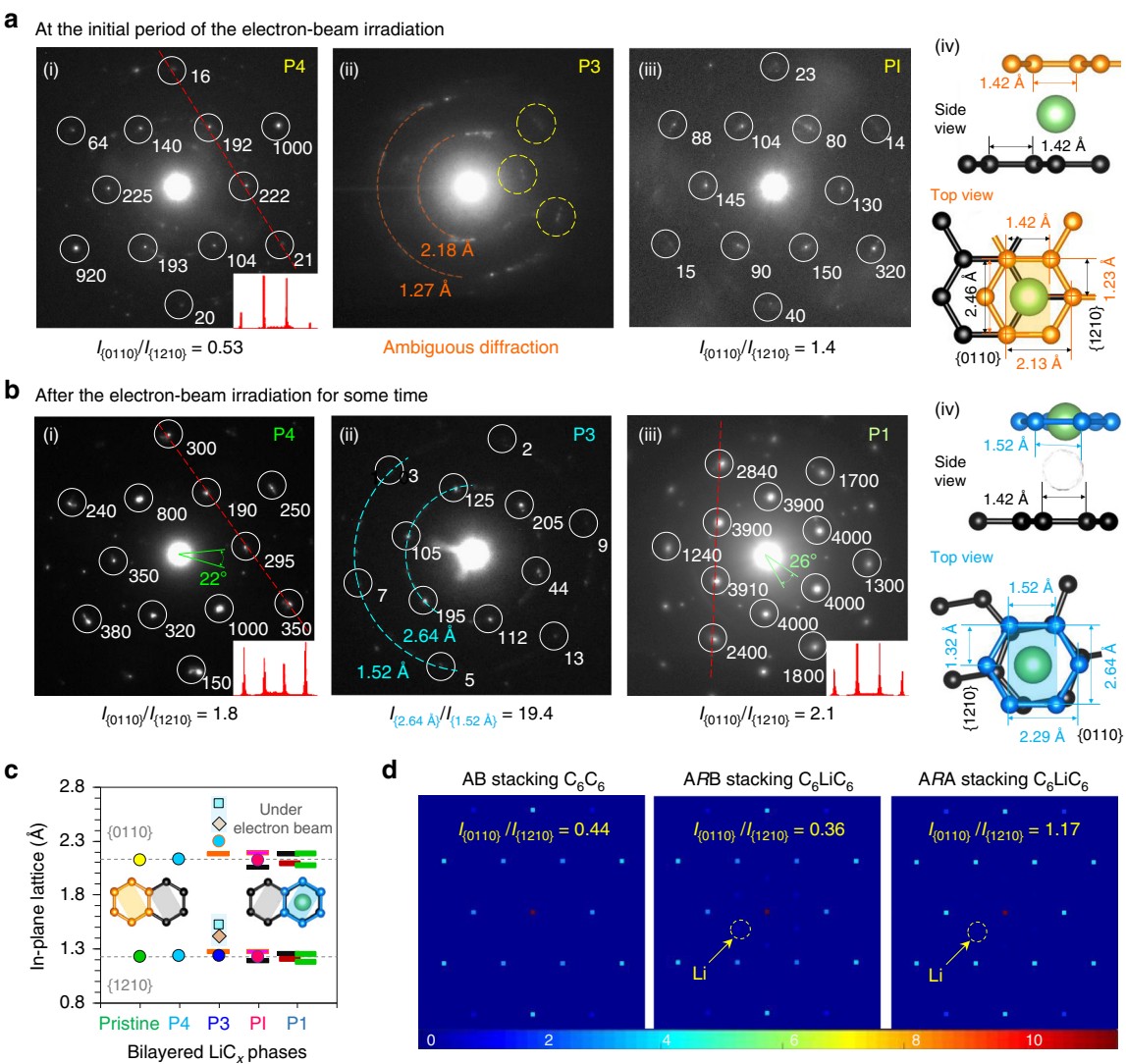

**Fig. 3** Identification of the SAED patterns for the Li-bilayer graphene phases. **a**, **b** Normal-incidence SAED patterns captured after different irradiation time for the P4 (i), P3 (ii), and PI/P1 (iii) samples. The white circles are applied to geometrically orientate the target diffraction spots with sixfold symmetry, and the numbers refer to their individual brightness values to calculate the $I_{\{01\bar{1}0\}}/I_{\{1\bar{2}10\}}$ intensity. The two (iv) illustrations show the intrinsic and electron beam-induced stacking configurations based on the P3 phase (refer to the arbitrary commensurate rotations between the two hexagonal patterns shown in **b**(i, iii) and Supplementary Fig. 17b–d), respectively. The inlaid Li atoms in the honeycomb lattices would enhance the diffraction intensity of the shaded planes compared to the $\{01\bar{1}0\}$ and $\{1\bar{2}10\}$ planes (**a**(ii) and **b**(ii)), and finally cause a 7% expansion of the $C_6$-ring unit (refer to **b**(ii, iv) and the shadow-marked lattice parameters in **c**). **c** In-plane lattice spacings of the electrode materials (Supplementary Fig. 17a–d) vs. the theoretical spacings of the $\{01\bar{1}0\}$ and $\{1\bar{2}10\}$ planes (refer to **a**(iv)). **d** Simulated brightness of the electron diffraction patterns for pure and Li-saturated bilayer graphenes with different stacking modes (Supplementary Table 1)

predominant metallic Li 1s and relatively few C 1s species) and P4 ($C_{42.7}LiC_{42.7}$, possibly with two Li-embedded sites according to Fig. 4e, f and the Discussion below); the coexisting A$R$B and SP (unstable) modes at P3 ($C_{14}LiC_{14}$) with a large enough Li concentration (Supplementary Fig. 16); and the A$R$A mode at PI ($C_6LiC_6$) with ionic Li 1s species of high BE (Fig. 4d, f, with the electrostatic charge of Li possibly at +0.9|e| in the $C_6$-Li constitutional unit)[11,14,28,47] and a simultaneous nearly single C 1s species. In brief, upon the Li intercalation, the holistic ionic character of Li atoms is enhanced along with the stacking configuration transformation, while the metallic character decreases (Fig. 4d, f), providing an explanation for the regular variations of the measured electrochemical impedance/condensance (Supplementary Fig. 11b, c)[38,42]. Besides, the Li-plating behavior at the limited defective sites seemed to lead P1 to deviate

from the normal A$R$A configuration (Supplementary Fig. 21 and refer to its other characterization results in Supplementary Figs 13−15), suggesting the weak (van der Waals-like) out-of-plane interaction responsible for the relative slipping of adjoining graphene sheets in achieving the configuration transformation[46]. Thus, by linking the regular evolutions of C 1s, Li 1s, and O 1s spectra, it is reasonable to imagine a continuous configuration transformation of bilayer graphene as the Li intercalation progresses (see below).

## Discussion

The electrochemical CD and CV measurements (Fig. 2a, b) initially suggest the C$R$C-stacking configuration and stoichiometric $LiC_x$ compositions of the staged Li-intercalated bilayer graphene phases. Then, the geometric and kinetics analyses

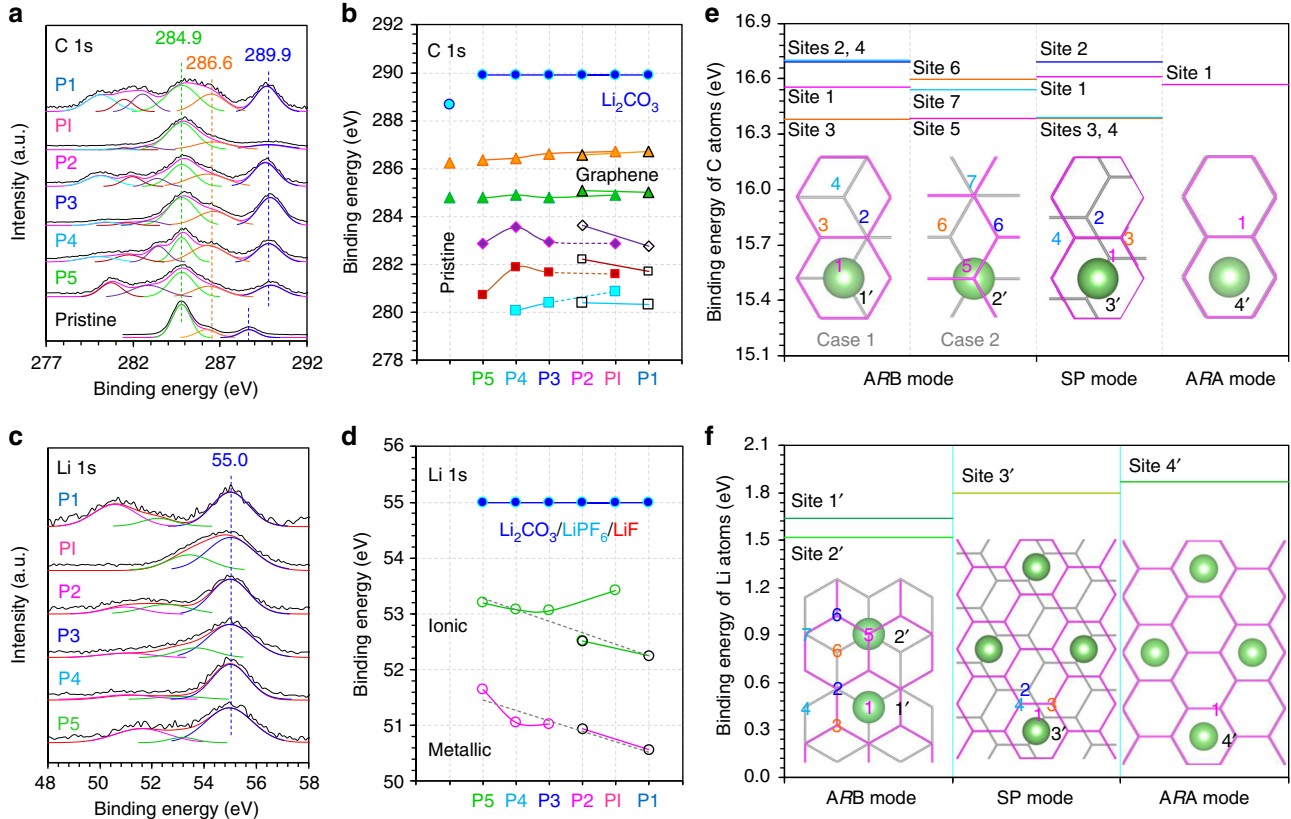

**Fig. 4** Composition identification of the bilayered $LiC_x$ phases during the Li-intercalation process. **a**, **b** C 1s XPS spectra and evolution of the BEs of C 1s species. **c**, **d** Li 1s XPS spectra and evolution of the BEs of Li 1s species. The enhanced peak intensities at P2 and P1 were related to the mixed components of their samples in reality (Supplementary Figs 16 and 21). **e**, **f** DFT-simulated BEs of C and Li elements in different stacking configurations of $C_6LiC_6$. Sites 1−7 and 1'−4' indicate the locations of C and Li atoms in different coordination environments (inset images: top view), respectively. By comparison, Li atom tends to locate Site 1' in the ARB mode, and the ARA configuration is more energetically favorable than the SP mode

(Fig. 2c–e) reveal a 2D model for Li storage in the interlayer of graphenic carbon. Furthermore, the subsequent ex situ and in situ characterizations clearly display the structural evolution induced by Li intercalation and dependent on its staged concentrations, in terms of related physicochemical phenomena, such as (1) the increasing interlayer spacings (Fig. 5a) depending on the charge amounts of neighboring graphene layers and, in particular, on their stacking orders by the phase-wise XRD and SAED patterns (Supplementary Figs 16 and 17a,d), (2) the relatively stable in-plane lattice spacings of graphene sheet by the experimental SAED patterns and computer modeling[28] (Fig. 3c), and (3) the alternating configurations from A–B to A–A stacking modes (Fig. 5b) according to the XRD (002) diffraction peaks (Supplementary Fig. 16), the observed-simulated brightness ratios of the {0110} and {1210} diffraction spots (Fig. 3a, d), and the regular XPS data assisted by the BEs of C and Li elements from the DFT calculations (Fig. 4). All these interrelated and consistent results from a wide range of analyses (also including the in situ Raman spectra and impedance analyses, as well as the other electron microscopy data) indicate that, the integrated Li-intercalation process in Fig. 5 is a reasonable mechanism for the fundamental Li storage into graphenic carbon with few defects. In a word, the $Li^+$ ions entering the graphene interlayer under the electro-dynamic forces tend to diffuse along and become stabilize by $Li^+ + e^- \rightarrow Li$ at those overlapped sites with the most balanced electronic cloud density committed by the staged $LiC_x$ units (Fig. 5b).

In summary, by developing a high-quality freestanding bilayer graphene foam and systematically studying various aspects of

Li storage in it (such as capacity, process, kinetics, and resistances as well as the regular composition and configuration evolution of staged products), we have demonstrated that Li atoms can only intercalate into graphene interlayer and proposed a planar Li-storage model for 2D graphenic carbon material. It is found that there is no fundamental difference between bilayer or few-layer graphenes and graphite electrodes in the Li-storage manner and kinetics behavior, and the whole Li-storage process goes through four-time fractal-way Li intercalations to achieve the saturated composition of $LiC_6$, with the final step the rate-controlling step. This study not only clarifies the Li-storage mechanism for commercial graphite anodes, but also highlights the potential of defect-free graphene in LIBs to guide its development in the energy-storage field.

## Methods

**CVD-preparation of nanoporous graphene foam**. Nanoporous Ni foam, by directly dealloying the $Ni_{30}Mn_{70}$ foil in 2.0 M $(NH_4)_2SO_4$ solution for 10 h, was put at the center of a quartz tube ($\varphi 30 \times \varphi 27 \times 1000$ mm) of the CVD furnace. The initial flow of Ar was 500 sccm until that the temperature reached the low temperature $T_l$ shown in Supplementary Fig. 2b. Before reaching the high temperature $T_h$, the flow of Ar and $H_2$ was gradually increased to be 2500 and 100 sccm, respectively. After the above reduction pre-treatment, benzene (0.5 mbar, 99.8%, anhydrous) was introduced with the gas flow of Ar and $H_2$ for graphene growth. For bilayer graphene, the initially total pressure and the partial pressure at the tube's front end should be controlled at 5–6 and 0.5–0.7 Mpa, respectively. The minimum $T_l$ value was determined to be 700 °C, below which a large amount of defects would generate on the graphene sheets. The furnace was immediately opened to let the inner quartz tube rapidly cool to room temperature with a fan (in 15–20 s from $T_l$ to 400 °C, Supplementary Fig. 2a, b). The nanoporous Ni substrate of the graphene@$np$-Ni composite was dissolved by 2.0 M HCl and the as-obtained initial sample

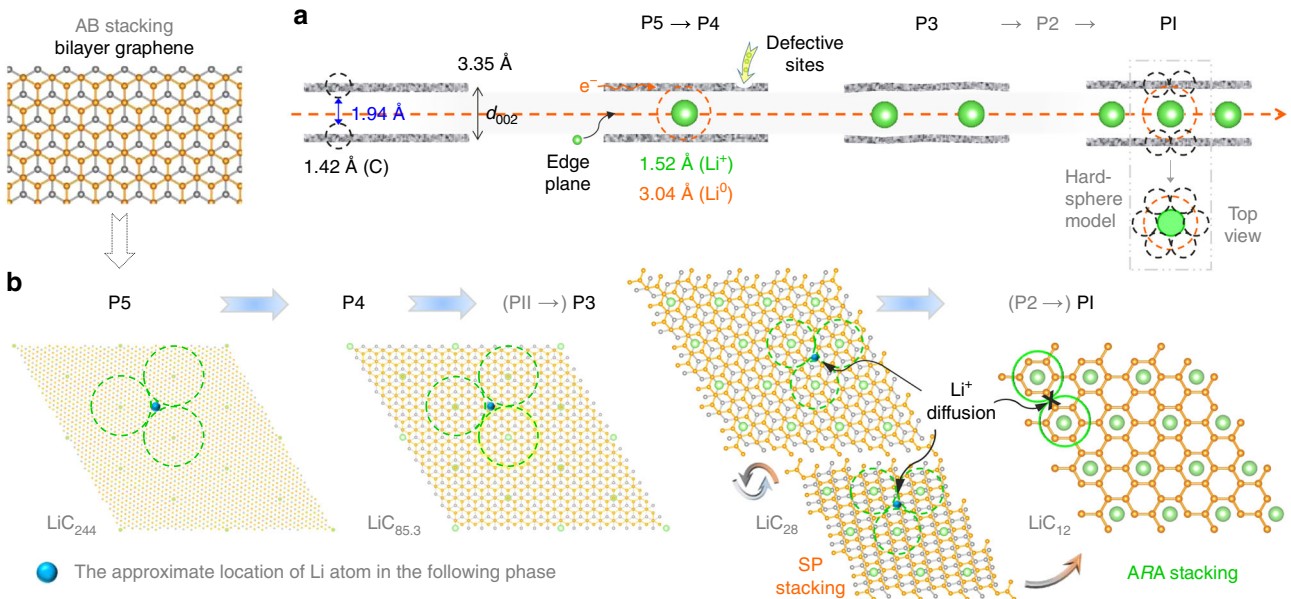

**Fig. 5** Schematic of Li intercalation into bilayer graphene. **a** Evolution of the interplanar spacing $d_{002}$ of bilayer graphene induced by the staged Li intercalation (side view). Empirical diameters of C and Li atoms and Li$^+$ ion are employed in the hard sphere model for the $C_6$-Li constitutional unit. The stacking configuration transformation from A$R$B ($d_{002} = 3.35-3.37$ Å) to A$R$A ($d_{002} = 3.64$ Å) occurs at the transitional/transient P2 stage. **b** Evolutions of the Li distribution in the interlayer of bilayer graphene and the stacking configuration determined by the staged Li concentrations (top view, Supplementary Fig. 9b). The green circles refer to the general sizes of the involved LiC$_x$ units. The P3 phase with a large enough Li concentration possesses a stacking configuration easy to be disturbed

was repeatedly washed by water and isopropanol before the final supercritical $CO_2$ drying process (or the other post-processing process) for the target material.

**Electrochemical measurements**. Coin 2032-type test cells were assembled in a high-purity argon-filled glove box ($H_2O < 0.5$ p.p.m., $O_2 < 0.5$ p.p.m., MBraun, Unilab) with LiPF$_6$ solution of 1 M in ethylene carbonate (EC)/dimethyl carbonate (DMC) (EC:DMC = 1:1 vol/vol) as the electrolyte, the freestanding graphene foams, the graphene@$np$-Ni composites, monolayer graphene on plate Cu foil, bilayer graphene on plate Ni foil directly as the working electrodes, fresh Li foil as the counter electrode, and Whatman glass fiber as the separator. No binder was adopted for the coin-cell assembly. For the graphite electrode supported by Rany Ni current collector or not, polytetrafluoroethene (with the mass ratio to graphite powder at 1:9) dissolved in ethanol was used as the binder. Both GCD and CV measurements were carried out at room temperature on an electrochemical workstation and a battery-measurement equipment. The alternating-current electrochemical impedance spectrum with varied open circuit voltages were recorded simultaneously at a frequency range from 100 kHz to 0.1 Hz and an amplitude of 5 mV. ZView software was applied to simulate the Nyquist plots in view of the reported equivalent electric circuit proposed for graphite electrode[42].

**Characterization of structure and composition**. The microstructure was characterized by both of a field-emission SEM (JEOL JIB–4600F, at an accelerating voltage of 15 kV) and a field-emission TEM (JEOL JEM-2100F, at an accelerating voltage of 100–200 kV) equipped with two aberration correctors (CEOS GmbH) for the image- and probe-forming objective lens systems. The SAED patterns of each sample were recorded from areas of 30–100 nm in diameter. High-resolution EELS spectra of the samples were collected using a Gatan Enfina spectrometer, and energy dispersion was 0.5 eV per pixel. The XRD patterns were collected on X-ray diffractometer (SmartLab) using Cu $K\alpha$ radiation and nickel filter ($\lambda = 0.15406$ nm), and the operating voltage and current were 40 kV and 30 mA, respectively. A micro-Raman spectrometer (Renishaw InVia RM 1000) with laser wavelength of 514.5 nm (excitation energy: 2.41 eV) was used for Raman measurements. The laser power was set at 2.0 mW to avoid possible damage by laser irradiation. The spectrum acquisition time was 150 s. The surface species of the samples were determined by XPS (AxIS-ULTRA-DLD, exceptional small spot capabilities <15 μm) with an Al $K\alpha$ (mono) anode at 150 W in a vacuum of $10^{-7}$ Pa. Both the Raman and XPS spectra were recorded from micron-sized regions.

**Preparation and characterization of Li-graphene phases**. To prepare lithiated graphene samples for the ex situ TEM and XPS measurements, in case of oxidation, each coin cell loading the target Li-graphene product (Fig. 2b) was disassembled in an argon-filled glove box and then the specimen was fully rinsed in pure EC/DMC

electrolyte. After the electrolyte evaporated completely, the as-obtained dry sample was fixed on the specimen holder for the XPS (using conductive adhesive tape) or TEM (using two Cu grids without carbon support film) measurements. Every time the holder was carefully sealed in the glove box before it was rapidly transferred into the testing equipment. After the XPS measurements, the same samples, which had been exposed in air for some time, were further checked by the ex situ SEM and Raman techniques (Supplementary Fig. 13). To prepare lithiated graphene samples for the ex situ XRD characterization (Supplementary Fig. 16), after disassembling their corresponding coin cells in the argon-filled glove box and washing them in the EC/DMC electrolyte, each dried specimen was fixed by Kapton tape on a glass slide. In addition, the XRD measurements were performed again for those samples after removing their surface tapes. A special Li-ion battery liquid cell was designed and assembled in the glove box for the in situ Raman characterization of the electrode material under the CV tests with different sweep rates (Supplementary Fig. 12).

**Electron diffraction simulation for Li-graphene phases**. The multislice method was adopted to simulate the diffraction spot brightness of various graphene-based structures. The scattering factors in ref. [51] was applied to generate projected electrostatic potential of the Li and C atoms. The applied accelerating voltage was set at 200 kV. Standard deviation of the random Gaussian distribution accounting for atom vibration was 0.085 Å, and phonon configuration number was 300. Considering the large area selected for the SAED, the periodic boundary condition was set in order to acquire intense Bragg peaks. Different rectangle supercells were constructed to meet the rectangle dimensional of simulated image size (1024 × 1024). Meanwhile, CaRine Crystallography 3.1 software was applied to simulate the diffraction patterns of various structure models to obtain the information of lattice spacings.

**DFT calculations**. First-principles calculations were performed by using the Vienna ab initio simulation package, based on the spin-polarized DFT. The electron-ion interactions were presented by the frozen core all-electron projector augmented wave pseudopotentials, and generalized gradient approximation of the electron exchange-correlation functional was adopted. The atomic layer distance was fully optimized until the Hellmann–Feynman force was <0.01 eV Å$^{-1}$. In the static calculations of total energy, a $10 \times 10 \times 1$ Monkhorst–Pack $k$ grid was employed and the electron wavefunctions were expanded by using a plane wave basis set with a cutoff energy of 400.0 eV.

On the one hand, to confirm the in-plane lattice-expansion phenomena of lithiated bilayer graphene induced by electron beam irradiation (Fig. 3c and Supplementary Figs 17b and 18), the Li-inlaid unit cells of LiC$_{14}$ sheet and LiC$_6$ sheet with normal in-plane lattice parameters (as well as many other configuration

models by adjusting the $Z$-axis spatial positions of Li atoms) were constructed in the first step. Then, we restricted the $Z$-axis relaxation and merely allowed atoms to relax in the $X$–$Y$ plane (namely to achieve the smallest system energy) to obtain the optimized structure achievable in theory. Finally, the above CaRine Crystallography 3.1 software was applied to simulate the in-plane lattice parameters of the new structure (namely the optimized unit cell) used for comparison with those experimental SAED data.

On the other hand, considering different configured Li sites relative to carbon lattice in the $X$–$Y$ plane, we fixed the interlayer spacing (namely the $d_{002}$ value), restricted the in-plane relaxation, and merely allowed individual atoms to relax along the $Z$-axis direction in the binding-energy calculations (Fig. 4). The lithium BE ($\Delta E_{Li}$) was calculated as

$$\Delta E_{Li} = E_{graphene+(n-1)Li} + \mu_{Li} - E_{graphene+nLi},$$

where $E_{graphene+(n-1)Li}$ and $E_{graphene+nLi}$ are the total energies of the bilayer graphene with $(n-1)$ and $n$ Li atoms interlaminated between two graphene atomic layers. $\mu_{Li}$ is the average Li chemical potential calculated as the unit cell merely contains isolated Li atoms. As for the BE of different equivalent C sites in different stacking configurations with Li interlaminated, its value was calculated as

$$\Delta E_{carbon} = E_{graphene+nLi+C_V} + \mu_C - E_{graphene+nLi},$$

where $E_{graphene+nLi+C_V}$ is the total energy of the Li-interlaminated bilayer graphene with a vacancy of the specific C site ($C_V$). Similarly, $\mu_C$ is the average C chemical potential calculated as the unit cell merely contains the specific isolated C atoms.

## Data availability

The authors declare that the major data supporting the findings of this study are available within the paper and its Supplementary Information. Extra data are available from the authors upon reasonable request.

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

## Acknowledgements

This work was sponsored by JSPS Grant-in-Aid for Scientific Research on Innovative Areas "Discrete Geometric Analysis for Materials Design" (grant number: JP18H04477), JSPS KAKENHI (grant numbers JP16J06828, JP17H01325, JP15H05473, JP18K14174, JP26107504, JP23246063, and JP15H02195), JST-PRESTO "Creation of Innovative Core Technology for Manufacture and Use of Energy Carriers from Renewable Energy" (JPMJPR1541), and the fusion research funds of "World Premier International (WPI) Research Center Initiative for Atoms, Molecules and Materials" from the Ministry of Education, Culture, Sports, Science and Technology (MEXT), Japan. This work was also partially sponsored by the THz evaluation group under the Center of World Intelligence Project for Nuclear S&T and Human Resource Development by MEXT, Japan. K.J. was sponsored by the JSPS fellowship program (JP16J06828), University of Tsukuba Basic Research Support Program Type S, and the Mukai science and technology foundation. We thank Professor M.W. Chen at AIMR and Ms. Kazuyo Omura at the Institute for Material Research in Tohoku University for experiment assistance. We also appreciate the comments by Professor Takashi Kyotani and Professor Hongmin Zhu at Tohoku University.

## Author contributions

K.J., J.H., H.K., and Y.I. prepared the graphene materials. K.J. carried out the electrochemical measurements and the Raman, XRD, and SEM characterizations. J.H. designed the liquid-cell device for the in situ Raman study. J.H. and K.J. prepared TEM and XPS test samples. T.F., A.H., P.L., Y.T., and J.F. contributed to the TEM/SAED characterizations. Y.S. and K.J. carried out the DFT simulations and the optimization of structure model. A.H., S.N., and K.J. performed the SAED simulations. K.J. analyzed all the experimental data, conceived the idea, and wrote the manuscript and response letters. K.J. and J.H. contributed equally to the experimental measurements. Y.I., J.H., A.H., and Y.S. commented on the related contents. K.J., J.H., Y.I., and Y.O. contributed to the project.

## Additional information

**Competing interests:** The authors declare no competing interests.

**Journal peer review information:***Nature Communications* thanks the anonymous reviewers for their contribution to the peer review of this work. Peer reviewer reports are available.

