## [Peer Review File · Nature Communications]

Reviewers' comments:

Reviewer #1 (Remarks to the Author):

The paper by Ji and coauthors reports a study of lithium ion intercalation into a high porosity graphenic construct. This material is taken to be representative of bilayer graphene, but with a large enough active mass and surface area that permits conventional electrochemical methods to be used for the study.

The issue of efficient and high-density charge storage remains a critical impediment to the resolution of contemporary problems in renewable energy storage and its widespread utilization. In that regard, studies such as these that probe the mechanism of these complex reactions are important for rational improvements to existing technologies.

The work presented in the paper is overall quite thorough, employing a range of electrochemical, microscopic, and spectroscopic probes to explore the intercalation and deintercalation reactions, together with DFT.

I think this work contains novel findings that will be of interest to the broad readership of Nature Communications and especially researchers in the field of energy storage, but I would request some significant revisions and clarifications before the paper can be considered for publication.

I have detailed the specific concerns I have with the article below:

1. On a general level, I think it is important for the authors to emphasize to a greater extent the eventual proposed mechanism for intercalation in these systems, since after all, this is perhaps one of the most important conclusions from the study. Yet, at present this is not explicitly illustrated. Along these lines, owing to the large number of experiments and results reported here, I think the paper requires some considerable work in organization to streamline the article.
2. I have some trouble understanding the structure of this porous graphenic material. In the schematic of Figure 1a, it looks as though the graphene only covers the outermost layer of the nickel foam, yet I would expect that it would conformally coat all surfaces including the interior pore surfaces. Indeed, that is my interpretation of the description in the text. Perhaps the figure could be improved and clarified.
3. On that note, if the graphene does coat the interior surfaces, to what extent do the researchers observe the layers collapsing on each other to form multilayer regions (4,6,8... layers) after etching the Ni away? Is this effect prevented for some reason? Why is this not observed?
4. I would like the authors to provide more details on how the ~90% bilayer coverage estimation is made.
5. In Supplementary Figure 6, what does "peak emerging order" mean? What is the significance of this?
6. It is not very clear what C1–C5 are as compared to C1 and C11.
7. On page 6, line 122 I think perhaps "parallel" should be changed to "corresponding".
8. In the figures showing the distribution of Li atoms/ions on the layer (Figure 2d and Supplementary

Figure 12), why are some of the Li atoms placed at the vertices or edge-sharing regions between adjacent C6 rings? Is it not accepted that Li atoms will bind in the middle of a C6 ring, as in Figure 3a(iv)?

9. Kuhne et al. reported Li diffusion coefficients up to $7\text{E}-5\text{ cm}^2/\text{s}$ (ref. 40). Could the authors discuss why the Li diffusion here is slower ($1\text{E}-9$ to $1\text{E}-5$)?

10. From Raman spectra data (Supplementary Figure 15a) it looks as though the G and 2D peaks disappear upon intercalation due to Pauli blocking, yet in the intensity evolution plots of Supplementary Figure 15d, this vanishing intensity is not exhibited. Could the authors clarify this?

11. Once the lithiated material is exposed to air, could the authors comment on the chemical and structural integrity/stability of the material that permitted them to acquire electron microscopy data (presumably ex situ)? Or was the sample never exposed to air?

Reviewer #2 (Remarks to the Author):

This communication presents a study of Li intercalation in bilayer carbon foam produced by chemical vapor deposition on a particular type of 3D Ni foam. The authors propose this system to address the question: Can 'lithiation' happen on single layer graphene or only in between graphene layers? The work has some significant deficiencies. Some suggestions are made about how it might be improved.

Q: (General - information) Only relevant information should be presented, it also includes the supplementary information.

Q: (Language problems) Language such as (Line 21) "The real capacity of ideal graphene..." or (Line 30) "The benign development of graphene ..." should be avoided (that is: are the authors wanting to draw attention to the fact that there might be nefarious uses for graphene?). Authors also can only claim they prove something if they present overwhelming evidence (Line 24 "we experimentally prove that ..."). (Instead, should be something like: "Experimental data show...") The authors should not assume that something is "correct" (Line 57), or "a major confusion" (Line 61), or that something is "impossible" (Line 83), based on their personal opinions. Similar language issues such as Line 54 "Logically speaking" (why would anyone speak illogically?) offer further challenges to the reader who is trying to discern what the science that has been done, is. Again: (Line 117) "Despite the too weak (002) peaks ... clearly reveal that ..."; (Line 193) "Despite this interference ...which vividly display". How can too weak peaks clearly reveal anything? How can such a level of interference lead to something, nonetheless, being vividly displayed? [Finally, there is the issue that the authors do not have graphene (as it is typically defined) but instead something that one might refer to as "graphenic carbon" or "sp²-bonded carbon layers in a 3D structure." But the reviewer acknowledges that there is a terminology problem in this field.]

It is suggested that the authors have the manuscript edited by a professional editor.

Q: (Line 102) The evidence that authors provide for no Li⁺ adsorption on the outer plane surface of what they refer to as the 'graphene' is that the charge-discharge capacities are close to ~180 mAh/g (as well as 'small' capacities in some other few-layer graphene tests), which is about half of LiC₆, and is shown as Figure 2a. Please provide more detail about such measurements. For example: show the difference of 1st and later cycles of charge-discharge profiles, and report the Coulombic efficiencies and the irreversible capacities. Capacity loss from the SEI formation and distribution in such 'graphene

materials' could be more serious than in graphite; one reason that high specific surface area carbons have not been effective in Li battery anodes is because of irreversible loss of too much Li in the SEI layer(s) that build up. It has been said that the basal planes of graphitic domains are coated by stable organic-rich SEI (comparing to inorganic-rich SEI regions on edges), and thus the SEI issue needs to be addressed. Provide references that state that it is a non-issue in your particular carbon studied here, or investigate and report about it.

Q: (Line 107) The authors should consider all possible options before choosing one as definitive: a configuration $\text{LiC}_{12}\text{C}_{12}\text{Li}$ (among others) has the same theoretical capacity as C_6LiC_6 .

Q: (Line 111) Provide references here.

Q: (Line 130) $\text{C}_{122}\text{LiC}_{122} \rightleftharpoons \text{C}_{42.6}\text{LiC}_{42.6} \rightleftharpoons \text{C}_{14}\text{LiC}_{14}$

These stoichiometric numbers, or more accurately, as stated in Figure S12 caption,

$122 - 42.7 - 12.7$ (alternative) $- 6$ (little bit off)

Can such numbers lead to a series of planar fractal triangles, if the statistics are not sufficient? It would seem (Figure S9d) that the numbers are actually:

$(186 \sim 270)/2 - (64 \sim 103)/2 - (28 \sim 32)/2 - (12)/2$

Not to mention the variation in the numbers among cells / CV cycle #.

What would the fractal series of 'stoichiometric numbers' be if starting from LiC_6 and how far off are these numbers from the experimental data? Why do the authors particularly report such accurate numbers ($122 - 42.7 - 12.7 - 6$) from their (it would seem, much 'coarser') experimental data, and confirm a fractal model even though the last 2 numbers are "off"?

The authors state that the P3' distribution "may be more exact to describe... LiC_{12} ". Please give references or prove that the P3' distribution is a thermodynamically most stable gallery arrangement of LiC_{12} , which has some Li^+ eclipsing the C atoms in the AA stacking configuration.

Q: (Line 177) The authors state that the gallery expansion from G-G to G-Li-G is from 0.335 nm to 0.336 nm (P2, $\text{LiC}_{12} \sim 28$), according to their XRD result. How is it possible to insert Li^+ in 0.001 nm? (The gallery height of the graphite intercalation compounds (GIC) LiC_x is 0.370 nm).

Reviewer #1: *The paper by Ji and coauthors reports a study of lithium ion intercalation*

into a high porosity graphenic construct. This material is taken to be representative of bilayer graphene, but with a large enough active mass and surface area that permits conventional electrochemical methods to be used for the study. The issue of efficient and high-density charge storage remains a critical impediment to the resolution of contemporary problems in renewable energy storage and its widespread utilization. In that regard, studies such as these that probe the mechanism of these complex reactions are important for rational improvements to existing technologies. The work presented in the paper is overall quite thorough, employing a range of electrochemical, microscopic, and spectroscopic probes to explore the intercalation and deintercalation reactions, together with DFT. I think this work contains novel findings that will be of interest to the broad readership of Nature Communications and especially researchers in the field of energy storage, but I would request some significant revisions and clarifications

before the paper can be considered for publication. I have detailed the specific concerns I have with the article below:

Question 1: *On a general level, I think it is important for the authors to emphasize to a greater extent the eventual proposed mechanism for intercalation in these systems, since after all, this is perhaps one of the most important conclusions from the study. Yet, at present this is not explicitly illustrated. Along these lines, owing to the large number of experiments and results reported here, I think the paper requires some considerable work in organization to streamline the article.*

Response: We thank the reviewer for the valuable suggestion. Accordingly, we have included a “Discussion” part and Scheme 1 by summarizing the foregoing conclusions to illustrate the whole Li-intercalation process precisely, as follows:

“Discussion. *Consequently, the electrochemical CD and CV measurements (Fig. 2a,b) initially suggest the CRC-stacking configuration and stoichiometric LiC_x compositions of the staged Li-intercalated bilayer-graphene phases. Then, the geometric and kinetics analyses (Fig. 2c–e) reveal a 2D model for Li storage in the interlayer of graphenic carbon. Furthermore, the subsequent combined ex- and in-situ characterizations roundly display the structure evolution information induced by Li intercalation by exploring variously related physicochemical phenomena, such as: (1) the slightly increasing interlayer space (Scheme 1a) by the XRD and SAED patterns, (2) the stable in-plane lattice spacings of graphene sheet by the experimental SAED patterns and computer modeling (Fig. 3c), and (3) the alternated configurations between A-B and A-A stacking modes dependent on Li density (Scheme 1b) by both the observed and simulated brightness ratios of the $\{0110\}$ and $\{1210\}$ diffraction spots (Fig. 3a,b,d) and the regular XPS phenomena assisted by the DFT calculations about the BEs of C and Li elements (Fig. 4). Therefore, all of these interrelated and well-matched phenomena and results, also including the Raman spectra, the impedance analysis, and the other electron-microscopy data despite the unavoidable irradiation*

influence of electron beam, can indicate the integrated Li intercalation process present in *Scheme 1* to be one reasonable mechanism to figure out the fundamental Li storage into graphenic carbon with little defects. In a word, the incoming Li^+ ions into the graphene interlayer under the electrodynamic forces tend to (diffuse along and) stabilize at ($\text{Li}^+ + e^- \rightarrow \text{Li}$) those overlapped sites with the most balanced electronic cloud density committed by the staged LiC_x units.” [Page 12–13]

Scheme 1 | Schematic of Li intercalation into bilayer graphene. **(a)** Evolution of the lamellar distance of bilayer graphene induced by the staged Li intercalation. **(b)** Evolution of the Li distribution in the interlayer of bilayer graphene and its stacking configuration decided by the staged Li densities (refer to Supplementary Fig. 9b). The inset green circles refer to the general sizes of the involved LiC_x units.

Besides, we have deleted some dispensable contents (summarized in another “**Deleted contents**” profile) to further streamline the manuscript, such as Fig. 1c inset, Fig. 2b, and Supplementary Figs 3e,g,i, 4, 5a,c, 6, 8a, 9a,c, 11, 15b–d, 17(ii), 18(top images), 21c, 22a(v–viii), 22b(iii,vii,xi), 22e(iii,vi), and 25a in the previous version.

Question 2: *I have some trouble understanding the structure of this porous graphenic material. In the schematic of Fig. 1a, it looks as though the graphene only covers the outermost layer of the nickel foam, yet I would expect that it would conformally coat all surfaces including the interior pore surfaces. Indeed, that is my interpretation of the description in the text. Perhaps the figure could be improved and clarified.*

Response: Thanks the reviewer for the comments. Only local area was presented in the initial Fig. 1a for illustration, and the graphene indeed conformally coats all surfaces including the interior pore surfaces. To illustrate the structure of this graphenic material well, Fig. 1a as well as the related Supplementary Fig. 2c has been modified accordingly, as shown below:

Figure 1 | Preparation and identification of bilayer graphene with 3D porous morphology. (a) Schematic of high-temperature CVD process by using 3D nanoporous nickel (np-Ni) substrate. Only local area of the ligament is presented for illustration.

Supplementary Fig. 2 | c, SEM images of the as-prepared graphene@np-Ni composite.

Question 3: *On that note, if the graphene does coat the interior surfaces, to what extent do the researchers observe the layers collapsing on each other to form multilayer regions (4,6,8... layers) after etching the Ni away? Is this effect prevented for some reason? Why is this not observed?*

Response: Thanks the reviewer for the comments. We employed the supercritical CO₂ drying technique to prepare the target sample, which can efficiently restrain the layers collapsing after etching Ni and help maintain the porous structure of the graphene@np-Ni composite. Related description has been included in the revised text, as follows: “which is ca. 30 μm thick after removing the Ni template and supercritical drying in case of the collapse of the 3D porous structure and thus the restacking of graphene sheets (Fig. 1b–e and Supplementary Fig. 3).” [Page 4, Line 65–67]

In addition, the below two images show the morphology of the 3D porous graphene material through supercritical drying or not.

Question 4: *I would like the authors to provide more details on how the ~90% bilayer coverage estimation is made.*

Response: Thanks the reviewer’s valuable suggestion. The related discussion have been added in the revised manuscript in virtue of the supplementary structure diagram of graphene foam in Fig. 1a, as follows (refer to the figure below): “through randomly collecting a large number of Raman spectra (with the laser spot diameter of > 5 μm, namely covering at least 4×10^8 C₆-ring units of 0.052 nm² by probe area) from different locations at the surface and cross section of the as-prepared samples (before and after removing the Ni substrate)”. [Page 5, Line 84–86]

Question 5: In *Supplementary Fig. 6*, what does "peak emerging order" mean? What is the significance of this?

Response: Thanks the reviewer for the comments. The "peak emerging order" in this figure means the observed order of appearance of the redox peaks as the CV sweep rate decreases from 100 to 0.2 mV s^{-1} . Only when the sweep rate is slow enough, one can observe some certain peaks from the CV curve. The figure indicates the relationship of the CV peaks and the sweep rate, namely their history and evolution process. Because the initial *Supplementary Fig. 5* has shown similar information and there is no strong correlation between this figure and the research theme, it has been deleted from the revised manuscript.

Question 6: It is not very clear what C1–C5 are as compared to C1 and CII.

Response: Thanks the reviewer for the comment. C1–C5 (*Fig. 2b* and *Supplementary Fig. 6a*) are the main cathodic peaks induced by the Li intercalation reactions ($\text{Li}^+ + \text{e}^- + \text{LiC}_x \rightarrow \text{LiC}_y$) under the electromotive force, when the "shoulder" peaks C1 and CII are deduced to correspond to the Li intercalation reactions associated with the transformation of the stacking configurations. Related discussion has been included in the revised text (**Page 6, Line 116–117**: "denoting the cathodic peaks as C1–5 and C1–II on behalf of the Li intercalation reactions of $\text{Li}^+ + \text{e}^- + \text{LiC}_x \rightarrow \text{LiC}_y$ ") and in the revised *Supplementary Figs 6b* and *9a*.

Question 7: *On page 6, line 122 I think perhaps "parallel" should be changed to "corresponding".*

Response: Thanks the reviewer's valuable correction.

Question 8: *In the figures showing the distribution of Li atoms/ions on the layer (Fig. 2d and Supplementary Fig. 12), why are some of the Li atoms placed at the vertices or edge-sharing regions between adjacent C₆ rings? Is it not accepted that Li atoms will bind in the middle of a C₆ ring, as in Fig. 3a(iv)?*

Response: Thanks the reviewer for the comments. In Fig. 2d as well as Supplementary Fig. 12 (corresponding to Fig. 2c and Supplementary Fig. 9 in the revised version, respectively), the whole C₆-ring (graphene) net is provided (and should be regarded) just as a reference substance/background (which has been included in the legend of the new Fig. 2c) to illustrate the distances or position relationships between Li atoms at the various LiC_x stages (on the X-Y plane, top view). Thus, this initial schematic doesn't consider the position change of Li atoms and hasn't involved the relative sliding of the two graphene sheets discussed in the latter Figs 3 and 4 in detail. The complementary Scheme 1 displays the more comprehensive phenomena of Li intercalation into bilayer graphene by including the stacking-configuration variation process after the SAED (Fig. 3) and XPS (Fig. 4) studies, namely from the A-B mode of pristine bilayer graphene or (quasi)-ARB or SP mode of low-Li-density LiC_x phases (where Li atoms placed at the vertices from the top view, refer to the insets of Fig. 4e,f) to the final ARA mode of the Li-saturated C₆LiC₆ (where Li atoms will bind in the middle of two C₆ rings, Fig. 3a(iv)).

In other words, a low density of Li atoms in the interlayer can't change the stable A-B stacking configuration of the pristine bilayer graphene, when the vertice (namely Site 1' in Fig. 4e,f) is the most stable position for a single Li atom; whereas, a high-enough density of Li atoms will cause the relative sliding of the two graphene sheets to yield their new stable stacking configuration(s) dependent on the system potential and Li density, when Li atoms will locate at new positions with the highest binding energies, such as Site 4' in the ARA mode or Site 3' (edge-sharing region) in

the transitional SP mode (if it can exist stably under the natural state). In any situation, from both the views of energy (chemical bond) and geometry, Li atom should locate at a site of high symmetry to keep the whole system stable and ordering (refer to Supplementary Fig. 9 of the revised manuscript).

Question 9: *Kuhne et al. reported Li diffusion coefficient D^δ up to $7E-5 \text{ cm}^2/\text{s}$ (ref. 40).*

Could the authors discuss why the Li diffusion here is slower ($1E-9$ to $1E-5$)?

Response: Thanks the reviewer's professional question.

After carefully reading this literature, we noticed the following points to explain the different results here, as follows:

(1) In *ref. 40*, the authors mentioned, "*The electrolyte, however, introduces additional disorder when applied on top of the ultrathin and sensitive layer. Through charged impurities and strain fluctuations, it exerts a detrimental influence on the properties of bilayer graphene (Page 895).*" Therefore, they excluded the electrolyte's effect deliberately to measure the Li diffusion, "*we have developed an on-chip electrochemical cell architecture in which the redox reaction that forces Li intercalation is localized only at a protrusion of the device so that the graphene bilayer remains unperturbed from the electrolyte during operation (Abstract).*" Thus, using the unique setup and employing the Hall measurements in a high vacuum with pressure $\leq 1 \times 10^{-6}$ mbar, their study was carried out in fact not in a real Li-ion-battery environment.

(2) The authors mentioned that, "*Experimental and theoretical studies on graphitic carbon suggest the general trend is D^δ decreases as n_{Li} (Li density) increases (Page 899)*", which, together with the D^δ values for bulk graphitic carbon of ca. " *10^{-12} – $10^{-5} \text{ cm}^2 \text{ s}^{-1}$* " (Page 895), is similar to our observation. Although their experimental findings also "*suggest a decrease in D^δ at the high-density end in Fig. 4*", however, the authors stated that they "*quickly face resolution limits on the quest to extract D^δ reliably at higher n_{Li} because of the dramatic drop of the Hall voltage*" and cannot "*ease this experimental challenge*" (Page 899). That is to say, although the authors applied an

enough low intercalation voltage (at 0.05 V vs. Li/Li⁺), the too short diffusion time at their experimental condition failed to achieve a high-Li-density phase (e.g., the P3 state in our study), suggesting that their as-obtained D^δ value possibly just corresponds to that for our P4 or even P5 phases.

(3) In addition, as shown in **Fig. 3** of *ref. 40*, the bilayer-graphene device looks like an “open” system (**refer to its (f) image shown below**) in comparison with that in a Li-ion cell. Even “*larger bilayer graphene flakes*” can overcome the above experimental challenge involved with the Hall voltage, as the authors mentioned, it may be difficult to guarantee that the whole bilayer-graphene (BLG) region is uniform in Li density in such an “open” system. In fact, just because of the uneven distribution of Li atoms, the authors observed “*a decrease in D^δ at the high-density end in Fig. 4*”.

Brief discussion has been included at *ref. 40* in the References part of the revised text, as follows: “*(The study measured the Li diffusion coefficient into bilayer graphene flake with locally varied Li densities by an “open” on-chip electrochemical cell to exclude the electrolyte’s influence.)*” [**Page 18, Line 373–375**]

Redacted

Question 10: *From Raman spectra data (Supplementary Fig. 15a) it looks as though the G and 2D peaks disappear upon intercalation due to Pauli blocking, yet in the intensity evolution plots of Supplementary Fig. 15d, this vanishing intensity is not exhibited. Could the authors clarify this?*

Response: Thanks the reviewer for the comments. In fact, both the position evolution plots in Supplementary Fig. 15c and the intensity-evolution plots in Supplementary Fig. 15d were obtained by individually processing each Raman spectrum by hand.

On the one hand, the deduction of the background for the Raman spectrum with weak bands assigned to graphene carbon (refer to Supplementary Fig. 15b), the collection of the intensity and position values in particular from the weaker target bands at low voltage ranges (refer to the below figure extractive from Supplementary Fig. 15a), and even the slope values estimated from the trend lines (Supplementary Fig. 15c,d), would accumulate undesired random errors to the final statics results.

On the other hand, from the below figure, one can see that although the *G* band became quite weak at the end, it didn't completely vanish, while the intensity of the *2D* band showed no remarkable change, which should be associated with their respective excitation mechanisms. For example, due to the final formation of the ARA configuration, the Li atoms embedded at Site 4' (refer to Fig. 3a(iv) and Fig. 4e,f insets) will restrain the doubly discrete E_{2g} mode at the Brillouin zone center (refer to the inset of the below figure), and thus the position and intensity of the related *G* band.

Because the *in-situ* Raman profile itself can clearly indicate the evolution trends of the band intensity and Raman shift, in the revised manuscript we have deleted Supplementary Fig. 15b–d only showing our statics data to avoid misleading others. Detailed Raman analysis can be found in the updated Supplementary Information (Supplementary Fig. 12, Page S19–S20).

Question 11: *Once the lithiated material is exposed to air, could the authors comment on the chemical and structural integrity/stability of the material that permitted them to acquire electron microscopy data (presumably ex situ)? Or was the sample never exposed to air?*

Response: Thanks the reviewer for the comments. As described in the part of **Materials and Methods** of Supplementary Information, namely “*Preparation and the ex-situ and in-situ characterizations of lithiated graphene electrodes*”, we prepared the related samples (sandwiched between two Cu grids) carefully for the *ex-situ* TEM measurements. Every time the TEM holder was carefully sealed in the glove box before it was rapidly transferred into the TEM apparatus (in 5 seconds). Thus, thanks to the covered Cu grids, the rapid transfer, and even the protection of the SEI film on the electrode surface, the lithiated graphene samples can well avoid the influence of air/O₂, as reflected by all of the electron microscopy data.

Furthermore, the *ex-situ* XPS results (Fig. 4 and Supplementary Fig. 21 in the present manuscript) can also confirm the reliability of our operation to prevent the oxidation in air during the sample transfer. In addition, from the SEM images of the samples after the XPS testing and being exposed in air for a long time (new Supplementary Fig. 13a), they all kept the well-defined 3D nanoporous morphology, indicating the high structure stability. However, their Raman spectra (new Supplementary Fig. 13b) were different from the *in-situ* case by exhibiting enhanced *D* bands due to the oxidation in air. The XRD results (new Supplementary Fig. 16) can also reflect the influence of oxidation, indicating the chemical sensibility of the lithiated bilayer-graphene material and thus the necessity of the above protection.

Reviewer #2: *This communication presents a study of Li intercalation in bilayer carbon foam produced by chemical vapor deposition on a particular type of 3D Ni foam. The authors propose this system to address the question: Can ‘lithiation’ happen on single layer graphene or only in between graphene layers? The work has some significant deficiencies. Some suggestions are*

made about how it might be improved.

Question 1: *(General - information) Only relevant information should be presented, it also includes the supplementary information.*

Response: We thank the reviewer for good advice. Accordingly, we have deleted the some dispensable contents (summarized in another “**Deleted contents**” profile) to streamline the manuscript, such as Fig. 1c inset, Fig. 2b, and Supplementary Figs 3e,g,i, 4, 5a,c, 6, 8a, 9a,c, 11, 15b–d, 17(ii), 18(top images), 21c, 22a(v–viii), 22b(iii,vii,xi), 22e(iii,vi), and 25a in the previous version.

Question 2: *(Language problems) Language such as (Line 21) “The real capacity of ideal graphene...” or (Line 30) “The benign development of graphene ...” should be avoided (that is: are the authors wanting to draw attention to the fact that there might be nefarious uses for graphene?).*

Authors also can only claim they prove something if they present overwhelming evidence (Line 24 “we experimentally prove that ...”). (Instead, should be something like: “Experimental data show...”)

The authors should not assume that something is “correct” (Line 57), or “a major confusion” (Line 61), or that something is “impossible” (Line 83), based on their personal opinions. Similar language issues such as Line 54 “Logically speaking” (why would anyone speak illogically?) offer further challenges to the reader who is trying to discern what the science that has been done, is.

Again: (Line 177) “Despite the too weak (002) peaks ... clearly reveal that ...”; (Line 193) “Despite this interference ...which vividly display”. How can too weak peaks clearly reveal anything? How can such a level of interference lead to something, nonetheless, being vividly displayed?

Finally, there is the issue that the authors do not have graphene (as it is typically defined) but instead something that one might refer to as

“graphenic carbon” or “sp²-bonded carbon layers in a 3D structure.” But the reviewer acknowledges that there is a terminology problem in this field.

It is suggested that the authors have the manuscript edited by a professional editor.

Response: We thank the reviewer very much for pointing out the language problems. We have checked the writing of the manuscript carefully and made a thorough revision via the English language editing service from the Editage company (**Job Code: HQZWB_1**).

Question 3: *(Line 102) The evidence that authors provide for no Li⁺ adsorption on the outer plane surface of what they refer to as the ‘graphene’ is that the charge-discharge capacities are close to ~180 mAh/g (as well as ‘small’ capacities in some other few-layer graphene tests), which is about half of LiC₆, and is shown as Fig. 2a. Please provide more detail about such measurements. For example: show the difference of 1st and later cycles of charge-discharge profiles, and report the Coulombic efficiencies and the irreversible capacities. Capacity loss from the SEI formation and distribution in such ‘graphene materials’ could be more serious than in graphite; one reason that high specific surface area carbons have not been effective in Li battery anodes is because of irreversible loss of too much Li in the SEI layer(s) that build up. It has been said that the basal planes of graphitic domains are coated by stable organic-rich SEI (comparing to inorganic-rich SEI regions on edges), and thus the SEI issue needs to be addressed. Provide references that state that it is a non-issue in your particular carbon studied here, or investigate and report about it.*

Response: Thanks for the reviewer’s specialized and valuable comments. The supplementary information below has been added in the revised manuscript (Supplementary Fig. 4d–e), from which one can see the stable enough performance of the 3D nanoporous bilayer-graphene material during our study and the exemption of

the serious influence from the surface SEI generated at above 0.5 V during the initial electrochemical cycles (*refs 27,29–31* in the main text).

Supplementary Fig. 4 | b,c, CD profiles (1–30 circles) of one used bilayer-graphene electrode (after the CV test) at 1.0 A g^{-1} between 0.5 and 0.02 V. **d,e**, CD profiles (1–70 circles) of one fresh bilayer-graphene electrode at 0.1 A g^{-1} between 0.5 and 0.005 V. These results verified both the stable enough performance of the 3D nanoporous bilayer-graphene material during a long period of testing and the exemption of the serious influence from the surface SEI generated at above 0.5 V during the initial electrochemical cycles, guaranteeing the reliability of our experimental results.

Question 4: (Line 107) The authors should consider all possible options before choosing one as definitive: a configuration $\text{LiC}_{12}\text{C}_{12}\text{Li}$ (among others) has the same theoretical capacity as C_6LiC_6 .

Response: Thanks the reviewer for the comments. Here the C_6LiC_6 (or Li_3C_{12}) just refers to the simplest molecular formula to indicate the stoichiometric ratio of Li and C atoms based on the theoretical capacity of 186 mAh g^{-1} (see Supplementary Fig. 1a) and the minimum C_6 -ring unit (refer to Supplementary Fig. 9b of the present manuscript). To make the expression unambiguous, the initial sentence has been

modified as (**Page 6, Line 108**), “its maximum GCD capacity only reached $\sim 180 \text{ mAh g}^{-1}$ at $50\text{--}0.2 \text{ A g}^{-1}$ (**Fig. 2a**), suggesting that the interlayer should be the sole space for bilayer graphene to store Li atoms to generate stoichiometric LiC_{12} rather than Li_3C_{12} of 558 mAh g^{-1} (**Supplementary Fig. 1a**).” By the way, Kuhne et al. also proved the interlayer to be the sole Li-diffusion pathway in **ref. 40**.

Question 5: (Line 111) Provide references here.

Response: Thanks the reviewer’s correction.

Question 6: (Line 130) $\text{C}_{122}\text{LiC}_{122} \rightleftharpoons \text{C}_{42.6}\text{LiC}_{42.6} \rightleftharpoons \text{C}_{14}\text{LiC}_{14} \dots$

*These stoichiometric numbers, or more accurately, as stated in **Fig. S12** caption, $122 - 42.7 - 12.7(\text{alternative}) - 6$ (little bit off)*

*Can such numbers lead to a series of planar fractal triangles, if the statistics are not sufficient? It would seem (**Fig. S9d**) that the numbers are actually: $(186\sim 270)/2 - (64\sim 103)/2 - (28\sim 32)/2 - (12)/2$, not to mention the variation in the numbers among cells/CV cycle #. What would the fractal series of ‘stoichiometric numbers’ be if starting from LiC_6 and how far off are these numbers from the experimental data? Why do the authors particularly report such accurate numbers ($122 - 42.7 - 12.7 - 6$) from their (it would seem, much ‘coarser’) experimental data, and confirm a fractal model even though the last 2 numbers are “off”? The authors state that the P3’ distribution “may be more exact to describe to describe the as-claimed Li-graphite intercalation product of LiC_{12} ”. Please give references or prove that the P3’ distribution is a thermodynamically most stable gallery arrangement of LiC_{12} , which has some Li^+ eclipsing the C atoms in the AA stacking configuration.*

Response: Thanks for the reviewer’s valuable comments. To better answer these questions, we have plotted all the planar images for the related LiC_x compositions, as shown below (corresponding to **Supplementary Fig. 9a** in the present version). From

this figure, one can know that the positions of Li atoms at each phase, calculated based on these x values, are very close and regular/organized, which enable us to speculate the possibly approximate location of each-phase Li atoms and further apply the mean x values after a little reasonable adjustment (namely $122 - 42.7 - 14 - 6$, Supplementary Fig. 9b) to obtain the universal position relationship of the staged Li atoms (Supplementary Fig. 9c,d and Scheme 1b in the revised manuscript).

Supplementary Fig. 9 | **a**, Ideally areal distribution of Li atoms on the X-Y plane according to the estimated Li-intercalation/deintercalation process shown in Supplementary Fig. 7d, namely the range and the mean of the seven x values for each LiC_x phase. **b**, Distribution of Li atoms on the graphene net according to the average LiC_x phases (see Scheme 1b), namely $\text{C} \rightleftharpoons \text{LiC}_{122} \rightleftharpoons \text{LiC}_{42.7} \rightleftharpoons \text{LiC}_{14} \rightleftharpoons \text{LiC}_6$. **c,d**, Geometrical position relationships of Li atoms at the staged Li-graphene phases based on **(b)**. **e**, Theoretical planar image of LiC_6 plotted by following the fractal Li-intercalation way. The balls of alternating color refer to the intercalated Li atoms at each step. (See the detail in the revised Supplementary Information)

In particular, from the P3 image shown in Supplementary Fig. 9a, one can see the structure of $C_{12}LiC_{12}$ in fact does not exist, whose unit cell can't be constructed due to the lack of periodicity. That is to say, its Li atoms will be always off from the geometrical center of the graphene carbons (imagine the large-range distribution and refer to Fig. 4), which represents an instability state of energy from the chemical-bond view. The conjoint $C_{12.7}LiC_{12.7}$ and $C_{10.8}LiC_{10.8}$ compositions are reasonable in theory. Therefore, if the previous estimation of LiC_{12} was relatively accurate for graphite, according to the above rule, this Li-intercalation composition (P3') may be revised as $LiC_{12.7}$. In addition, the P3 image indicates that, no matter what the P3 compositions are (with the x value at 12.7–16), the final intercalation of Li atoms to form C_6LiC_6 will always cause the previous Li atoms to move, in line with the kinetics results shown in Fig. 2d,e and the presence of the P2 phase (with a possible composition of $C_{8.67}LiC_{8.67}$, refer to the black circle in the P3 image of Supplementary Fig. 9a and to the relatively smaller CI shoulder peaks). Here the static model (Fig. 2c and Supplementary Fig. 9c) only figures out the intercalated result but cannot display the dynamic Li-intercalation phenomena or process.

Furthermore, according to the reviewer's suggestion, we have also obtained the "backward" fractal series starting from LiC_6 (see the figure below and Supplementary Fig. 9e), which suggest a process of $LiC_6 \rightarrow LiC_{18} \rightarrow LiC_{54} \rightarrow LiC_{162} \rightarrow C$ in terms of the pure math. In this ideal mode, all the intermediate phases (**P5**, **P4**, and **P3**) possess looser Li distribution densities than those experimental ones (e.g., $C \rightleftharpoons LiC_{122} \rightleftharpoons LiC_{42.7} \rightleftharpoons LiC_{14} \rightleftharpoons LiC_6$), which may not lead to increasing Li^+ diffusion resistances even for the final formation of C_6LiC_6 because of the always large enough space between two neighboring Li atoms.

Question 7: (Line 177) The authors state that the gallery expansion from G-G to G-Li-G is from 0.335 nm to 0.336 nm (P_2 , LiC_{12-28}), according to their XRD result. How is it possible to insert Li^+ in 0.001 nm? (The gallery height of the graphite intercalation compounds (GIC) LiC_x is 0.370 nm).

Response: We are sorry to make the reviewer misunderstand here. Here we intend to state that, as the Li density increases in the 2D interlayer of bilayer graphene, the d_{002} value (or the average gallery height; refer to the below supplementary Scheme 1a in the revised manuscript) will become larger and larger due to the enhanced interaction of the middle Li layer and the bilateral graphene layers.

Scheme 1 | (a) Evolution of the lamellar distances induced by staged Li intercalation.

Related discussion after modification can be seen at **Page 9** of the present manuscript, as follows: “For that the (002) peaks of the X-ray diffraction (XRD)

patterns for bilayer graphene and its lithiated products were too weak, one multilayer sample was employed to study the analogous interplanar distance (i.e., the d_{002} value, ca. 3.355 Å for graphite^{28,42} and 3.369 Å for it; Supplementary Fig. 16). The results showed that, in the wake of Li intercalation this value decreased firstly and then increased gradually from ca. 3.351 Å at P4 to 3.364 Å at P2. In addition to being consistent with the literature^{3,4,29,38,42} and the previous DFT simulations for both graphenes¹⁰ and graphite⁴⁵, the latter trend was also displayed by the following SAED results for Li-bilayered graphene samples (from ca. 3.42 Å at P4 to 3.50 Å at P1; Supplementary Fig. 17a,d).” [Page 9, Line 177–185]

Reviewers' comments:

Reviewer #1 (Remarks to the Author):

The authors have addressed all my concerns in a satisfactory manner.

In my opinion the manuscript should be accepted for publication, though I strongly recommend some additional efforts to streamline/organize the text and edit the manuscript for grammar and language issues. Doing so will make the paper significantly more readable. But in terms of the science, I find the work is sound and will be a valuable contribution to the community.

Reviewer #2 (Remarks to the Author):

Review

This paper presents a study of Li intercalation in bilayer carbon foam produced by chemical vapor deposition on a particular type of 3D Ni foam. The authors propose this system to elucidate the mechanism of Li storage in bilayer graphene foam.

Although the newer version of this work is improved, there are still a number of questions/queries to be addressed:

Q: (Supplementary Information page 7 and supplementary Figure 2).

The authors should explain the importance of rapid cooling during the growth process and specify the cooling rate.

Q: (Line 79-94). The authors mention that 90% of their sample exhibits Raman spectra (figure 1-e) similar to graphene bilayer and bilayer on Ni. Thus, a question is whether 90% of your sample is flat. The corresponding Raman modes of curved single- and bi-layer graphene should shift towards lower wavenumbers relative to those of unperturbed (flat) graphene films as reported in doi.org/10.1021/nl203359n. The Raman spectra in figure 1-e do not seem to be typical spectra of a 3D porous graphene foam. A question then arises: Could the effect of the curvature be similar to that of multilayer graphene and could it entail the emergence of the Raman spectrum similar to that of unperturbed bilayer graphene?

Q: (Figure 1-d). The authors present the SAED pattern of a flat region and so: Is this region representative of a 3D porous graphene foam? The curvature of the carbon foam should affect the intensity ratios in the SAED patterns as has been reported in the cited reference 19

Q: (Line 75-79) The authors mention that the ratio of the (0110) peak intensity to the (1210) peak intensity was measured to be about 0.49, which is in a good agreement with the computational and experimental values reported. The authors should provide the relevant references. Besides, the effect of the curvature on the SAED patterns should be taken into account. It is also unclear how the intensity ratios are calculated.

(Line 186) In figures 3a and 3b the intensity distributions over the 1st and 2nd order diffraction peaks do not correlate with the values given for the corresponding intensity ratios. – Why does Fig. 3a show a higher intensity for (0-110) and (1-100) peaks and a lower intensity for the (-1010) peak?

Q: Figure 3(iv) shows the model of an expanded C₆-ring with Li located at the center of the ring in the

graphene basal plane. Yet, the authors fail to provide direct evidence to prove this statement. The evolution of the SAED pattern suggests only lattice expansion due to the electron beam irradiation. Doesn't it contradict the conclusion by the authors that Li can only be stored between layers?

Q: (Line 177) The response given by the authors is not convincing.

Since upon lithiation the interlayer distance increases by 0.001-0.002 nm from 0.335 nm for G-G to 0.336-0.337 nm for G-Li-G then, considering the Li⁺ diameter, it is unclear how Li can intercalate between the layers and fit the aforementioned spacing. Besides, if some of the Li⁺ in the planar fractal model were placed at the vertices of C₆ rings, then the gallery expansion would be even bigger than putting Li⁺ only at the centers of C₆ rings, which is typically meaning a 0.370 nm layer spacing in LiC₆.

Q: (Response Letter to Referee #2 Q4) The authors aimed at answering the question whether lithiation occurs between graphene layers only. Yet, the intercalation of Li between layers has been reported already by Kuhne et al (40). Thus, the novelty of the current work is unclear.

Reviewer #1: *The authors have addressed all my concerns in a satisfactory manner.*

In my opinion, the manuscript should be accepted for publication, though I strongly recommend some additional efforts to streamline/organize the text and edit the manuscript for grammar and language issues. Doing so will make the paper significantly more readable. But in terms of the science, I find the work is sound and will be a valuable contribution to the community.

Response: We greatly appreciate the reviewer's positive comments and valuable suggestion to help us further improve the manuscript. A thorough revision has been made to address the language issues, with the help in English language editing from Editage (**Job Code: HQZWB_1_3**).

Reviewer #2: *This paper presents a study of Li intercalation in bilayer carbon foam produced by chemical vapor deposition on a particular type of 3D Ni foam. The authors propose this system to elucidate the mechanism of Li storage in bilayer graphene foam. Although the newer version of this work is improved, there are still a number of questions/queries to be addressed:*

Q1: *(Supplementary Information page 7 and Supplementary Fig. 2). The authors should explain the importance of rapid cooling during the growth process and specify the cooling rate.*

Response: We thank the reviewer for this valuable suggestion. Previously, Yu et al. found that the cooling rate is critical to produce thin graphene films with fewer than 10 layers on smooth Ni foil by non-equilibrium surface segregation.^[Note R1] At their “**extremely fast cooling rate**” of 20 °C s⁻¹ (similar to our case), the solute atoms lose their mobility due to a quenching effect. Using computer modeling, Al-Shurman and Naseem [“CVD graphene growth mechanism of on Ni thin films”, in *The Proceedings of the 2014 COMSOL Conference in Boston*, **Figures 11** and **12**] showed the formation of bilayer graphene on Ni film cooling from 1000 to 725 °C.

To address your concern, a complementary diagram for the growth mechanism of bilayer graphene has been added to our revised manuscript (namely Supplementary Fig. 2c, below), and the relevant discussions are as follows:

*“On the one hand, such an operation can efficiently limit the carbon uptake and release to enable bilayer graphenic carbon to initially generate on Ni at T_1 ^{S7} rather than at the moment of rapid cooling as before^{S9,S11} (**b**, **c**). On the other hand, thanks to*

the resistance from the improving graphene crystal with less defects and the decreased diffusivity of C element in the fast-cooling Ni (in 15–20 s and ca. 1 min from the displayed T_i to 400 and then 60 °C, **b**), it would also restrain the precipitation of excess C atoms at the Ni-graphene interface to yield high-quality bilayer graphene^{S8,S11} (c).”

Supplementary Fig. 2c (newly added)

Q2: (Line 79-94). The authors mention that 90% of their sample exhibits Raman spectra (Fig. 1e) similar to graphene bilayer and bilayer on Ni. Thus, a question is whether 90% of your sample is flat.

The corresponding Raman modes of curved single-and bi-layer graphene should shift towards lower wavenumbers relative to those of unperturbed (flat) graphene films as reported in doi.org/10.1021/nl203359n. The Raman spectra in Fig. 1e do not seem to be typical spectra of a 3D porous graphene foam. A question then arises: Could the effect of the curvature be similar to that of multilayer graphene and could it entail the emergence of the Raman spectrum similar to that of unperturbed bilayer graphene?

Response: We thank the reviewer for sharing the technical insights.

About the first question: Because the detection range of our Raman technique, the laser spot diameter ($> 5 \mu\text{m}$, Fig. 1a) is larger than the typical pore diameters in the graphene foam (ca. $500 \text{ nm} - 1 \mu\text{m}$, Fig. 1b,c and Line 73), the Raman spectra (Fig. 1e) could not be obtained for a given flat region, unlike the TEM-SAED technique (refer to the diagram below). That is to say, the 3D region covered by the laser spot is always consisted of graphenic carbons with various numbers of layers (if any), spatial orientations, and/or geometric shapes/curvatures. Hence, although the statistical Raman data showed that $\sim 90\%$ region of the sample was constructed by bilayer graphene (refer to **Referee #1: Q4** in our previous Response Letter), this does not mean that 90% of the sample is actually flat. Or rather, the graphene sheets in the graphene foam are not flat at the micron level (e.g., in the Raman characterization), but it can be regarded as flat in the nanoscale (e.g., in the SAED characterization) especially when compared with the tiny Li atom and the C_6 -ring unit.

About the second question: In the paper you indicated (Zabel et al.,

doi.org/10.1021/nl203359n), the Raman spectra (shown below) in **Figure 3a** for single-layer and **Figure 4a** for bilayer graphene bubbles (average diameter: 5–10 μm) were recorded using the 488-nm laser line with a laser spot size of only 400 nm from two different positions, namely at the bubble center and on the flat area on the Si/SiO_x substrate, respectively. Accordingly, we would like to point out three subtle issues:

(1) The authors mentioned that graphene carbons at the two locations indeed possessed different chemical compositions, even if they believed it was not the substrate doping effect to cause the peak position shift (*without proving any other direct reason*).

(2) The much larger bubble diameter and the very small laser spot size means their experimental condition (which yielded local information in 2D space) was almost the opposite of our Raman test (500 nm to 1 μm for the pore diameter of 3D graphene foam and $> 5 \mu\text{m}$ in laser spot size, which yields statistical information). Meanwhile, the particular N₂ atmosphere and differential pressure in that study were quite different from ours (refer to their influence in **Figure 4b** shown below).

(3) The Raman spectra recorded by Zabel et al. at **488 nm** seemed not to exhibit the well-known features for monolayer and bilayer graphene (in terms of the intensity ratio and relative height of the *G* and *2D* bands), but their bilayered spectra at **514.5 nm** did (see their **Figure 4b**, the same excitation laser wavelength as in our case).

In summary, the above analysis and the limited information provided by Zabel et al. suggest it is difficult to conclude that the graphene morphology may affect the Raman shift of its *G* and *2D* bands. In comparison, the laser line (*ref. 23*) and the differential pressure (**Figure 4b** by Zabel et al., shown below) are more likely reasons.

The many similar Raman spectra reported for graphene foam (e.g., *ref. 17*, *Nature Mater.* 10 (2011) 424, and *Nanoscale* 8 (2016) 13303) and planar graphenes (e.g., *refs 24,25*) further support our characterization results. We consider the fundamental cause of the above phenomena should lie in the origin of the Raman bands, e.g., the *G* band corresponds to the E_{2g} phonon at the Brillouin zone center (Zabel et al.; refer to the figure for **Referee #1: Q10** in our previous Response Letter). Relevant discussion has been included in the revised manuscript (**Line 81**), “*Raman spectroscopy can provide more definitive identification for graphene materials with diverse numbers of layers regardless of the macroscopic appearance*^{17,21,23–25}”.

Figure 3. (a)
at 488 nm, single-layer

Figure 4. (a)
at 488 nm, bilayer

Figure 4. (b)
at 514.5 nm, bilayer

Redacted

(Figures from Zabel et al., doi.org/10.1021/nl203359n)

Q3: (*Fig. 1d*). The authors present the SAED pattern of a flat region and so: Is this region representative of a 3D porous graphene foam? The curvature of the carbon foam should affect the intensity ratios in the SAED patterns as has been reported in the cited *ref. 19*.

Response: We thank the reviewer for this question. As we mentioned in the response to **Q2** (and the first figure therein), a piece of 3D porous graphenic carbon foam can

possess graphene regions with various shapes or morphogenesis, such as the sufficiently flat region in Supplementary Fig. 3d (for the pattern of Fig. 1d) and the curly regions with varied curvature in Supplementary Fig. 3e. **Thus, the region flat at the nanoscale is just one possible structure in the graphene foam.**

Nevertheless, such flat regions were abundant in our sample (refer to the diagrams **a** and **b** below). Due to the obvious differences in SAED patterns from the flat and curved regions (Supplementary Fig. 3d,e insets, also see *refs 19,43*), it is also feasible to determine whether the as-obtained normal-incidence pattern comes from a flat enough region for further analysis, as discussed at Supplementary Fig. 17a. Besides, this SAED technique, as well as the Raman spectroscopy and the electrochemical capacity (Supplementary Fig. 1a), can effectively identify bilayer/monolayer graphenes with unique and distinctive features but not the few/multi-layered ones currently. Therefore, we could ensure the dominance of bilayer structure in the as-fabricated graphene foam at different size scales (from nm to cm). **In this sense, the flat region at the nanoscale is representative of the bilayer graphene foam.**

Q4: (Line 75-79) The authors mention that the ratio of the $\{0110\}$ peak intensity to the

{1210} peak intensity was measured to be about 0.49, which is in a good agreement with the computational and experimental values reported. The authors should provide the relevant references. Besides, the effect of the curvature on the SAED patterns should be taken into account. It is also unclear how the intensity ratios are calculated.

(Line 186) In Figs 3a and 3b the intensity distributions over the 1st and 2nd order diffraction peaks do not correlate with the values given for the corresponding intensity ratios. Why does Fig. 3a show a higher intensity for (0-110) and (1-100) peaks and a lower intensity for the (-1010) peak?

Response: We thank the reviewer for the valuable suggestion. The references are (20,21) in Line 79, and the peak intensities (measured using the DigitalMicrograph software) together with the calculation method for the $I_{\{0110\}}/I_{\{1210\}}$ ratio have been included in Supplementary Fig. 3d (refer to the figure below).

$$\begin{aligned}
 & I_{\{0110\}}/I_{\{1210\}} \\
 &= (40 + 31 + 12 + 39 + 20 + 18)/ \\
 & \quad (17 + 121 + 9 + 10 + 155 + 16) \\
 &\approx \mathbf{0.488}
 \end{aligned}$$

Supplementary Fig. 3d (revised inset)

Of the effect of the curvature on the SAED patterns, on the one hand, as discussed in Q3, all the SAED patterns used for subsequent analysis were from flat enough regions, which were similar with the cases in ref. 19 and ref. 43 (by the same research

group) when the incidence angle between the electron beam and graphene sheet is 0° (**Figure 2g** in *ref. 19*). On the other hand, local curvature was not and cannot be completely avoided in *refs 19, 43*^[Note R2], neither could it be eliminated in this work. As a result, non-identical intensities of individual SAED peaks were observed experimentally (refer to our theoretical results in **Fig. 3d**). In addition, although the authors of *ref. 43* explained the influence of curvature on individual diffraction peaks in detail (**Figure 3** in *ref. 43*), they did not specify its effect on the total $I_{\{0110\}}/I_{\{1210\}}$ ratio. In view of the varied values of this ratio from different reports (e.g., < 1 in *ref. 19*, 0.4 in *ref. 20*, 0.5 in *ref. 21*, and 0.49 here) and our simulated value of 0.44, we think that the local curvature will not change the $I_{\{0110\}}/I_{\{1210\}}$ ratio at 0° tilt angle (0.4–0.5) too much, and thus it would not influence our final conclusion. Related discussion has also been included at Supplementary Fig. 3 [Page 8].

As of the discrepancy shown in **Fig. 3a,b(i)**, the previous version of our manuscript gave a brief explanation in **Line 200–206**. For clarity, detailed discussion about this important phenomenon has been added in the revised version [**Line 211–221**], as follows: “*For example, in the case of Fig. 3a(i), there should be a very small amount of Li atoms entering the bottom graphene sheet to yield some Li-inlaid bilayer graphene regions (refer to Fig. 3b(iv)). Due to the different properties between the Li-inlaid graphene sheet and the pristine one, such local configurations would allow some planes to exhibit monolayered SAED feature (Fig. 3a(i) inset), but most regions of A-B stacking will not be disturbed (when $I_{\{0110\}}/I_{\{1210\}} = 0.53$). Once all the in-between Li atoms enter the bottom sheet (Fig. 3b(i)), the two distinctive sheets would display their respective*

monolayered features (when $I_{\{0110\}}/I_{\{1210\}} > 1$). However, there are still some Li-free bilayered regions (Fig. 3b(i) inset) due the limited population of Li atoms at this phase (refer to the case in Fig. 3b(iii)). Thus, both the previously reported local curvature effect^{19,43} and this Li-inlay effect would lead to structural inhomogeneity to affect the diffraction intensities of various planes.”

Finally, it needs to be emphasized that, even after a short duration of irradiation, the SAED patterns in Fig. 3a(i-iii) still deviated from the intrinsic ones slightly. Refer to the simulated $I_{\{0110\}}/I_{\{1210\}}$ values in Fig. 3d and Supplementary Table 1 (e.g., 0.40–0.44 for P4 and 1.17 for PI), the contradiction in Fig. 3a(i), and the enlarged lattice spacings in Fig. 3a(ii) accompanied by Fig. 3a(iv). Therefore, both the Li-inlay effect and the local curvature effect discussed above may cause the varied intensities of the (0-110), (1-100), and (-1010) peaks in Fig. 3a(iii). Related discussion has been included in the revised text [Line 195–197]. Of course, other factors yet-to-be-revealed in the literature may also contribute to this complicated phenomenon.

Q5: *Fig. 3b(iv) shows the model of an expanded C_6 -ring with Li located at the center of the ring in the graphene basal plane. Yet, the authors fail to provide direct evidence to prove this statement. The evolution of the SAED pattern suggests only lattice expansion due to the electron beam irradiation. Doesn't it contradict the conclusion by the authors that Li can only be stored between layers?*

Response: We thank the reviewer for the valuable comments. On the one hand, because the model was established mainly through DFT-based computation and a trial-and-error

process, we can only obtain the final relaxed structure (with the minimum system energy in theory) that accounts for the electron beam-disturbed experimental results for P3 (Fig. 3b(ii) and Supplementary Fig. 17b) as well as P1 (refer to the discussion at Supplementary Fig. 18). The detailed implementation is described in the revised **Supplementary Information (Page 4: DFT calculations)**, as follows:

“In the static calculations of total energy, a $10 \times 10 \times 1$ Monkhorst-Pack k grid was employed and the electron wavefunctions were expanded by using a plane wave basis set with a cutoff energy of 400.0 eV.

On the one hand, to confirm the lattice-expansion phenomena of lithiated bilayer graphene under electron beam irradiation (Fig. 3c and Supplementary Figs 17b and 18), the Li-inlaid unit cells of LiC_{14} and LiC_6 with normal lattice parameters (as well as many other configuration models by adjusting the spatial positions of Li atom) were constructed in the first step. Then, we restricted the Z-axis relaxation and merely allowed atoms to relax in X-Y plane (namely to achieve the smallest system energy) to obtain the optimized structure achievable in theory. Finally, the above CaRine Crystallography 3.1 software was applied to simulate the lattice parameters of the new structure used for comparison with those experimental values by SAED.”

On the other hand, as discussed above, such a configuration is in fact a result of certain physical processes (namely electron bombardment). However, our conclusion that Li can only be stored between graphene interlayer refers to electrochemical reactions in the Li-ion battery with a limited working voltage. Given the different contexts, there is no contradiction between the two. Moreover, making it clear the

extrinsic Li-graphene configuration can help prove the intrinsic one under the natural state. This point has been further stressed at Supplementary Fig. 17d [Page 31] and in the revised manuscript [**Line 198–200**], “*these comparative but regular extrinsic patterns induced by beam irradiation can further illustrate the intrinsic CRC configuration of the generated Li-bilayer graphene phases in the LIB (Supplementary Figs 18 and 19)*”.

Q6: *(Line 177) The response given by the authors is not convincing. Since upon lithiation the interlayer distance increases by 0.001-0.002 nm from 0.335 nm for G-G to 0.336-0.337 nm for G-Li-G then, considering the Li⁺ diameter, it is unclear how Li can intercalate between the layers and fit the aforementioned spacing. Besides, if some of the Li⁺ in the planar fractal model were placed at the vertices of C₆ rings, then the gallery expansion would be even bigger than putting Li⁺ only at the centers of C₆ rings, which is typically meaning a 0.370 nm layer spacing in LiC₆.*

Response: We thank the reviewer very much for the further discussion.

On the one hand, firstly, being smaller than the interplanar spacing (ca. $c_0 = 3.35$ Å), initial Li⁺ ions (empirical diameter: 1.52 Å) apparently enable the free diffusion between the graphene interlayer (ref. 40^{[Note R3]}), until the resultant electrodynamic force becomes too small at the PI stage of C₆LiC₆ (Scheme 1b). Secondly, for the Li atoms (empirical diameter: 3.04 Å) stabilizing at the graphene interlayer (by Li⁺ + e⁻ → Li), according to the studies in refs. 11, 14 and 44 and our XPS results (Fig. 4d), in

fact they are always ionic due to the charge redistribution (*ref.* 7^[Note R4]) or interaction between it and its nearest neighbor C atoms (analogous to ordinary ionic compound), ca. $+0.9|e|$ for Li at the center of C_6 rings (see the structure in Fig. 3a(iv)). As for the case that Li atom locates at the vertice of a C_6 ring (still at the center of another C_6 ring, refer to the insets of Fig. 4e and 4f for Case 1), this situation only occurs at the initial phases with low Li concentration, like P5 and P4 with ARB stacking configuration (Scheme 1b). Thus, the quite limited number of C_6 -Li units would not greatly influence the average layer spacing determined by the pristine graphene sheets, in particular for bilayer graphene to enable only one Li layer without the interference of adjacent layers.

On the other hand, *ref.* 7 (“*Intercalation compounds of graphite*”, *Adv. Phys.* 51 (2002) 1–186; a review article originally published in 1981 by M. S. Dresselhaus and G. Dresselhaus), gave two structural models (proposed 40–50 years ago) and detailed theoretical explanations (refer to its Table 10^[Note R4]) to describe the geometrical relations for first stage LiC_6 of graphite (pristine interplanar spacing $c_0 = 3.35 \text{ \AA}$). One was a **hard sphere** model using the radii of C (0.71 \AA) and Li atoms (Fig. 28a in *ref.* 7, namely Figure (a) below), and the other was a “**billiard ball**” model using Li^+ radius^[Note R4]. Considering the essential structure differences between thick graphite and graphenes (especially bilayer graphene), together with the inevitable model error (e.g., hard sphere \neq microscopic particles at the atomic level), instrument error/precision, and technique difference, the unequal interplanar lattice spacings, such as 0.372 nm for LiC_6 of graphite^[Note R4] (possibly decided by the Li layers with stronger interactions rather than the C layers, refer to Figure (b) below), 0.336 nm for P2 of multilayer graphene

(by XRD in Supplementary Fig. 16), and 0.350 nm for P1 of bilayer graphene (by SAED in Supplementary Fig. 17d), may be understandable, too. Despite the different specific d_{002} values, the lattice change trends induced by Li intercalation are consistent, in keeping with the varied charge distribution states of the C and Li layers as well.

Consequently, to better address this concern, we have added the structure model for the C_6 -Li unit (refer to **Figure (c)** below) in the revised Scheme 1, and related discussion can be seen in **Line 166, 183 and 257** of the updated manuscript, respectively.

Q7: (Response Letter to *Referee #2 Q4*) The authors aimed at answering the question whether lithiation occurs between graphene layers only. Yet, the intercalation of

Li between layers has been reported already by Kuhne et al (40). Thus, the novelty of the current work is unclear.

Response: We thank the reviewer for the comment. In *ref. 40*, Kuhne et al. observed the real-time Li-diffusion **process** (non-statistical information)^[Note R3], while our conclusion here is based on the Li-diffusion **outcome** (statistical information), namely the realized specific capacity of 186 mAh g⁻¹ for bilayer graphene. Related comment has been included in the previous annotation for *ref. 40*, “Using an “open” on-chip electrochemical cell to exclude the electrolyte’s influence, the study measured the Li diffusion coefficient into bilayer graphene flake with locally varying Li densities, and provided real-time evidence that the Li intercalation/diffusion only occurs at the graphene interlayer”. Furthermore, *ref. 40* only reported Li-diffusion rate in bilayer graphene interlayer, while the current work covers much more details to clarify the whole Li storage phenomena and mechanism in graphenic carbon.

Notes

[R1] Yu et al., Applied Physics Letters 93 (2008) 113103: “...different cooling rates led to different segregation behaviors, strongly affecting the thickness and quality of the graphene films”, and “extremely fast cooling rate (e.g., 20 °C s⁻¹) results in a quench effect in which the solute atoms lose the mobility before they can diffuse. With a wide range of medium cooling rates (e.g., 10 °C s⁻¹), a finite amount of carbon can segregate at the surface. The extremely slow cooling rate (e.g., 0.1 °C s⁻¹) allows carbon with enough time to diffuse into the bulk, so there will not be enough carbon segregated at the surface”. (The bold typeface referred to their experimental conditions.)

[R2] *ref. 19*: “...suspended graphene sheets are not perfectly flat: they exhibit intrinsic microscopic roughening such that the surface normal varies by several degrees and out-of-plane deformations reach 1 nm.”

[R3] ref. 40: “Their size being smaller than the lattice spacing, Li ions enable the large momentum transfer required for intervalley scattering events, that is, for scattering between electronic states near the K and K' points in momentum space.”

“In our experiment, Li diffusion along bilayer graphene may, in principle, occur at three different interfaces: $\text{SiO}_2/\text{graphene}$, graphene/graphene or graphene/vacuum . We demonstrate in the following that the second pathway dominates entirely, that is, Li diffuses in between the graphene sheets. Li intercalation and diffusion on top of or below the bilayer graphene cannot be detected under our experimental conditions.”

[R4] ref. 7: “Small variations in interplanar spacing are expected for graphite layers adjacent to the intercalate layer (the graphite bounding layers) because of the different charge distributions at this interface.” (Page 37)

“A ‘billiard ball’ model is used to compute the radii of the carbon and intercalate atoms based on the nearest-neighbour distances in the parent materials, so that the nearest-neighbour metallic distance defines the intercalate diameter.” (Page 50)

“Setton (1964) and Hérolde (1979) have also been able to account for the observed intercalate sandwich distances d_s in terms of a ‘billiard ball’ model where the diameters of the carbon atoms were taken to be equal to the c -axis graphite layer thickness $c_0 = 3.35 \text{ \AA}$, and for the intercalants, the ionic radii were used. One advantage of the model based on the metallic nearest-neighbour distances is that the same size ‘billiard balls’ are used to explain both in-plane and c -axis distances observed in the intercalate compounds.” (Page 50)

Reproduced **Table 10** at **Page 52**:

Redacted

Reviewers' comments:

Reviewer #1 (Remarks to the Author):

The new edits to the manuscript have made the paper considerably more understandable, which was my primary concern after the first round of edits.

The paper offers key insights into the mechanism of lithiation of graphenic materials, and will be of interest to a broad community.

It is my opinion that the paper should be accepted for publication.

Reviewer #2 (Remarks to the Author):

1. The response given by the authors about XRD results is still not convincing. I note the statement in the response,

...firstly, being smaller than the interplanar spacing (ca. $c_0 = 3.35\text{\AA}$), initial Li^+ ions (empirical diameter: 1.52\AA) apparently enable the free diffusion between the graphene interlayer (ref. 40[Note R3]).

This statement is inaccurate. Interplanar spacing is the distance between layers based on positions of the nuclei. There is no "free" space (recall the concept of the van der Waals diameter of carbon defined by fitting to parameters from graphite) in a pristine, 3.35\AA , interplanar space. Li^+ , and also other ions/molecules/atoms (with the possible exception of a proton) cannot freely diffuse inside/in-and-out or between, without increasing the graphite spacing along the c-axis [Ref A]. Ref. 40 is thus also unconvincing

LiC_6 - The hard sphere model is valid as Li^+ is small and located in the centers of C_6 rings. Li^+ does not "squeeze" into carbon atoms. There is thus no violation in the geometry of the model, "free" interlayer spacing is marked as spacing between dashed lines ($\sim 9.6\%$ increase in c spacing). The (002) distance of 0.372 nm fits well with 0.370 nm from experimental results, and this is the minimally expanded model that is given for Li^+ inside bilayer graphene.

However, the model given by the authors (below) is invalid based on simple geometric arguments:
 $d(002) = 2 \times \text{small, green } \text{Li}^+ \text{ radius} \not> (\text{in fact, } < <) 2 \times \text{carbon radius}$
 $d(002) = 0.3355\text{--}0.3369$ (as for lithiated bilayer graphene in this work in line 181-182). Even if intercalated Li are put in the ring-centers, it cannot be rationalized differently than the aforementioned hard sphere model, i.e., I do not see how the authors could rationalize a gallery height significantly smaller than 0.372 nm .

In addition, after lithiation the graphene layer stacking is AA, and DFT calculations shows that the interlayer distance in AA stacking is bigger than 3.5\AA [Ref B] with no Lithium, due to the van der Waals interactions, which means that the model proposed by the authors actually further lowers the C-to-C interlayer distance. It is hard to understand how this could be possible.

Also, from the "modified" model given in response (below), the authors seem to consider that the

intercalates (Li⁺ or Li⁰) can “squeeze” into carbon atoms, but this is not consistent with generally reported structures of GICs. Note for some large sized intercalates, there will be a smaller chance to even “squeeze” into the van der Waals diameter range of graphene after intercalation [Ref C].

2. Concerning the DFT calculations, they say they use the GGA approximation for the exchange-correlation functional. It is well known that GGA fails to reproduce van der Waals interactions, while the interaction of Li – C in this system should be dominated by such forces [Refs D, E]. Finally, a simple structure relaxation can fail to determine the interlayer distance accurately due to the possibility of local minima. Thus, in order to use a DFT calculation as the benchmark it has to be proved that this is the global minimum of the system (the same can be said about the case of the stated ring expansion).

Ref. A: S. Doyen-Lang, A. Charlier, L. Lang, M. F. Charlier, *Syn. Metals*. 58, 1993, 95.

Ref. B: X. Chen, F. Tian, C. Persson, W Duan, N Chen, *Sci. Rep.* 3, 2013, 3046.

Ref. C: H. Zhang, Y. Wu, W. Sirisaksoontorn, V. T. Reent, M. M. Lerner, *Chem. Mater.* 28, 2016, 969.

Ref. D: Z. Wang, S. M. Selbach, T. Grande, *RSC Adv.*, 4, 2014, 4069.

Ref. E: P. Ganesh, J. Kim, C. Park, M. Yoon, F. A. Roboredo, P. R. C. Kent, *J. Chem. Theory Comput.*, 10, 2014, 5318.

Reviewer #1: *The new edits to the manuscript have made the paper considerably more understandable, which was my primary concern after the first round of edits. The paper offers key insights into the mechanism of lithiation of graphenic materials, and will be of interest to a broad community. It is my opinion that the paper should be accepted for publication.*

Response: We greatly appreciate the reviewer's positive comments and quite valuable suggestions which have helped us improve the manuscript significantly.

Reviewer #2:

Q1-1: *The response given by the authors about XRD results is still not convincing. I note the statement in the response,*

... firstly, being smaller than the interplanar spacing (ca. $c_0 = 3.35 \text{ \AA}$), initial Li^+ ions (empirical diameter: 1.52 \AA) apparently enable the free diffusion between the graphene interlayer (ref. 40). This statement is inaccurate. Interplanar spacing is the distance between layers based on positions of the nuclei. There is no “free” space (recall the concept of the van der Waals diameter of carbon defined by fitting to parameters from graphite) in a pristine, 3.35 \AA , interplanar space. Li^+ , and also other ions/molecules/atoms (with the possible exception of a proton) cannot freely diffuse inside/in-and-out or between, without increasing the graphite spacing along the c -axis [Ref A]. Ref. 40 is thus also unconvincing.

Response: Firstly, thank the reviewer very much for the comments. However, after carefully checking Ref A, namely “Theoretical study of charge transfer in graphite intercalation compounds. S. Doyen-Lang, A. Charlier, L. Lang, M. F. Charlier, *Syn. Metals*. 58, **1993**, 95-107”, we find that this paper in fact didn’t draw the conclusion that “ions/molecules/atoms (with the possible exception of a proton) cannot freely diffuse inside/in-and-out or between, without increasing the graphite spacing along the c -axis”. The authors only calculated the “energy needed to separate the graphite layers” (e.g., ca. **1.495 kcal/mol** of C to form LiC_6 intercalation compound (3.706 \AA) from pristine graphite (3.350 \AA); **Table 3**), and that “needed to separate two first-neighbour carbon atoms” (or “to break the C-C in-plane bond”, ca. **4.252 kcal/mol** of C for LiC_6 under their given model; **Table 4**). Quite similar to our work and our cited papers (e.g., refs. 7,40), this theoretical work was also based on the **static** structure model of graphite intercalation compounds (GIC) proposed previously. Anyway, it is easy to know that

the very weak van der Waals forces can't limit the diffusion of Li^+ ions between the graphene interlayer, which in fact **are driven by the much stronger electromotive force in LIBs during the Li intercalation/de-intercalation process.**

By the way, please note that the published time of these related papers are all much earlier than **the discovery of graphene in 2004.** Thus, **the graphite-related structure models proposed by the researchers didn't consider the bilayer-graphene case, and in fact those authors usually mentioned that the graphene-sheet layers were over ~10 in the related models based on graphite** (refer to our detailed response to Referee #2: Q6 about the model in the 2nd round, which is included at the final Appendix).

Q1-2: *LiC₆ - The hard sphere model is valid as Li⁺ is small and located in the centers of C₆ rings. Li⁺ does not "squeeze" into carbon atoms. There is thus no violation in the geometry of the model, "free" interlayer spacing is marked as spacing between dashed lines (~9.6% increase in c spacing). The (002) distance of 0.372 nm fits well with 0.370 nm from experimental results, and this is the minimally expanded model that is given for Li⁺ inside bilayer graphene. However, the model given by the authors (below) is invalid based on simple geometric arguments:*

$$d_{(002)} - 2 \times \text{small, green Li}^+ \text{ radius} \not\approx (\text{in fact, } \ll) 2 \times \text{carbon radius}$$

$d_{(002)} = 0.3355\text{-}0.3369$ (as for lithiated bilayer graphene in this work in line 181-182). Even if intercalated Li are put in the ring-centers, it cannot be rationalized differently than the aforementioned hard sphere model, i.e., I do not see how the authors could rationalize a gallery height significantly smaller than 0.372 nm.

Response: About the above comments, it needs to stress that, our modified model in fact was just simply reproduced from the old hard sphere model **in Ref. 7** (shown below). In the simplified diagram in the response letter, we just **proportionally**: (1) changed the radii of Li atom and C atom (1.52 and 0.71 Å in **Fig. 28a**) into their diameters (3.04 and 1.42 Å, see the red and black circles, respectively), (2) added the empirical diameter of Li⁺ ion into it (1.52 Å, the green circle) in view of the **real valence state** of Li element in LiC₆ (ca. 0.9+ [refs. 11,14,44]), (3) added a purple scale bar of 3.36 Å on behalf of *d*₀₀₂, and (4) adjusted the viewing angle of the side view in **Fig. 28a** (equal to its left view) to make a more direct comparison for the related parameters. Hence, our modified figure is indeed equal to the old hard sphere model, and the relationship below is also valid and correct (refer to our simplified diagram below):

$$d_{(002)} - 2 \times \text{Li}^+ \text{ radius} (= 0.336 - 0.152 = 0.184 \text{ nm}) > 2 \times \text{carbon radius} (= 0.142 \text{ nm}).$$

In this way, the reviewer seemed to make an accounting error here.

To make the description in the updated manuscript more clear, the previous

Scheme 1a has been further modified, as follows:

B stacking
layer graphene

By the way, we noticed that **Ref A** pointed out that, “the process of intercalation results in a charge transfer between the intercalate layer and the host, generating **ionic binding**” and the charge transfer of Li in LiC₆ is ca. **0.94** (namely Li^{0.94+}, see its **Table 6**), which can favor our discussion at this part as well. Anyway, it is feasible for Li⁺ ion to diffuse in the graphene interlayer, whose d_{002} value is mainly decided by weak van der Waals interactions between the adjacent (lithiated) graphene sheets (refer to the side view for the **bilayer-graphene** case in the above **Figure (b)**).

Q1-3: In addition, after lithiation the graphene layer stacking is AA, and DFT calculations shows that the interlayer distance in AA stacking is bigger than 3.5 Å with no Lithium [**Ref. B**], due to the van der Waals interactions, which means that the model proposed by the authors actually further lowers the C-to-C interlayer distance. It is hard to understand how this could be possible.

Response: After carefully checking **Ref. B**, namely “**Interlayer interactions in graphites**. X. Chen, F. Tian, C. Persson, W Duan, N Chen, *Sci. Rep.* 3, 2013, 3046”, we

find that this paper was **a theoretical study only for pure graphite** and it didn't discuss the structure of C_6LiC_6 and bilayer graphene at all. That is to say, this paper didn't provide referable information with respect to our specific materials (just considering the structural changes caused the **ionic binding** of Li and C atoms mentioned in **Ref. A**). According to the explanations at **Q1-1,2** (also refer to our response to **Referee #2: Q6** included at the final **Appendix**), our experimental result about the d_{002} value should be comprehensible.

Q1-4: *Also, from the “modified” model given in response (below), the authors seem to consider that the intercalates (Li^+ or Li^0) can “squeeze” into carbon atoms, but this is not consistent with generally reported structures of GICs. Note for some large sized intercalates, there will be a smaller chance to even “squeeze” into the van der Waals diameter range of graphene after intercalation [Ref C].*

Response: We are sorry to make the reviewer misunderstand here again. As clarified above, our modified model strictly conforms to the old hard-sphere model, and they are both for the stable-state LiC_x phase. Thereinto, the upper one refers to **the side view of the configuration**, and the below one refers to its **top view**, which is not to express that “the intercalates (Li^+ or Li^0) can squeeze into carbon atoms” (refer to the updated **Scheme 1a** shown above). It is known that Li^+ ions enter into the graphene interlayer through edge planes or high-order defects (**Ref. 11** [*J. Am. Chem. Soc.* 134 (2012) 8646]). And, as explained above (i.e., $d_{(002)} - 2 \times \text{carbon radius} (= 3.36 - 1.42 = 1.94 \text{ \AA}) > Li^+$ diameter ($= 1.52 \text{ \AA}$)), the “free” interlayer spacing is large enough to perform the

Li^+ diffusion between it under the action of strong enough electromotive force in LIBs.

Based on these analyses (**Q1:1-4**), we believe that the reviewer misunderstood our statements about the XRD results and the cited hard-sphere model at this part, and there are no problems for both our experimental observation/explanation and the statement in **Ref. 40** [*Ultrafast lithium diffusion in bilayer graphene*. M. Kühne, F. Paolucci, J. Popovic, P. M. Ostrovsky, J. Maier, J. H. Smet, *Nature Nanotech.* 12, 2017, 895].

Q2: *Concerning the DFT calculations, they say they use the GGA approximation for the exchange-correlation functional. It is well known that GGA fails to reproduce van der Waals interactions, while the interaction of Li–C in this system should be dominated by such forces [Refs D, E]. Finally, a simple structure relaxation can fail to determine the interlayer distance accurately due to the possibility of local minima. Thus, in order to use a DFT calculation as the benchmark it has to be proved that this is the global minimum of the system (the same can be said about the case of the stated ring expansion).*

Response: We are sorry to make the reviewer misunderstand the DFT calculations.

On the one hand, judging from the description at **Line 184-187**, namely “*In addition to being consistent with the literature^{3,4,29,38,42} and previous DFT simulations for both graphenes¹⁰ and graphite⁴⁵, the later trend was also displayed by the following SAED results for Li-bilayered graphene samples (from ca. 3.42 Å at P4 to 3.50 Å at P1; Supplementary Fig. 17a,d)*”, the reviewer can know that we just cited the literature results [**Refs 10,45**] to support our experimental observations and we didn’t perform

the DFT calculation to study the interlayer distance d_{002} . By the way, we are sorry that Ref. 10 should be *J. Appl. Phys.* 112 (2012) 124323 not *Nature Nanotech.* 9 (2014) 739 in the previous version.

On the other hand, although we say “*generalized gradient approximation (GGA) of the electron exchange-correlation functional was adopted*” (see the **DFT calculations** part of the SI), however, the reviewer can note that we constructed “*the Li-inlaid unit cells of LiC₁₄ and LiC₆ with normal lattice parameters*” and “*restricted the Z-axis relaxation and merely allowed atoms to relax in X-Y plane*”. That is to say, our object of study here in fact was the Li-inlaid **monolayer graphene** sheet for LiC₆ as well as for LiC₁₄ (refer to **Supplementary Fig. 19b**), aiming to confirm the expansion phenomena of the **in-plane lattice** (or C₆-ring). Thus, at this condition, our DFT calculation only for **monolayer graphene** doesn't involve the determination of the interlayer distance d_{002} between **two graphene sheets** with **weak van der Waals (vdw) interactions**, and there is no problem in optimizing the structure (“*structure relaxation*”) of Li-inlaid **monolayer graphene** with **dominative strong chemical bonds** using merely GGA approximations (**Line 228-233**) [**Refs D, E**].

Besides, we noticed that in **Ref. D** (i.e., *RSC Adv.* 4 (2014) 4069), the difference of the optimized interlayer spacing **in graphite** between GGA and GGA+vdw-D2 is about **0.1~0.2 Å**. In fact, in our specific study, we also considered the influence of the interlayer distance on the simulated SAED pattern and thus tried different d_{002} values to check its influence (e.g., 3.33 Å, 4.00 Å, and 3.405 Å; see **Supplementary Figs 17c(x) and 19a**). The results show that the $d_{\{0110\}}$ and $d_{\{1210\}}$ values can keep constant

at 2.13 and 1.23 Å, indicating that the d_{002} value won't affect the **in-plane lattice spacings**. **Ref. E** (i.e., *J. Chem. Theory Comput.* 10 (2014) 5318) adopted a similar strategy, namely also using three d_{002} values as well as different levels of DFT theory (refer to its **Figure 3 below**), to calculate the in-plane diffusion barrier of Li atom (**not Li⁺ ion here**) across and along a C-C bond.

Redacted

By the way, given a fixed d_{002} value, the second-time DFT calculation, being performed to confirm the XPS-peak splitting phenomena by assuming three configurations of C_6LiC_6 (**Line 250** and **Fig. 4e,f**), had nothing to do with the determination of the interlayer distance, too.

All in all, our two DFT calculations are not involved with the case pointed out by the reviewer, namely needing to consider the weak van der Waals interactions between graphene sheets or to determine the interlayer distance accurately. We only carried out qualitative DFT study to explain the experimental phenomena.

To make the description more clear, the **Supplementary Information (Page 4:**

DFT calculations) has been further modified in the updated manuscript, as follows:

“... On the one hand, to confirm the *in-plane* lattice-expansion phenomena of lithiated bilayer graphene induced by electron beam irradiation (Fig. 3c and Supplementary Figs 17b and 18), the Li-inlaid unit cells of LiC₁₄ sheet and LiC₆ sheet with normal *in-plane* lattice parameters (as well as many other configuration models by adjusting the *Z-axis* spatial positions of Li atom) were constructed in the first step. Then, we restricted the *Z-axis* relaxation and merely allowed atoms to relax in *X-Y* plane (namely to achieve the smallest system energy) to obtain the optimized structure achievable in theory. Finally, the above CaRine Crystallography 3.1 software was applied to simulate the *in-plane* lattice parameters of the new structure (namely the optimized unit cell) used for comparison with those experimental values by SAED.

On the other hand, considering different configured Li sites relative to carbon lattice in *X-Y* plane, we fixed the interlayer spacing (namely the *d₀₀₂* value), restricted the *in-plane* relaxation, and merely allowed individual atoms to relax along *Z-axis* direction in the binding-energy calculations...”

Appendix:

(1) The 1st-round comments by Reviewer #2 and our response:

Question 7: (Line 177) The authors state that the gallery expansion from G-G to G-Li-G is from 0.335 nm to 0.336 nm (P2, LiC₁₂₋₂₈), according to their XRD result. How is it possible to insert Li⁺ in 0.001 nm? (The gallery height of the graphite intercalation compounds (GIC) LiC_x is 0.370 nm).

Response: We are sorry to make the reviewer misunderstand here. Here we intend to state that, as the Li density increases in the 2D interlayer of bilayer graphene, the d_{002} value (or the average gallery height; refer to the below supplementary Scheme 1a in the revised manuscript) will become larger and larger due to the enhanced interaction of the middle Li layer and the bilateral graphene layers.

Scheme 1 | (a) Evolution of the lamellar distances induced by staged Li intercalation.

(2) The 2nd-round comments by Reviewer #2 and our response:

Q6: (Line 177) The response given by the authors is not convincing. Since upon lithiation the interlayer distance increases by 0.001-0.002 nm from 0.335 nm for G-G to 0.336-0.337 nm for G-Li-G then, considering the Li⁺ diameter, it is unclear how Li can intercalate between the layers and fit the aforementioned spacing. Besides, if some of the Li⁺ in the planar fractal model were placed at the vertices of C₆ rings, then the gallery expansion would be even bigger than putting Li⁺ only at the centers of C₆ rings, which is typically meaning a 0.370 nm layer spacing in LiC₆.

Response: We thank the reviewer very much for the further discussion.

On the one hand, firstly, being smaller than the interplanar spacing (ca. $c_0 = 3.35$ Å), initial Li⁺ ions (empirical diameter: 1.52 Å) apparently enable the free diffusion

between the graphene interlayer (*ref. 40*^[Note R3]), until the resultant electrodynamic force becomes too small at the PI stage of C₆LiC₆ (*Scheme 1b*). Secondly, for the Li atoms (empirical diameter: 3.04 Å) stabilizing at the graphene interlayer (by Li⁺ + e⁻ → Li), according to the studies in *refs. 11, 14 and 44* and our XPS results (*Fig. 4d*), in fact they are always ionic due to the charge redistribution (*ref. 7*^[Note R4]) or interaction between it and its nearest neighbor C atoms (analogous to ordinary ionic compound), ca. +0.9|e| for Li at the center of C₆ rings (see the structure in *Fig. 3a(iv)*). As for the case that Li atom locates at the vertex of a C₆ ring (still at the center of another C₆ ring, refer to the insets of *Fig. 4e and 4f* for Case 1), this situation only occurs at the initial phases with low Li concentration, like P5 and P4 with ARB stacking configuration (*Scheme 1b*). Thus, the quite limited number of C₆-Li units would not greatly influence the average layer spacing determined by the pristine graphene sheets, in particular for bilayer graphene to enable only one Li layer without the interference of adjacent layers.

On the other hand, *ref. 7* (“*Intercalation compounds of graphite*”, *Adv. Phys.* 51 (2002) 1–186; a review article originally published in 1981 by M. S. Dresselhaus and G. Dresselhaus), gave two structural models (proposed 40–50 years ago) and detailed theoretical explanations (refer to its **Table 10**^[Note R4]) to describe the geometrical relations for first stage LiC₆ of graphite (pristine interplanar spacing $c_0 = 3.35$ Å). One was a **hard sphere** model using the radii of C (0.71 Å) and Li atoms (**Fig. 28a** in *ref. 7*, **namely Figure (a) below**), and the other was a “**billiard ball**” model using Li⁺ radius^[Note R4]. Considering the essential structure differences between thick graphite and graphenes (especially bilayer graphene), **together with the inevitable model error (e.g., hard sphere ≠ microscopic particles at the atomic level), instrument error/precision, and technique difference**, the unequal interplanar lattice spacings, such as 0.372 nm for LiC₆ of graphite^[Note R4] (possibly decided by the Li layers with stronger interactions rather than the C layers, refer to **Figure (b) below**), 0.336 nm for P2 of multilayer graphene (by XRD in *Supplementary Fig. 16*), and 0.350 nm for P1 of bilayer graphene (by SAED in *Supplementary Fig. 17d*), may be understandable, too. Despite the different specific d_{002} values, the lattice change trends induced by Li

intercalation are consistent, in keeping with the varied charge distribution states of the C and Li layers as well.

Consequently, to better address this concern, we have added the structure model for the C₆-Li unit (refer to **Figure (c)** below) in the revised **Scheme 1**, and related discussion can be seen in **Line 166, 183 and 257** of the updated manuscript, respectively.

[R3] *ref. 40*: “Their size being smaller than the lattice spacing, Li ions enable the large momentum transfer required for intervalley scattering events, that is, for scattering between electronic states near the K and K’ points in momentum space.”

“In our experiment, Li diffusion along bilayer graphene may, in principle, occur at three different interfaces: SiO₂/graphene, graphene/graphene or graphene/vacuum. We demonstrate in the following that the second pathway dominates entirely, that is, Li diffuses in between the graphene sheets. Li intercalation and diffusion on top of or below the bilayer graphene cannot be detected under our experimental conditions.”

[R4] *ref. 7*: “Small variations in interplanar spacing are expected for graphite layers adjacent to the intercalate layer (the graphite bounding layers) because of the different

charge distributions at this interface.” (Page 37)

“A ‘billiard ball’ model is used to compute the radii of the carbon and intercalate atoms based on the nearest-neighbour distances in the parent materials, so that the nearest-neighbour metallic distance defines the intercalate diameter.” (Page 50)

*“Setton (1964) and Hérold (1979) have also been able to account for the observed intercalate sandwich distances d_s in terms of a ‘billiard ball’ model where the diameters of the carbon atoms were taken to be equal to the c-axis **graphite layer thickness** $c_0 = 3.35 \text{ \AA}$, and for the intercalants, the ionic radii were used. One advantage of the model based on the metallic nearest-neighbour distances is that the same size ‘billiard balls’ are used to explain both in-plane and c-axis distances observed in the intercalate compounds.” (Page 50)*

Reproduced **Table 10** at **Page 52**:

Redacted

Reviewers' comments:

Reviewer #3 (Remarks to the Author):

Tracking #: NCOMMS-17-27749-B

Review of Manuscript

Lithium intercalation into bilayer graphene

By Kemeng Ji^{1,2,†,*}, Jiuhui Han^{1,2,†}, Akihiko Hirata¹, Takeshi Fujita¹, Yuhao Shen^{1,3}, Shoucong Ning⁴, Pan Liu¹, Hamzeh Kashani^{1,2}, Yuan Tian^{1,2}, Yoshikazu Ito^{5,6,*}, Jun-ichi Fujita⁵, Yutaka Oyama^{2,*}

Detailed experimental work has been done and important data has been collected, however, I would like to bring two main concerns for the authors which may strongly affect to the model they considered in this article and their follow up conclusions.

Therefore, at this stage I would not recommend publishing the manuscript in Nature Communications journal.

1. Definition of Bilayer - The Authors confirmed predominate Bernal bilayer based on SAED results following by confirmation of G and 2D modes of Raman spectra (line 71-88). Bernal Bilayer like graphite and Single layer graphene performs similar SAED pattern-single hexagonal pattern. In the case of misoriented multilayer, circle of dots may appear on SAED pattern without any Lithium Intercalation (T. Paronyan et al, <https://www.nature.com/articles/srep39944>).

In this case, SAED confirms a rotated misoriented Bilayer (not a Bernal stacking as the authors confirmed) where the Rotation angle and the interlayer van der Waals interaction can be changed significantly (Ref. a) J. Hicks, M. Sprinkle, K. Shepperd, and F. Wang, Symmetry breaking in commensurate graphene rotational stacking: Comparison of theory and experiment, PHYSICAL REVIEW B 83, 205403 (2011) which reflects also on Raman Spectra (Ref. b) K. Kim et al., Raman Spectroscopy Study of Rotated Double-Layer Graphene: Misorientation-Angle Dependence of Electronic Structure, Phys. Rev. Lett. 108, 246103 (2012), Ref. c) P. Ponchara Phys. Rev. B 78, 113407 (2008).

Also, the intensity ratio of G and 2D modes of misoriented Multilayer graphene and their shift is sensitive to the rotation angle between layers, and here, the definition of the number of graphene layers by Raman G and 2D modes for misoriented graphene can be not proper (Fig. 1e), specially using 514.5 nm excitation Laser.

Therefore, the authors must consider the case of Misoriented Bi-layer (or more) based on their SAED and Raman data) which may directly affect to their Calculations as the van der Waals interplanar forces of misoriented Bilayer is much weaker compared to Bernal Bilayer.

2. XRD data of lithiated Graphene.

If the authors are considered Bernal Bilayer, they should consider this earlier calculations by authors K. Rytkönen, J. Akola, and M. Manninen (Ref. e) PHYSICAL REVIEW B 75, 075401 2007) where they showed that "The adatoms bind at the hollow site graphite hexagon, with Li lying closest to 1.84 Å and Cs farthest 3.75 Å from the surface" which is in conflict of their result assuming no extension of interlayer distance.

In the case of Misoriented/incommensurate Bilayer graphene XRD peak can be weakened or

disappeared by depletion of interlayer interaction due to Lithium layer absorption into interlayer space. Once Bilayer or multilayer graphene loses interlayer interaction, XRD peak may disappear as it may act as a Single layer Graphene (no XRD Bragg peaks) in Bilayer configuration. No change in XRD peak position can be caused if partially there is no intercalation into Bernal stacking areas.

Reviewer #3: *Detailed experimental work has been done and important data has been collected, however, I would like to bring two main concerns for the authors which may strongly affect to the model they considered in this article and their follow up conclusions. Therefore, at this stage I would not recommend publishing the manuscript in Nature Communications journal.*

Q1: Definition of Bilayer.

The authors confirmed predominate Bernal Bilayer based on SAED results following by confirmation of G and 2D modes of Raman spectra (line 71-88). Bernal Bilayer like graphite and Single layer graphene performs similar SAED pattern-single hexagonal pattern. In the case of Misoriented Multilayer, circle of dots may appear on SAED pattern without any Lithium Intercalation (Ref. a).

In this case, SAED confirms a rotated misoriented Bilayer (not a Bernal stacking as the authors confirmed) where the rotation angle and the interlayer van der Waals interaction can be changed significantly (Ref. b), which reflects also on Raman Spectra (Refs. c,d).

Also, the intensity ratio of G and 2D modes of misoriented Multilayer graphene and their shift is sensitive to the rotation angle between layers, and here, the definition of the number of graphene layers by Raman G and 2D modes for misoriented graphene can be not proper (Fig. 1e), specially using 514.5 nm excitation Laser.

Therefore, the authors must consider the case of Misoriented Bilayer (or more) based on their SAED and Raman data, which may directly affect to their calculations as the van der Waals interplanar forces of Misoriented Bilayer is much weaker compared to Bernal Bilayer.

Notes:

[Ref. a] Paronyan, T. M. *et al.* Incommensurate graphene foam as a high capacity lithium intercalation anode. *Sci. Rep.* 7 (2017) 39944.

[Ref. b] Hicks, J. *et al.* Symmetry breaking in commensurate graphene rotational stacking: Comparison of theory and experiment, *Phys. Rev. B* 83 (2011) 205403.

[Ref. c] Kim, K. *et al.* Raman spectroscopy study of rotated double-layer graphene: Misorientation-angle dependence of electronic structure, *Phys. Rev. Lett.* 108 (2012) 246103.

[Ref. d] Poncharal, P. *et al.* Raman spectra of misoriented bilayer graphene. *Phys. Rev. B* 78 (2008) 113407.

Response: Thank the reviewer very much for reminding us of the possible presence and influence of the rotated/misoriented bilayer graphene in our analysis.

Firstly, as 2D carbon crystals, it is well known that **flat** Bernal bilayer graphene, graphite, and single-layer graphene share similar **regular hexagonal** SAED patterns.

However, as fully proved (refer to **Refs. 19–23,43** in the manuscript), the intensity ratio of the {0110} to {1210} peaks for Bernal bilayer graphene ($I_{\{0110\}}/I_{\{1210\}} \approx 0.4-0.5$) are quite different from those of other graphenic carbons including monolayer graphene ($I_{\{0110\}}/I_{\{1210\}} > 1$) and Bernal multilayer graphene ($I_{\{0110\}}/I_{\{1210\}} < 1$). This is also the reason that we were able to identify the bilayer feature of our graphene foam material by acquiring SAED patterns from the **flat enough region** of its graphene sheets, which can help avoid the possible influence from the defects and fold structure at the sheet boundaries (**Ref. c**). As to the SAED pattern with circle of dots for **multilayer graphene sheets** in **Ref. a** (see its Supplementary Fig. 9 **included in the below Figure R1**), we think this pattern was more possibly from a polycrystalline area assembled by several multilayered graphene sheets with varied spatial directions and layers, rather than a standalone graphene sheet with **incommensurate/misoriented** stacking configuration (refer to Fig. 3 from **Ref. 15 in Figure R1**). In fact, as indicated by many reports including **Refs. b–d** cited by the reviewer, Bernal A-B stacking is the conventional/common construction and it is really difficult to use the general methods (including chemical synthesis and mechanical exfoliation) to obtain graphenic carbon in which the two graphene layers are relatively rotated by an arbitrary angle. For example, in **Ref. b** the authors used the **special substrate** of SiC (0001) to grow multilayer epitaxial graphene with non-Bernal stacking (also refer to the quasicrystalline 30° twisted bilayer graphene grown on 4H-SiC in *Science* 361 (2018) 782 by Ahn. et al.), and in **Refs. c,d** the bilayer graphenes with different/random rotation angles were obtained via **artificial stacking**. Therefore, if the authors of **Ref. a** would

like to use the SAED technique to prove their **CVD-grown graphene** material is incommensurate/misoriented (namely not Bernal A-B stacking), they should apply a single flat sheet **but not** such a polycrystalline region by following the related studies/operations like shown in **Ref. c**, which says: “*We are cautious to avoid samples with tilt grain boundaries [27: ACS Nano 5 (2011) 2142] or local fold structure from the transfer process [30: Phys. Rev. B 83 (2011) 245433], which would give more than two sets of hexagonal diffraction spots over the sample area*”. The same situation can also be observed in our TEM specimen using pristine graphene foam shown in **Supplementary Fig. 3e** (e.g., **Region 2**, see it in the below **Figure R2**). However, it is worth mentioning that this specific image for suspended and damaged graphene sheets would display different curvatures from those of the most intact ones constructing the large entire graphene foam (refer to our **Supplementary Figs. 2d and 3c**). The reason that we elucidated these curved regions with distorted SAED patterns, was only to illustrate that only the **normal-incidence** SAED pattern from the **flat region** and thus with **regular** hexagonal diffraction pattern can be used for the typical mechanism/structural analysis in this study. Similarly, with too much contingency and being not representative of the sample’s dominant morphology/feature, defect-rich graphene boundaries and too small regions (including **local misoriented area**, refer to the schematic from **Ref. b** **included in Figure R2** as well as Figs. 2,6 in **Ref. a** and Fig. 1(d,e) in *ACS Omega* 2 (2017) 3691 by our collaborators), were also avoided using in the mechanism/structural analysis. Therefore, using HRTEM results (**Supplementary Fig. 14**) we only discussed the influences of Li intercalation and beam irradiation.

Supplementary Figure 9. SAED analysis of pristine graphene sheets. (a) HRTEM image of incommensurate multilayer sheets. (b) SAED pattern of the same area. Reproduced from Ref. a.

Figure R1. Comparative SAED patterns of pristine monocrystalline and polycrystalline graphene sheets.

Redacted

[Ref. c] **FIG. 1a,b** (a) TEM image of misoriented double-layer graphene. The graphene sample is suspended in a hole of 2 μm diameter. (b) Diffraction pattern of the graphene sample shown in (a). The two sets of hexagonal patterns are relatively rotated by 21° . An electron beam size of $\sim 1 \mu\text{m}$ is used for diffraction acquisition.

[Ref. b] **FIG. 1a** Schematic of the commensurate rotation of two graphene sheets. The commensurate angle θ leads to a large supercell structure.

Figure R2. Distinction between the concepts of curvature and misorientation.

Secondly, our related description at Supplementary Fig. 3 was as follows: “*The bended graphene sheets in (e) exhibited **distorted hexagonal diffraction patterns**, quite different from that from the **flat part (d)** and indicating the effects of the **curvature**”.*

Obviously, we did not confirm or say that it was “*a rotated misoriented Bilayer*” or “*not a Bernal stacking*”. As mentioned above, these SAED patterns in Supplementary Fig. 3e in fact can only indicate that the **curved/bended graphene** regions will produce **non-regular** hexagonal SAED patterns based on the current literature reports (including Refs. b–d studying only incommensurate/misoriented/rotated **graphene flat sheets**) and our years of experience on this kind of graphene foam (refer to Ref. 17: Angew. Chem. Int. Ed. 53 (2014) 4822 by Ito *et al.*). Just based on the above, it can be clearly seen that the SAED patterns we adopted for analysis were all carefully captured from the internal regions of flat graphene sheets, which were further distinguished using a geometrical method to confirm their availability for the subsequent structure analysis.

Furthermore, as indicated above, A-B stacking is the natural structure for graphenic carbons, and the “*curved/bended graphene*” (sampled in our study) and “*rotated misoriented graphene*” (studied in Refs. b–d) are actually two totally different concepts (refer to **Figure R2**). The former (Supplementary Fig. 3e) describes the **relatively macro** geometrical shape of the entire graphenic material/sheet, while the latter is to illustrate the **micro** stacking configuration of its two graphene layers (refer to FIG. 1a in Ref. b and FIG. 1a,b in Ref. c included in **Figure R2**). In other words, Refs. c,d in fact discussed the influence of **rotation/misorientation angle** rather than **geometrical curvature** on the Raman spectra. As to the latter’s influence, this macro appearance

seems to exert no influence on the Raman bands based on our characterization/research results over the past few years (refer to the description at **Line 81** in the manuscript) and in view of the origin of the *D*, *G*, and *2D* bands (refer to **Ref. c** and our related discussion in the **SI**). Hence, we believe there are no problems using the static Raman data for the definition of the number of graphene layers in our study (in particular with the other parallel verifications including the measured specific capacities in **Supplementary Fig. 7a,b**). After all, it is easy to identify bilayer graphene (also monolayer graphene) thanks to its quite different Raman features from those of few- and multi-layer graphenes as well as graphite.

When it comes to the **interlayer van der Waals force**, we notice that the “**interlayer interaction**” in the cited papers **Refs. b–d** in fact discussed **the coherent interlayer movement of electrons** (associated with electronic band structure/Dirac cones in **Physics**) but not the **chemical** intermolecular force between the two graphene layers relevant to this study. It is well known that this molecular interaction, namely electrostatic attractive or repulsive interaction between molecules or atoms by definition, is rather weak compared with various binding forces or electromotive forces in LIBs (considering the easily relative sliding of two graphene sheets). Therefore, it obviously cannot affect the Li^+ intercalation/de-intercalation process (refer to our kinetics analysis based on **Fig. 2d,e**) accompanied by the slight change of the interlayer spacing (i.e., d_{002} value) and **the continuous transformation of stacking configurations** of bilayer graphene. Related description can also be found in **Ref. a**: “*When lithium intercalates within commensurately-stacked graphene layers such as*

graphite, the sheets rotate from AB ($d = 3.35 \text{ \AA}$) into AA stacking ($d = 3.6 \text{ \AA}$) bringing carbon honeycombs directly above and below one another⁴²”, in favor of our discussion. (By the way, the misoriented-bilayer and Bernal-bilayer structures discussed above should also occur during the configuration transformation. This means that the initial stacking modes in fact would be not so important for the electrochemical Li storage in bilayer graphene, for that the highly reversible Li^+ intercalation/de-intercalation processes (Supplementary Fig. 4b-d) have the ability to adjust the relative rotation angle between layers.) In addition, the theoretical calculations of Li-storage capacities in Supplementary Fig. 1a also have nothing to do with the van der Waals interplanar force, which were based on the most basic LiC_6 unit (372 mAh g^{-1} in the theoretical capacity, well-known for fully graphitic carbon anodes in LIBs). So do the other calculations in this study, including the DFT ones (refer to our previous response to **Referee #2: Q2** in the 3rd-round review, which is included at the final Appendix).

Based on these analyses, we believe that there are no problems about our discussions and conclusions based on the related SAED and Raman data, as well as about the definition of bilayer graphene and various calculations. However, thanks to the reviewer’s reminder, we find that the long-time beam irradiation may also cause local misorientation/rotation of the **lithiated bilayer graphene** except the influences indicated in the initial manuscript. Related discussions now have been included at Supplementary Figs 14, 17c, and 18. Meanwhile, to describe the SAED patterns of curved/folding graphene sheets more clearly, Supplementary Fig. 3e has been modified properly by including the references such as Ref. c and *Phys. Rev. B* 83 (2011) 245433).

Q2: XRD data of lithiated Graphene.

If the authors are considered Bernal Bilayer, they should consider this earlier calculations by authors K. Rytönen, J. Akola, and M. Manninen (Ref. e), where they showed that “The adatoms bind at the hollow site graphite hexagon, with Li lying closest to 1.84 Å and Cs farthest 3.75 Å from the surface” which is in conflict of their result assuming no extension of interlayer distance.

In the case of Misoriented/incommensurate Bilayer graphene XRD peak can be weakened or disappeared by depletion of interlayer interaction due to Lithium layer absorption into interlayer space. Once Bilayer or Multilayer graphene loses interlayer interaction, XRD peak may disappear as it may act as a Single Layer Graphene (no XRD Bragg peaks) in Bilayer configuration. No change in XRD peak position can be caused if partially there is no intercalation into Bernal stacking areas.

Notes:

[Ref. e] Rytönen, K. *et al.* Density functional study of alkali-metal atoms and monolayers on graphite (0001). *Phys. Rev. B* 75 (2007) 075401.

Response: Thank the reviewer for the deep consideration about the influence of the stacking configuration of two graphene sheets for bilayer graphene on the XRD result.

However, it seems that the reviewer misunderstood our study by saying “*their result assuming no extension of interlayer distance*”, because we really made no such assumption in this study, as definitely shown in Supplementary Fig. 16b,c and Scheme 1a and discussed at **Line 184-187** and **Line 274-275** (e.g., “*such as (1) the slightly increased interlayer space (Scheme 1a) by the XRD and SAED patterns*”). According

to the theoretical work Ref. e cited by the reviewer, we notice that its authors in fact intended to show that, “*The surface separation of alkali-metal adatoms (Li, Na, K, Rb, Cs) [Fig. 1(b); shown below] increases as the atomic radius increases, and the d_{\perp} values (namely ‘the vertical displacement from the uppermost graphene plane’) range between 1.84 Å (for Li adatom) and 3.44 Å (for Cs adatom). For the (2×2) monolayers, d_{\perp} increases systematically as the metal films decouple from the surface, and the corresponding range of values is 2.02 Å (for Li)–3.75 Å (for Cs).*” According to the results, the formed Li-metal layer in graphene interlayer (considering the final PI phase) would enable a larger d_{002} lattice spacing than a single Li adatom (considering the initial P5/P4 phases), in accordance with our experimentally observed trend in Supplementary Fig. 16c. Also, we notice that the comparative DFT simulated data in its Table II (shown below) can support our discussion about the model in Scheme 1a (refer to our detailed response to **Referee #2: Q1** in the 3rd-round review, included at the final Appendix).

Redacted

As for the XRD peak of **misoriented/incommensurate bilayer graphene**, we agree that the lithium layer intercalated into its interlayer would possibly influence the peak intensity and position of its XRD pattern, as reflected by the analogous results we obtained using multilayered graphene-based foam samples (Supplementary Fig. 16) and the aforesaid description in Ref. a (see the response to **Q1**). However, as what we indicate at the same time, “*the (002) peak was too weak to be observed from pristine bilayer graphene foam as well as its lithiated products*” (refer to Supplementary Fig. 16a). In other words, the current resolution/sensitivity of the XRD technique is still too low to perform such an experimental study on bilayer graphene (in particular using its plane sheet with tiny mass). A great deal of effort should be required in this field for the direct observation on bilayer graphene.

Appendix — The 3rd-round comments by Reviewer #2 and our response:

Q1-1: *The response given by the authors about XRD results is still not convincing. I*

note the statement in the response,

... firstly, being smaller than the interplanar spacing (ca. $c_0 = 3.35 \text{ \AA}$), initial Li^+ ions (empirical diameter: 1.52 \AA) apparently enable the free diffusion between the graphene interlayer (ref. 40). This statement is inaccurate. Interplanar spacing is the distance between layers based on positions of the nuclei. There is no “free” space (recall the concept of the van der Waals diameter of carbon defined by fitting to parameters from graphite) in a pristine, 3.35 \AA , interplanar space. Li^+ , and also other ions/molecules/atoms (with the possible exception of a proton) cannot freely diffuse inside/in-and-out or between, without increasing the graphite spacing along the c-axis [Ref A]. Ref. 40 is thus also unconvincing.

Response: Firstly, thank the reviewer very much for the comments. However, after carefully checking Ref A, namely “Theoretical study of charge transfer in graphite intercalation compounds. S. Doyen-Lang, A. Charlier, L. Lang, M. F. Charlier, *Syn. Metals*. 58, 1993, 95-107”, we find that this paper in fact didn’t draw the conclusion that “ions/molecules/atoms (with the possible exception of a proton) cannot freely diffuse inside/in-and-out or between, without increasing the graphite spacing along the c-axis”. The authors only calculated the “energy needed to separate the graphite layers” (e.g., ca. **1.495 kcal/mol** of C to form LiC_6 intercalation compound (3.706 \AA) from pristine graphite (3.350 \AA); **Table 3**), and that “needed to separate two first-neighbour carbon atoms” (or “to break the C-C in-plane bond”, ca. **4.252 kcal/mol** of C for LiC_6

under their given model; **Table 4**). Quite similar to our work and our cited papers (e.g., **refs. 7,40**), this theoretical work was also based on the **static** structure model of graphite intercalation compounds (GIC) proposed previously. Anyway, it is easy to know that the very weak van der Waals forces can't limit the diffusion of **Li⁺ ions** between the graphene interlayer, which in fact **are driven by the much stronger electromotive force in LIBs during the Li intercalation/de-intercalation process**.

By the way, please note that the published time of these related papers are all much earlier than **the discovery of graphene in 2004**. Thus, **the graphite-related structure models proposed by the researchers didn't consider the bilayer-graphene case, and in fact those authors usually mentioned that the graphene-sheet layers were over ~10 in the related models based on graphite** (refer to our detailed response to Referee #2: Q6 about the model in the 2nd round, which is included at the final **Appendix**).

Q1-2: *LiC₆ - The hard sphere model is valid as Li⁺ is small and located in the centers of C₆ rings. Li⁺ does not “squeeze” into carbon atoms. There is thus no violation in the geometry of the model, “free” interlayer spacing is marked as spacing between dashed lines (~9.6% increase in c spacing). The (002) distance of 0.372 nm fits well with 0.370 nm from experimental results, and this is the minimally expanded model that is given for Li⁺ inside bilayer graphene. However, the model given by the authors (below) is invalid based on simple geometric arguments:*

$$d_{(002)} - 2 \times \text{small, green Li}^+ \text{ radius} \not\approx (\text{in fact, } \ll) 2 \times \text{carbon radius}$$

$$d_{(002)} = 0.3355 - 0.3369 \text{ (as for lithiated bilayer graphene in this work in line 181-}$$

182). Even if intercalated Li are put in the ring-centers, it cannot be rationalized differently than the aforementioned hard sphere model, i.e., I do not see how the authors could rationalize a gallery height significantly smaller than 0.372 nm.

Response: About the above comments, it needs to stress that, our modified model in fact was just simply reproduced from the old hard sphere model in Ref. 7 (shown below). In the simplified diagram in the response letter, we just **proportionally**: (1) changed the radii of Li atom and C atom (1.52 and 0.71 Å in **Fig. 28a**) into their diameters (3.04 and **1.42 Å**, see the red and black circles, respectively), (2) added the empirical diameter of Li⁺ ion into it (**1.52 Å**, the green circle) in view of the **real valence state** of Li element in LiC₆ (ca. **0.9+** [*refs. 11,14,44*]), (3) added a purple scale bar of **3.36 Å** on behalf of *d*₀₀₂, and (4) adjusted the viewing angle of the side view in **Fig. 28a** (equal to its left view) to make a more direct comparison for the related parameters. Hence, our modified figure is indeed equal to the old hard sphere model, and the relationship below is also valid and correct (refer to our simplified diagram below):

$$d_{(002)} - 2 \times Li^+ \text{ radius } (= 0.336 - 0.152 = 0.184 \text{ nm}) > 2 \times \text{carbon radius } (= 0.142 \text{ nm}).$$

In this way, the reviewer seemed to make an accounting error here.

To make the description in the updated manuscript more clear, the previous

Scheme 1a has been further modified, as follows:

A-B stacking
bilayer graphene

By the way, we noticed that **Ref A** pointed out that, “the process of intercalation results in a charge transfer between the intercalate layer and the host, generating **ionic binding**” and the charge transfer of Li in LiC₆ is ca. **0.94** (namely Li^{0.94+}, see its **Table 6**), which can favor our discussion at this part as well. Anyway, it is feasible for Li⁺ ion to diffuse in the graphene interlayer, whose d_{002} value is mainly decided by weak van der Waals interactions between the adjacent (lithiated) graphene sheets (refer to the side view for the **bilayer-graphene** case in the above **Figure (b)**).

Q1-3: *In addition, after lithiation the graphene layer stacking is AA, and DFT calculations shows that the interlayer distance in AA stacking is bigger than 3.5 Å with no Lithium [Ref. B], due to the van der Waals interactions, which means that the model proposed by the authors actually further lowers the C-to-C interlayer distance. It is hard to understand how this could be possible.*

Response: After carefully checking Ref. B, namely “*Interlayer interactions in graphites*. X. Chen, F. Tian, C. Persson, W Duan, N Chen, *Sci. Rep.* 3, 2013, 3046”, we find that this paper was a **theoretical study only for pure graphite** and it didn’t discuss the structure of C₆LiC₆ and bilayer graphene at all. That is to say, this paper didn’t provide referable information with respect to our specific materials (just considering the structural changes caused the **ionic binding** of Li and C atoms mentioned in Ref. A). According to the explanations at Q1-1,2 (also refer to our response to Referee #2: Q6 included at the final Appendix), our experimental result about the d_{002} value should be comprehensible.

Q1-4: *Also, from the “modified” model given in response (below), the authors seem to consider that the intercalates (Li^+ or Li^0) can “squeeze” into carbon atoms, but this is not consistent with generally reported structures of GICs. Note for some large sized intercalates, there will be a smaller chance to even “squeeze” into the van der Waals diameter range of graphene after intercalation [Ref C].*

Response: We are sorry to make the reviewer misunderstand here again. As clarified above, our modified model strictly conforms to the old hard-sphere model, and they are

both for the stable-state LiC_x phase. Thereinto, the upper one refers to **the side view of the configuration**, and the below one refers to its **top view**, which is not to express that “the intercalates (Li^+ or Li^0) can squeeze into carbon atoms” (refer to the updated **Scheme 1a** shown above). It is known that Li^+ ions enter into the graphene interlayer through edge planes or high-order defects (**Ref. 11** [*J. Am. Chem. Soc.* 134 (2012) 8646]). And, as explained above (i.e., $d_{(002)} - 2 \times \text{carbon radius} (= 3.36 - 1.42 = 1.94 \text{ \AA}) >$ Li^+ diameter ($= 1.52 \text{ \AA}$)), the “free” interlayer spacing is large enough to perform the Li^+ diffusion between it under the action of strong enough electromotive force in LIBs.

Based on these analyses (**Q1:1-4**), we believe that the reviewer misunderstood our statements about the XRD results and the cited hard-sphere model at this part, and there are no problems for both our experimental observation/explanation and the statement in **Ref. 40** [*Ultrafast lithium diffusion in bilayer graphene*. M. Kühne, F. Paolucci, J. Popovic, P. M. Ostrovsky, J. Maier, J. H. Smet, *Nature Nanotech.* 12, 2017, 895].

Q2: *Concerning the DFT calculations, they say they use the GGA approximation for the exchange-correlation functional. It is well known that GGA fails to reproduce van der Waals interactions, while the interaction of Li–C in this system should be dominated by such forces [Refs D, E]. Finally, a simple structure relaxation can fail to determine the interlayer distance accurately due to the possibility of local minima. Thus, in order to use a DFT calculation as the benchmark it has to be proved that this is the global minimum of the system (the same can be said about the case of the stated ring expansion).*

Response: We are sorry to make the reviewer misunderstand the DFT calculations.

On the one hand, judging from the description at **Line 184-187**, namely “*In addition to being consistent with the literature^{3,4,29,38,42} and previous DFT simulations for both graphenes¹⁰ and graphite⁴⁵, the later trend was also displayed by the following SAED results for Li-bilayered graphene samples (from ca. 3.42 Å at P4 to 3.50 Å at P1; Supplementary Fig. 17a,d)*”, the reviewer can know that we just cited the literature results [Refs 10,45] to support our experimental observations and we didn’t perform the DFT calculation to study the interlayer distance d_{002} . By the way, we are sorry that Ref. 10 should be *J. Appl. Phys.* 112 (2012) 124323 not *Nature Nanotech.* 9 (2014) 739 in the previous version.

On the other hand, although we say “*generalized gradient approximation (GGA) of the electron exchange-correlation functional was adopted*” (see the **DFT calculations** part of the SI), however, the reviewer can note that we constructed “*the Li-inlaid unit cells of LiC₁₄ and LiC₆ with normal lattice parameters*” and “*restricted the Z-axis relaxation and merely allowed atoms to relax in X-Y plane*”. That is to say, our object of study here in fact was the Li-inlaid **monolayer graphene** sheet for LiC₆ as well as for LiC₁₄ (refer to **Supplementary Fig. 19b**), aiming to confirm the expansion phenomena of the **in-plane lattice** (or C₆-ring). Thus, at this condition, our DFT calculation only for **monolayer graphene** doesn’t involve the determination of the interlayer distance d_{002} between **two graphene sheets with weak van der Waals (vdw) interactions**, and there is no problem in optimizing the structure (“*structure relaxation*”) of Li-inlaid **monolayer graphene** with **dominative strong chemical**

bonds using merely GGA approximations (**Line 228-233**) [Refs D, E].

Besides, we noticed that in **Ref. D** (i.e., *RSC Adv.* 4 (2014) 4069), the difference of the optimized interlayer spacing **in graphite** between GGA and GGA+vdw-D2 is about **0.1~0.2 Å**. In fact, in our specific study, we also considered the influence of the interlayer distance on the simulated SAED pattern and thus tried different d_{002} values to check its influence (e.g., 3.33 Å, 4.00 Å, and 3.405 Å; see **Supplementary Figs 17c(x)** and **19a**). The results show that the $d_{\{0110\}}$ and $d_{\{1210\}}$ values can keep constant at 2.13 and 1.23 Å, indicating that the d_{002} value won't affect the **in-plane lattice spacings**. **Ref. E** (i.e., *J. Chem. Theory Comput.* 10 (2014) 5318) adopted a similar strategy, namely also using three d_{002} values as well as different levels of DFT theory (refer to its **Figure 3 below**), to calculate the in-plane diffusion barrier of Li atom (**not Li⁺ ion here**) across and along a C-C bond.

Redacted

By the way, given a fixed d_{002} value, the second-time DFT calculation, being performed to confirm the XPS-peak splitting phenomena by assuming three

configurations of C_6LiC_6 (Line 250 and Fig. 4e,f), had nothing to do with the determination of the interlayer distance, too.

All in all, our two DFT calculations are not involved with the case pointed out by the reviewer, namely needing to consider the weak van der Waals interactions between graphene sheets or to determine the interlayer distance accurately. We only carried out qualitative DFT study to explain the experimental phenomena.

To make the description more clear, the **Supplementary Information (Page 4: DFT calculations)** has been **further modified** in the updated manuscript, as follows:

*“... On the one hand, to confirm the **in-plane** lattice-expansion phenomena of lithiated bilayer graphene **induced by** electron beam irradiation (Fig. 3c and Supplementary Figs 17b and 18), the Li-inlaid unit cells of LiC_{14} **sheet** and LiC_6 **sheet** with normal **in-plane** lattice parameters (as well as many other configuration models by adjusting the **Z-axis** spatial positions of Li atom) were constructed in the first step. Then, we restricted the Z-axis relaxation and merely allowed atoms to relax in X-Y plane (namely to achieve the smallest system energy) to obtain the optimized structure achievable in theory. Finally, the above CaRine Crystallography 3.1 software was applied to simulate the **in-plane** lattice parameters of the new structure (namely the **optimized unit cell**) used for comparison with those experimental values by SAED.*

*On the other hand, considering different configured Li sites relative to carbon lattice in X-Y plane, we **fixed the interlayer spacing (namely the d_{002} value)**, restricted the in-plane relaxation, and merely **allowed individual** atoms to relax along Z-axis direction in the binding-energy calculations...”*

Reviewers' comments:

Reviewer #3 (Remarks to the Author):

Tracking #: NCOMMS-17-27749-B

Review of Manuscript

Lithium intercalation into bilayer graphene

By Kemeng Ji^{1,2,†,*}, Jiuhui Han^{1,2,†}, Akihiko Hirata¹, Takeshi Fujita¹, Yuhao Shen^{1,3}, Shoucong Ning⁴, Pan Liu¹, Hamzeh Kashani^{1,2}, Yuan Tian^{1,2}, Yoshikazu Ito^{5,6,*}, Jun-ichi Fujita⁵, Yutaka Oyama^{2,*}

Discussions of Authors answers.

I would like to bring most significant arguments to the Authors after 1st revision which is significant making decision for publishing. At this stage, I still have significant concerns allowing it for publication to "Nature Communications".

I would like to thank the authors for detailed studying all the references I have mentioned in the previous review. I appreciate their approach and I won't argue some of their assumption regarding earliest published calculations. However, I would like to bring some important discussions regarding their results and conclusions.

The analysis of SAED patterns and confirmation of curved, misoriented or polycrystalline graphene needs to be always supported by Raman spectra features. The authors have mentioned "Hence, we believe there are no problems using the statics Raman data for the definition of the number of graphene layers in our study (in particular with the other parallel verifications including the measured specific capacities in Supplementary Fig. 7a,b). After all, it is easy to identify bilayer graphene (also monolayer graphene) thanks to its quite different Raman features from those of few and multi-layer graphenes as well as graphite."

In General, the Raman features of G and 2D bands such as the peak width, peak position, Lorentzian fits and their intensity ratio can distinguish between Monolayer, bilayer, few layers and stacking order. In fact, Lorentzian peaks should be fitted into graphene Raman 2D band and confirm either bilayer graphene is Bernal or misoriented. Lorentzian analysis of Raman 2D bands of "Bilayer" in Fig. 1 would be the best confirmation of Bilayer. If single Lorentzian has better fit into 2D band (which seems the case in Fig. 1 "bilayer") than 4 Lorentzian peaks that would be confirmation of multilayer (Bilayer or more) misoriented graphene. We would like to see Lorentzian analysis of Raman 2D band added in Fig. 1.

As for the Lithium intercalation, the authors used graphene foam which suggests the existence of randomly curved graphene sheets, then Raman analysis (not an SAED) is the best to characterize pristine graphene. Therefore, if Graphene 2D band (analyzed by Raman mapping) shows Single Lorentzian fit, it is very difficult to conclude graphene either Bilayer, Single or misoriented Multilayer. Although, if Raman 2D band shows consistently 4 Lorentzian fits, then we can confirm the evidence of predominated Bilayer graphene.

Similar analysis needs to be done for ex-situ Raman study as Authors have confirmed that "even the highly lithiated graphene foam preserves the initial bilayered architecture well without Li-dendrite or graphene restacking problems, fully demonstrating the merit of this 3D porous morphology and the protection of the internal Li layer (denoted as R) by the outer graphene sheets in the CRC-stacking configuration".

Highly lithiation of Graphene samples can result decoupling of multilayer/Bilayer into Single graphene sheets which would have the same Raman 2D band with Single Lorentzian fit (only 2D bandwidth would be changed). If pristine graphene exhibits Single Lorentzian in the case of Graphene Bilayer then before and after lithiation, authors may see similar Raman 2D and G ratio. Therefore, authors need to show Lorentzian analysis being sure that initial Bilayer has not transformed separated Single layers with extended distances.

"The (002) peaks of the X-ray diffraction (XRD) patterns for bilayer graphene and its lithiated products were too weak, so a multilayer sample was employed instead to study the analogous interplanar distance (i.e., $d_{002} = \text{ca. } 3.355 \text{ \AA}$ for graphite and 3.369 \AA for it; Supplementary Fig. 16)."

Analysis of XRD (002) peak by replacing "Bilayer" with Multilayer and confirming no significant (or slightly) position change was not a good choice for the Authors. If initial graphene foam has misoriented/randomly curved graphene layers than interlayer d-spacing can be varied from 3.3-3.5Å depend on their internal rotation or number of layers. Using multilayer instead of Bilayer can show slightly increase even without intercalation, which means areas that have no intercalation at all can exhibit XRD(002) peak with slightly extension of interlayer d-spacing. And the areas with highly intercalation wouldn't show any (002) peak or significant shift to at least $d=3.6\text{\AA}$ (AA stacking). That is why consideration of misoriented layers can be very important for your analysis.

I agree that is difficult to perform XRD analysis for Bilayer before and after Lithium intercalation, therefore, it is not correct to make conclusions based on multilayer XRD analysis instead of Bilayer. In fact, in general, XRD/SAED analysis is the best method to confirm the change in interlayer spacing although using multilayer instead of Bilayer can't be the same as authors used here.

Line- 274- "physicochemical phenomena, such as (1) the slightly increased interlayer space (Scheme 1a) by the XRD and SAED patterns" can be caused by misorientaion of multilayer/ bilayer graphene.

This assumption would not be correct here based on XRD and SAED analysis as SAED analysis would be inaccurate if they don't have Bilayer and multilayer misoriented layers exist at the same spot.

Though, in-situ Raman analysis of Li-intercalation process would allow authors demonstrate the shift of Lorentzian 4 peaks during intercalation and transforming it into Single Lorentzian when AB stacking transforms to ARA stacking. Eventually, Graphene 2D peak should be disappear for fully intercalated sample due to inserted Lithium layers which seems Authors registered, however detailed 2D peak transformation by Lorentzian peaks would be essential to show. Also, Lorentzian analysis of Graphene G band also would show intercalation process.

Reviewer #3: I would like to bring most significant arguments to the Authors after 1st revision which is significant making decision for publishing. At this stage, I still have significant concerns allowing it for publication to “*Nature Communications*”.

I would like to thank the authors for detailed studying all the references I have mentioned in the previous review. I appreciate their approach and I won't argue some of their assumption regarding earliest published calculations. However, I would like to bring some important discussions regarding their results and conclusions.

The analysis of SAED patterns and confirmation of curved, misoriented or polycrystalline graphene needs to be always supported by Raman spectra features. The authors have mentioned “*Hence, we believe there are no problems using the statics (corrected as “statistical”)* Raman data for the definition of the number of graphene layers in our study (in particular with the other parallel verifications including the measured specific capacities in Supplementary Fig. 7a,b). After all, it is easy to identify bilayer graphene (also monolayer graphene) thanks to

its quite different Raman features from those of few and multi-layer graphenes as well as graphite.”

In General, the Raman features of G and $2D$ bands such as the peak width, peak position, Lorentzian fits and their intensity ratio can distinguish between Monolayer, bilayer, few layers and stacking order. In fact, Lorentzian peaks should be fitted into graphene Raman $2D$ band and confirm either bilayer graphene is Bernal or misoriented. Lorentzian analysis of Raman $2D$ bands of “Bilayer” in Fig. 1 would be the best confirmation of Bilayer. If single Lorentzian has better fit into $2D$ band (which seems the case in Fig. 1 “bilayer”) than 4 Lorentzian peaks that would be confirmation of multilayer (Bilayer or more) misoriented graphene. We would like to see Lorentzian analysis of Raman $2D$ band added in Fig. 1.

As for the Lithium intercalation, the authors used graphene foam which suggests the existence of randomly curved graphene sheets, then Raman analysis (not an SAED) is the best to characterize pristine graphene. Therefore, if Graphene $2D$ band (analyzed by Raman mapping) shows Single Lorentzian fit, it is very difficult to conclude graphene either Bilayer, Single or misoriented Multilayer. Although, if Raman $2D$ band shows consistently 4 Lorentzian fits, then we can confirm the evidence of predominated Bilayer graphene.

Similar analysis needs to be done for *ex-situ* Raman study as Authors have confirmed that “*even the highly lithiated graphene foam preserves the initial bilayered architecture well without Li-dendrite or graphene restacking problems, fully demonstrating the merit of this 3D porous morphology and the protection of the internal Li layer (denoted as R) by the outer graphene sheets in the CRC-stacking configuration*”.

Highly lithiation of Graphene samples can result decoupling of multilayer/Bilayer into Single graphene sheets which would have the

same Raman 2D band with Single Lorentzian fit (only 2D bandwidth would be changed). If pristine graphene exhibits Single Lorentzian in the case of Graphene Bilayer then before and after lithiation, authors may see similar Raman 2D and G ratio. Therefore, authors need to show Lorentzian analysis being sure that initial Bilayer has not transformed separated Single layers with extended distances.

“The (002) peaks of the X-ray diffraction (XRD) patterns for bilayer graphene and its lithiated products were too weak, so a multilayer sample was employed instead to study the analogous interplanar distance (i.e., $d_{002} = ca. 3.355 \text{ \AA}$ for graphite and 3.369 \AA for it; Supplementary Fig. 16).”

Analysis of XRD (002) peak by replacing “Bilayer” with Multilayer and confirming no significant (or slightly) position change was not a good choice for the Authors. If initial graphene foam has misoriented/randomly curved graphene layers than (the?) interlayer d -spacing can be varied from 3.3–3.5 Å depend on their internal rotation or number of layers. Using multilayer instead of Bilayer can show slightly increase even without intercalation, which means areas that have no intercalation at all can exhibit XRD (002) peak with slightly extension of interlayer d -spacing. And the areas with highly intercalation wouldn't show any (002) peak or significant shift to at least $d = 3.6 \text{ \AA}$ (AA stacking). That is why consideration of misoriented layers can be very important for your analysis.

I agree that it is difficult to perform XRD analysis for Bilayer before and after Lithium intercalation, therefore, it is not correct to make conclusions based on multilayer XRD analysis instead of Bilayer. In fact, in general, XRD/SAED analysis is the best method to confirm the change in interlayer spacing although using multilayer instead of Bilayer can't be the same as authors used here.

Line-274-“*physicochemical phenomena, such as (1) the slightly increased interlayer space (Scheme 1a) by the XRD and SAED patterns*” can be caused by misorientation of multilayer/ bilayer graphene.

This assumption would not be correct here based on XRD and SAED analysis as SAED analysis would be inaccurate if they don't have Bilayer and multilayer **misoriented layers** exist at the same spot.

Though, *in-situ* Raman analysis of Li-intercalation process would allow authors demonstrate the shift of Lorentzian 4 peaks during intercalation and transforming it into Single Lorentzian when AB stacking transforms to ARA stacking. Eventually, Graphene 2D peak (**G peak?**) should be disappear for fully intercalated sample due to inserted Lithium layers which seems Authors registered, however detailed 2D peak transformation by Lorentzian peaks would be essential to show. Also, Lorentzian analysis of Graphene G band also would show intercalation process.

Response: Thank you very much for further sharing these technical insights and constructive suggestions.

Firstly, we agree that the analysis of SAED patterns and confirmation of curved, misoriented or polycrystalline graphene should be supported by Raman spectra features and that the Raman features of G and 2D bands such as the **peak width, peak position, Lorentzian fits** and their **intensity ratio** generally can distinguish between **monolayer, bilayer, few-layer, and stacking order**. Accordingly, we have carefully performed Lorentzian analyses for the *ex-situ* Raman G and 2D bands (Figures R1–3 in this letter) of both the pristine and lithiated-oxidized samples (Fig. 1e and Supplementary Fig. 13b in the manuscript) by referring to extensive literature (as displayed in Figure R4a). However, it should be noted that the Raman spectra and their Lorentzian analyses reported in previous literature^[R1,R2,R8–R11,R15] were based on **an individual graphene sheet** supported by a plane substrate, for which the 2D band shape and position can be **good fingerprints** of the monolayer and Bernal bilayer to

distinguish them from few-layer, multilayer, or misoriented bilayer. In contrast, each Raman spectrum in our study ($\sim 5 \mu\text{m}$ diameter of measurement area) corresponded to **a large number of graphene sheets** (refer to Figure R4b) and collected tremendous amounts of structural information regarding layer numbers, stacking order, misorientation, and so on, possibly causing the $2D$ band **fingerprints** of the Bernal bilayer to **disappear** or become **indefinable** (see the comparative Raman spectra in Figure R4a). To this end, further following the Lorentzian analysis in *Sci. Rep.* 7 (2016) 39944^[R18] (Figure R4a), each Raman spectrum in our study was carefully analyzed with the consideration of the co-existence of a small amount of monolayer, few-layer, or misorientation for understanding the ‘statistical’ spectrum (namely without distinguishable fingerprints) and identifying the graphene stacking order.

Particularly, it is known that **the G band is generally well-symmetric and can be fitted using a single Lorentzian peak**^[R1–R4,R6]. Such common phenomena lie in the generation mechanism of this **isotropic** band, associated with degenerated E_g phonon modes in **Bernal bilayer** graphene and E_{2g} phonon modes in **monolayer** graphene, respectively. Despite its general **featureless peak shape** (invalid for Lorentzian analysis, refer to our *ex-situ* results in Figure R1), we agree that the *in-situ* Raman spectra mapping (Supplementary Fig. 12, reproduced partially in Figure R5) can reveal the Li intercalation/de-intercalation processes unambiguously due to the influences of continuously changing strains (associated with the Li concentration) and charge carrier concentrations (consider the operando CV curve) on both its **peak intensity and position**^[R6] (refer to Figure R5c and the specific discussions colored in **red** in the updated SI [Pages 19–22]). From such an understanding, we have further proposed **two reasons** in the SI [Page 24] to explain the differences between the *in-situ* and *ex-situ* Raman spectra, focusing on the peak intensity and shift of the G band; namely, the **inherent** Li-doping effect (discussed at Figure R1) and the **extrinsic** configuration change caused by Li-oxidation (discussed at Figure R3). We believe clarifying these will help in understanding the intrinsic Raman phenomena of Li intercalation into the graphene interlayer.

Regarding the 2D bands, comparative Lorentzian analyses (i.e., single-peak analysis vs. multiple-peak analysis) have been performed according to the fitting methods/parameters reported in the literature^[R2,R11,R15,R18] (refer to Figure R4a). By contrast, **the single-peak fittings** (Figures R2a and R3a) demonstrate a lower degree of overlap between the experimental and Lorentzian fitting data and a smaller adjusted R -square value (e.g., with R^2 at 0.990 vs. 0.998 for the sampled pristine bilayer in Figure R2a,b, and at < 0.980 vs. > 0.994 for the lithiated bilayers exposed in air in Figure R3a,b), suggesting a greater rationality for **the multiple-peak analyses** (Figures R2b and R3b). Even so, it has been reported that **the single-peak analysis** can help determine the stacking order by virtue of peak position and bandwidth values (referred to as **FWHM**). As revealed by a previous statistical study based on an individual graphene sheet (refer to Figure R2c)^[R12], the FWHM and I_{2D}/I_G values are 41.0–59.5 cm^{-1} and 0.7–2.7 for the Bernal bilayer, respectively, while they are 36.0–40.5 cm^{-1} and 2.8–5.1 for a ‘disoriented’/misoriented stacked bilayer. Accordingly, our two results, ca. 41.5–48.0 cm^{-1} /1.9–2.3 for the two pristine bilayer spectra (supported by Ni or not) and 48.7–55.9 cm^{-1} /1.4–1.9 for the oxidized lithiated ones (Figures R2a and R3a), imply that all the graphene foams should possess a Bernal stacking configuration (see the further explanation about the oxidized lithiated ones below). Furthermore, according to **the combined Lorentzian analysis results for the bilayer samples** (Figures R2b and R3b), the monolayer or misorientation contents (represented by the pink-peak areas) are possible at 3%–7%. Consequently, both the fitting analyses prove that our pristine bilayer graphene foam is dominated by Bernal configuration. Besides the **SAED result** present in Fig. 1d, the **updated XRD data** (Supplementary Fig. 16, as discussed at Figure R6) can serve as additional evidence for the stacking configuration. We have added the related Raman analyses in Fig. 1e and discussion in the revised main text [Line 90], as follows: ‘Moreover, the specialized Lorentzian fitting analyses of the 2D bands in Fig. 1e(ii)^{23,26}, together with the matched FWHM and I_{2D}/I_G values²⁷, can help confirm the predominant Bernal stacking configuration of this CVD-grown bilayer graphene foam. This is also in line with subsequent XRD results showing an interplanar

spacing d_{002} of $\sim 3.35 \text{ \AA}$ (ca. 3.58 \AA at the AA stacking mode²⁸).

All the oxidized lithiated bilayers also exhibited AB-stacked Raman features (even PI with the definite ARA configuration; Figure R3), possibly due to the easy oxidation and thus aggregation of the intercalated Li atoms^[R22] (refer to their corresponding XRD patterns in Figure R6). That is, the **AR'B** stacking mode (where *R*' refers to the oxidized Li layer) should be the **energetically preferable** configuration for the extrinsic oxidation products. In fact, by virtue of **the lowest stacking energy**, the Bernal configuration is known as the most common structure for pristine graphenes/graphite, e.g., ca. 80% in single-crystal graphite and 90% in CVD-grown graphene plane sheets reported (refer to the similar XRD patterns for our graphene foams with varied layers in Figure R6a)^[R5,R13,R14,R21]. Moreover, **from an energy point of view**, even if there were some misoriented bilayered regions in the pristine graphene foam, the repeated electrochemical process in LIBs (stable and highly reversible, Supplementary Fig. 4) would enable these regions to become Bernal-structured after Li deintercalation. To this end, the experimental and theoretical analyses still support the 2D Li-intercalation model built on AB stacking bilayer graphene (Scheme 1).

Furthermore, we agree that **a highly lithiated individual graphene sheet** (with the ARA-stacking configuration) should exhibit monolayer features on the Raman 2D band by '*decoupling of the bilayer into single graphene sheets*'. For our **graphene foam** electrode material (exhibiting no obvious fingerprint feature as indicated above), however, it is unfortunately difficult to use a single Lorentzian fit to unambiguously analyze the present *in-situ* 2D band as **its signal-to-noise ratio is too large** (Figure R5a,b). This could be a single fit, but the accuracy is too low. Notwithstanding, as mentioned above, the *in-situ* **G band** exhibited a significant change in **peak intensity** compared with the relatively stable 2D band and the other bands (Figure R5c). In view of the usually much lower height and intensity of the **monolayer G band** than the 2D band (yielding a high I_{2D}/I_G value, e.g., at 3.6 in Fig. 1e), the phenomenon that the G band always becomes rather weaker after 0.09 V during the Li intercalation process (Supplementary Fig. 12b–e), should clearly display the Li intercalation-induced

‘decoupling of the bilayer into single graphene sheets’. Please note that here, *‘decoupling’* in our understanding should differ from *‘mechanical separation’*, like the production of graphene by the liquid-phase exfoliation of graphite (in an temperature-driven open system^[R6]), because there are still strong Li-C interactions combining the two graphene layers together for the graphene electrode in the electrochemical-driven closed battery system (refer to the stable GCD performances in Supplementary Fig. 4 and the structure-function relationship in Supplementary Fig. 1a). In addition, the distinct *G* band evolution at the above voltage range (after 0.09 V, corresponding to the **P2** stage) seems to also reflect the stacking configuration transformation between **ARB** (for the **P3** phase) and **ARA** (for the **PI** phase), in line with the XRD (Figure R6B) and XPS (Fig. 4 and Supplementary Fig. 21) phenomena (to be discussed below), further verifying the supplementary Lorentzian fitting analyses in Fig. 1e (namely the identification of the AB stacking configuration of the initial/pristine sample). We have added these discussions for Supplementary Fig. 12 [**Pages 19 and 22**].

Regarding the XRD data (Supplementary Fig. 16), we thank you for your reminder that we have supplemented the weak-signal patterns of some typical **bilayer** phases (P5, P3, and PI, reproduced from a previously deleted figure in **Page 79 of the Peer Review Profile**), and further measured the **multilayer PI** pattern for confirmation in the revised manuscript. Thus, based on the re-analyses of the typical XRD patterns of the parallel bilayer and multilayer samples (Figure R6) and on the closely related XRD study on the lithiated and further oxidized **graphite electrodes** in Ref. R22 (see its Figure 2b), we have revised the previous discussions in the main text [**Line 184**] as follows: *‘Furthermore, according to the simultaneous XRD patterns of the typical bilayer and multilayer electrode samples (Supplementary Fig. 16a–c), in the wake of Li intercalation (along with the transition from Li⁺ (radius: 0.76 Å) to Li atom (radius: 1.52 Å))^{7,28}, the *d*₀₀₂ value (~3.35 Å, similar for graphites possessing the predominant AB stacking configuration^{7,31,45}) seemed to increase slightly in the foregoing P5–P3 phases of ARB stacking (< 0.02 Å), only relying on a small population of Li atoms. However, after the unstable P2 phase⁴⁸ with a theoretically transitional stacking*

configuration from ARB to ARA²⁸ (Fig. 2b and Supplementary Figs 8 and 16), it appeared to become 3.64 Å for the bilayer PI sample (namely the ARA-stacking C₆LiC₆, Supplementary Fig. 16b). This was smaller than the multilayer (~3.70 Å, similar to that for LiC₆ of graphite electrode^{28,48}, Supplementary Fig. 16c) but larger than the pristine AA stacking graphite (~3.58 Å)²⁸. These inequalities likely result from the varied electrostatic repulsions between neighbouring graphene layers bonding with the interbedded Li atoms of different volume concentrations (namely Li/C atom ratios, Supplementary Fig. 16d), which can account for the above slight *d*₀₀₂ increase at the initial ARB mode, too. The overall increasing trend of this average spacing upon Li intercalation is consistent with literature reports for graphites^{3,4,10,32,41,45,48}. The other related characterization results also favour these observations and discussions, including the above *in-situ* Raman spectra phenomena as well as the SAED and XPS data (please refer to the below discussions). That is to say, Li intercalation into the interlayers of various graphenic carbons not only follows similar mechanisms in the 2D plane (Fig. 2c, refer to the analogous CV and SAED data in Supplementary Figs 8c and 17b,e for bilayer and multilayer samples), but also along the Z axis. (Out of the above considerations, we therefore deleted the weak-signal bilayer XRD data at the 2nd review and only showed the multilayer data to illustrate the varying *d*₀₀₂ spacings for simplify.)

Regarding the SAED results for the interlayer spacings of the Li-bilayered graphene samples, namely ‘from ca. 3.42 Å at P4 to 3.50 Å at P1’, we really appreciate your reminder and sincerely apologize for neglecting the **local misorientation/incommensurate structure** induced by electron beam irradiation (please refer to the revised Supplementary Fig. 18b) and improperly regarding these two **extrinsic** *d*₀₀₂ values observed as the **intrinsic** structure parameters. Thus, in view of the stacking configurations of P4 (ARB) and P1 (ARA) and the interlayer spacings of **pristine** AB (3.36 ± 0.01 Å) and AA (3.58 Å) stacking graphenes/graphite, the real *d*₀₀₂ values should be < 3.42 Å at P4 and > 3.50 Å at P1 judging from only the SAED patterns (Line 201 and Figure R7, please see the detailed discussions at the updated Supplementary Figs 17a and 17d). This result is still in line with the above XRD results

with superior statistical characteristics to eliminate the interference of local misorientation (please refer to Figure R4b for the huge difference between the two techniques in the analysis ranges). Moreover, by taking the misorientation/incommensurate structure (namely the irradiation effect) into account, we have now better understood all the SAED patterns (extrinsic but regular). To this end, we have further modified the related Li-graphene structure models (Figs 3a(iv) and 3b(i,iii,iv) and Supplementary Figs 17–19) and reorganized some related discussions in the main text, as follows: ‘*In this way, both the originally uniform Li distribution and standard graphene stacking order are disturbed over some local regions or even the overall selected area, resulting in changing diffraction patterns for each sample. This irradiation effect is reflected by contradictions between several observed values/phenomena, including: total $I_{\{0110\}}/I_{\{1210\}}$ ratios and the branched diffracted intensities for the P4 phase with a low Li population (see the two insets taken along the red lines marked in Figs 3a(i) and 3b(i) versus the case in Fig. 3b(iii)); the emergence of paired orthohexagonal diffraction patterns **with an arbitrary rotation angle** but varied spot brightness (on behalf of incommensurate/misoriented stacking configuration (Fig. 3b(i,iii) and Supplementary Fig. 17); the changing $I_{\{0110\}}/I_{\{1210\}}$ ratios listed in Figs 3a(i,iii) and 3b(i,iii) and Supplementary Fig. 17d; and the disparate in-plane lattice spacings for the Li-inlaid (with a ~3% lattice expansion²⁸) and Li-free graphene sheets at the highly lithiated PI or P1 phases (Fig. 3c and Supplementary Figs 17c,d and 18)’ [**Line 212**], and ‘*which can drive the local-region graphene sheets to rotate arbitrarily and thus bring about various graphene stacking configurations over the selected area*’ [**Line 232**].*

Accordingly, based on the clearer **structural information** revealed by the deeper Raman, XRD, and SAED analyses, the related discussions for the **XPS data** have also been modified appropriately to serve as a clearer proof of the **chemical composition**, as follows [**Line 264**]: ‘*Thus, from the point of view of chemical composition change, the combined experimental and theoretical results further reveal and verify the phase-wise stacking configurations suggested by XRD and SAED*

characterizations performed at different analytical ranges. This is seen in: the maintained ARB structure at P5 ($C_{122}LiC_{122}$, showing predominant metallic Li 1s and relatively few C 1s species) and P4 ($C_{42.7}LiC_{42.7}$, possibly with two Li-embedded sites according to Scheme 1b and Fig. 4e,f); the coexisting ARB and SP (unstable) modes at P3 ($C_{14}LiC_{14}$) with a large enough Li concentration (Supplementary Fig. 16); and the ARA mode at P1 (C_6LiC_6) with ionic Li 1s specie of high BE (Fig. 4d,f, with the electrostatic charge of Li possibly at $+0.9|e|$ in the C_6 -Li constitutional unit)^{11,14,28,47} and a simultaneous nearly single C 1s specie. In brief, upon the Li intercalation, the holistic ionic character of Li atoms is enhanced along with the stacking configuration transformation, while the metallic character decreases (Fig. 4d,f).

Consequently, based on the above analyses, we believe that the statistical data from the combined Raman and XRD/SAED techniques provide consistent and solid evidence for the predominant Bernal stacking structure of the pristine graphene foam samples. Benefited by your constructive comments and reminder regarding the Raman, XRD, and SAED results, we can now deduce: (i) the stacking configurations in both P3 and P2 phases by combining the composition information reflected by the XPS results, (ii) the change rule of the interlayer spacing (*'the increasing interlayer spacings (Scheme 1a) depending on the charge amounts of neighbouring graphene layers and, in particular, on their stacking orders by the phase-wise XRD and SAED patterns (Supplementary Figs 16 and 17a,d)'* [**Line 290**]), (iii) the full influences of beam irradiation on the local structure of the lithiated graphene samples, and thus (iv) the currently unambiguous and more complete mechanism of Li intercalation into graphene interlayer (see the modified Scheme 1). Due to the many rounds of review since our initial submission on 24th October 2017 (refer to the Peer Review Profile), this is our last chance to address your concerns for the acceptance of this work by Nature Communications according to its review policy. We sincerely hope the above experimental analysis will be satisfactory this time. Thank you very much for your technical insights again and you are welcome to contact us if you have any new questions. (Please see Figures R1–R7 and the references for this response letter below.)

Figures R1–R7 with detailed analyses for this response letter:

Figure R1. Lorentzian fitting analyses of Raman *G* bands. **(a)** Results for the pristine graphene foams (Fig. 1e). **(b)** Results for the lithiated bilayer-graphene foams exposed in air (Supplementary Fig. 13b). The peak position and FWHM in the updated figures were obtained by this single peak fitting (with the adjusted R^2 value at 0.990–0.995).

Result interpretation: The *G* band, whose peak position/center **cannot** serve as a fingerprint to identify graphene^[R1], generally displays a symmetric single peak based on its generation mechanism (namely the in-plane stretching vibration of sp^2 carbon atoms (E_{2g} mode symmetry)^[R2–R4]), although it may split into two components (with some down-shift possible up to 30 and 15 cm^{-1} , respectively^[R5]) due to possible effects from strain^[R5,R6] and charge carrier concentration^[R6]. Moreover, the presence of the quite weak *D'* shoulder band (at 1620 cm^{-1} for nanocrystalline graphite) may slightly affect the Lorentzian fitting^[R7].

Regarding the pristine samples here **(a)**, as the number of graphene layers increased, the estimated FWHM values gradually decreased from 23.8 to 19.3 cm^{-1} . There are no obvious or regular variations in the peak position (1584 ± 1.6 cm^{-1}).

Regarding the lithiated bilayered samples **(b)**, which had been oxidized in air, the estimated FWHM values (30.9–23.4 cm^{-1}) were relatively stable and a little larger than that of the pristine **bilayer** (22.7 cm^{-1}). The peak position (1583.4 ± 1.3 cm^{-1}) and intensity did not show obvious change regularities which were observed in the *in-situ* Raman spectra mapping (Supplementary Fig. 12, refer to the deleted figures (partly shown in Figure R5c) at Pages 76–77 of the Peer Review Profile). **One reason** could lie in the continuously changed currents, strain, and doping levels for the *in-situ on-line* tested graphene foam electrode^[R6,R8–R10]. Please see the **other reason** in Figure R3.

Figure R2. Comparative Lorentzian fitting analyses of Raman $2D$ bands for the pristine Raman spectra shown in Fig. 1e. **(a)** Single peak-fitting results ($R^2 < 0.992$), showing the peak position/center and FWHM information. **(b)** Combined-fitting results ($R^2 > 0.995$) in view of the coexistences of graphene sheets with different structure characteristics at the same Raman analysis area (such as layer number, curvature, orientation, and stacking order). The single pink fitting curve in each spectrum refers to the contribution of the **monolayer** or **misorientation** (only $\sim 6\%$ in the integral area for the **bilayer** spectra according to the area ratios alongside). Four green fitting curves are applied to represent the contribution of the **Bernal bilayer**, for which the peak distance of the strong middle two are kept at $20\text{--}22\text{ cm}^{-1}$ according to previous research^[R5,R11,R15]. **(c)** The histograms of statistical FWHM values of $2D$ band and I_{2D}/I_G ratio for the **Bernal bilayer** and **disoriented/misoriented bilayer** (reproduced from Ref. R12).

Result interpretation: For an individual plane graphene sheet with a specific layer number (generally applied in the literature as illustrated in Figure R4a), its Raman $2D$ band can usually display quite distinctive fingerprint features. Thus, it can allow 1 Lorentzian peak to fit well for the $2D$ band of the monolayer^[R1,R2,R8,R13] and disoriented bilayer^[R11–R14], 4 peaks for the Bernal bilayer^[R1,R2,R8,R11–R17], 6 peaks for the trilayer^[R7,R9,R10,R14,R17], and 2 Lorentzian fits for graphite^[R1,R2,R17]. However, for a graphene foam like in our study (refer to the middle image in Figure R4b), each captured Raman spectrum would cover a large amount of graphene sheets with varied layers (dominated by the bilayer in our sample) and different spatial orientations. Thus, the integrated feature at the $2D$ band (summation of those for graphene sheets of all kinds) would possibly become quite ambiguous and hardly show the usually differentiable Raman fingerprints (refer to the comparison in Figure R4a). Therefore, combined fitting^[R18] (as shown in Figure R2b) should be more reasonable than a single-

peak Lorentzian analysis (as illustrated in Figure R2a). Moreover, it can be seen that Figure R2b indeed displays better fitting results than Figure R2a, as reflected by both the larger R^2 values and higher overlap ratios between the experimental and fitting data.

In particular, regarding the Lorentzian analysis of the Raman 2D bands of the two **bilayers** (supported by Ni or not), the better fitting results in Figure R2b show that the **Bernal bilayer** is predominant (> 93%) in our bilayer-graphene foam. In fact, besides the evidence provided by the comparative analysis here, **the XRD results** (Supplementary Fig. 16, reproduced in Figure R6B) and the statistical results in Figure R2c should also help confirm the predominant Bernal configuration of our bilayer-graphene foam. Please note that the differences between the two **bilayer** Raman spectra should possibly lie in the substrate-induced doping effect widely discussed in the literature^[R1,R8,R10,R15].

Furthermore, for your reference, Lorentzian analyses of the Raman 2D bands for the **pristine monolayer**, **few-layer**, and **multilayer** were handled similarly. For the monolayer (Figure R2b), the contribution of the A-B stacked bilayer (most likely to coexist here) was taken into account and varied at 25–35% based on the fitting results. In view of the close Raman features of the few-layer (2–5 layers) and bilayer (Figure R2a), 6 Lorentzian peaks (typical for the trilayer, refer to Figure R4a) were adopted in the analysis of the Raman 2D band of the few-layer spectrum (Figure R2b), which yielded a peak area ratio of ~5% for the misorientation or monolayer (marked in pink). As for the multilayer (6–10 layers; also with Bernal stacking according to the XRD results in Figure R6B), because its Raman spectrum (Figure R4a) ‘*becomes hardly distinguishable from that of bulk graphite*’^[R2], here we adopted 3 peaks to illustrate its Lorentzian fitting (Figure R2b).

All in all, the combined characterization techniques (Figure R4b), including the **SAED**, **Raman**, **XRD**, and **XPS** results, provide solid evidence that our CVD-grown bilayer-dominated graphene foam possesses a mainly Bernal stacking configuration.

Figure R3. Comparative Lorentzian fitting analyses of Raman 2D band for the **lithiated** bilayer-graphene foams **exposed in air** (Supplementary Fig. 13b). **(a)** Single peak-fitting results (with R^2 at 0.97–0.98), showing the peak position and FWHM information. **(b)** Combined-fitting results ($R^2 > 0.994$) using the same processing method for **bilayer** in Figure R2b. The image on the right (reproduced from Ref. R11) displays the four scattering processes of the Raman 2D band of bilayer graphene corresponding to the P₁₁, P₁₂, P₂₁ and P₂₂ peaks, respectively.

Result interpretation: Although the **macroscopic** morphology of these **oxidized lithiated** bilayer graphene samples (like porous structure) was well-maintained, their **microscopic** stacking configurations transformed into the lowest system energy structure due to the Li oxidation and aggregation (denoting the generated Li₂O_x layer as R^*), possibly with the AR'B mode in view of their **XRD (002)** peaks in Figure R6. Besides, according to the statistical results in Figure R2c, the present FWHM and I_{2D}/I_G values (48.7–55.9 cm⁻¹ and 1.4–1.9), similar to those of the pristine **Bernal bilayer** (48.0 cm⁻¹ and 2.3), can further help indicate the A-B stacking configuration of these oxidation products. This could be **the second reason** why these *ex-situ* spectra displayed significantly different Raman responses from those traced from the *in-situ* Raman spectra (Supplementary Fig. 12, refer to Figure R5c). In particular, for the *in-situ* observation, the intercalated Li atoms can limit the in-plane stretching vibration of sp² carbon atoms (E_{2g} mode symmetry) associated with the peak intensity of the G band (as well as its downshift position discussed in Figure R1), but will not significantly influence the intensity of the 2D band according to its different generation mechanism (namely a double resonance process associated with electron-phonon interactions)^[R2,R5,R11]. As a result, as the Li densities increased, the G band became very weak until it vanished below 0.09 V (associated with the changed stacking configuration), while the 2D band was relatively stable, finally yielding ever-increasing I_{2D}/I_G values to approach that for the monolayer (refer to Figure R5c(ii)).

Figure R4. (a) Comparative Raman 2D bands of **graphene foam** (e.g., in this work and Ref. R18, refer to the middle image in (b)) and **an individual graphene sheet** (e.g., in Refs. R2, R7, R17, R11, and R10, refer to the first image in (b)). All the figures were reproduced from the corresponding literature. The CMLG and IMLG terms in the figure of Ref. R18 refer to the commensurate and incommensurate multilayer graphene 3D network, respectively. **(b)** The figure provided in the 2nd-round review for **Q2** of **Reviewer #2** [Page 34 of the Peer Review Profile], which shows the varied detection ranges of different characterization techniques applied in this study.

Result interpretation: It can be seen clearly that the Raman 2D bands of graphene foam (also refer to the spectra reported by H. M. Cheng etc. in *Nat. Mater.* 10 (2011) 424^[R19]) hardly possess the evident fingerprint features dependent on the layer number of an individual plane graphene sheet (refer to the purple curves inset in (a) for this work and Ref. R2).

Figure R5. (a) *In-situ* Raman spectra reproduced from Supplementary Fig. 12e. (b) Sample *in-situ* Raman spectrum recorded at 0.17–0.16 V (deleted in the 1st-round review, see Page 75 of the Peer Review Profile). (c) Evolutions of the peak position and intensity of some typical Raman bands defined in (b) during the Li-intercalation process (reproduced from the deleted Supplementary Fig. 15c,d in the 1st-round review, see Page 21 and Pages 76–77 of the Peer Review Profile). Due to the inevitable error from a single point, the slope values of the trend lines were applied to identify their changes. The increasing trend of the I_{2D}/I_G ratio was obvious based on (ii).

Result interpretation: Judging from Figure R5a,b, due to the very high signal-to-noise ratio and the ever-changing peak shape (even at a same phase, a(i, iii)), the *in-situ* data of Raman 2D band (a(ii)) cannot be used for exact Lorentzian analysis.

Figure R6. (A) Previous Supplementary Fig. 16. (B) Updated Supplementary Fig. 16. (a) XRD patterns of pristine graphenic carbon foams with increased graphene layers (until the average n value > 10). (b) XRD patterns of the bilayer graphene-based samples before and after being oxidized in air (the previously deleted Supplementary Fig. 21c shown at Page 81 of the Peer Review Profile). Kapton tape displays no sharp XRD diffraction peak. (c) XRD patterns of the fresh multilayer graphene-based samples and their oxidation products. (d) Schematic of the Z-axis structures of the PI phases (with LiC₆ stoichiometry in plane) for (i) bilayer graphene and (ii) graphite/multilayer graphene, assuming that the PI pattern in (b) is accurate and the valences of the embedded Li atoms are constant at +0.9 (or with a quite slight change).

Result interpretation (based on Figure R6B): All the **pristine** graphene foam samples

(Figure R6a) should possess the predominant AB stacking configuration in view of their larger 2θ angles (with the typical interlayer spacings d_{002} at 3.36 ± 0.01 Å) than that for AA stacking (24.8° , $d_{002} = 3.58$ Å)^[R18,R21], in line with the Raman results and discussion (Fig. 1e and Figure R2). The slight shift of the XRD (002) peak for the bilayer and multilayer samples (also considering the reported varied d_{002} values for different graphites, 3.34–3.37 Å) may result from the altered fractions of their coexisting misorientation configurations. The (002) peak of the pristine bilayer sample (as well as its lithiated products, Figure R6b) is rather weak due to the sole crystal face for diffraction (refer to Figure R6d(i)).

In view of the patterns of the three fresh **bilayer** products (Figure R6b), a new peak seemed to emerge at 24.4° at the PI phase (with ARA configuration in theory). Although its corresponding d_{002} value (3.64 Å, if correct) was smaller than that for LiC_6 of graphite electrodes (~ 3.70 Å, 24.0°), it is understandable by considering the different average valences of C atoms in C_6LiC_6 (bilayer graphene) and LiC_6 (graphite) and thus the varied repulsive forces between the adjacent graphene sheets (refer to Figure R6d and the above smaller d_{002} value for the pristine AA stacking graphite), respectively. After exposure to air, however, each oxidation product seemed to display a sole diffraction peak at about 26.5° (no LiC_x peaks), as further confirmed by the stronger-signal spectra of the oxidized multilayer samples (c). This specific phenomenon should indicate that all the oxidation products possessed the $AR'B$ structure (R' : Li_2O_x layer) regardless of the intrinsic stacking configurations of their fresh counterparts, possibly due to the random oxidation and thus aggregation of Li atoms at the interlayer on this occasion, in favour of the Lorentzian fitting analyses of the *ex-situ* Raman spectra (Figure R3b/Supplementary Fig. 13b).

The XRD pattern of the applied pristine **multilayer** sample was double-checked (Figure R6c), from which its (002) diffraction peak was determined to be centred at 26.59° ($d_{002} = 3.35$ Å). In view of the typical patterns of the overall four fresh lithiated samples (Figure R6c), as the Li concentration increased, this original (002) peak was slightly blueshifted in the foregoing phases until 26.42° at P2 ($d_{002} = 3.37$ Å) and seemed to vanish at the final ARA-stacking **PI**, exhibiting a suspected new peak at 24.01° ($d_{002} = 3.70$ Å, quite close to that for LiC_6 of graphite). In addition, the pattern of the material for the transitional **P2** phase (refer to the highly overlapping CI and C2 CV peaks in Fig. 2b and Supplementary Fig. 8c and to the abnormal Li 1s and C 1s XPS spectra of P2 in Fig. 4a,c) was quite similar to that of the unstable LiC_{12} stoichiometry of the graphite electrode (identified to be a mixture of LiC_6 and ‘liquid-like’ LiC_{12-18} by disproportionation)^[R22], which displayed multiple diffraction peaks such as at 23.5° (3.78 Å), 23.8° (3.74 Å), 24.4° (3.64 Å), 25.1° (3.54 Å), and 25.7° (3.46 Å). Besides, in line with the P2 pattern, there was also a broad diffraction peak centred at ca. 25.7° (3.46 Å) for **P3** in addition to the retentive one at 26.4° (refer to the stronger and broader (002) peak of its oxidation product). In view of the definite LiC_x composition at P3 for either graphenes or graphite (different from that at P2), we consider its real product to possibly form two stacking configurations, including the initial ARB mode and another relatively balanced/stable SP stacking structure (namely a theoretically intermediate state between A-B and A-A, refer to Fig. 4e,f and

Supplementary Figs 6c,d and 17b(xi)), well-consistent with its Li 1s XPS spectrum containing two Li 1s species (Fig. 4c). Thus, P3 can be regarded as another transition phase due its relatively large Li density (refer to its multiply peculiar results summarized at Supplementary Fig. 19); however, due to its much lower Li density, it is more stable than P2 (possibly with a theoretically quasi-SP structure by following P3 and considering the ideal SP structure of either $C_{14}LiC_{14}$ for P3 (Scheme 1b) or C_6LiC_6 for P1 (Fig. 4e,f)). The single weak peak of 26.6° for **P4** indicates that this low-Li-density phase should retain the A-B structure of the pristine sample, and so should the **P5** phase. Even so, the intercalated Li atoms seems to be located at two sites at **P4** (namely Site 1' and Site 2' shown in Fig. 4e,f) in its ideal stacking configuration (Scheme 1b and Supplementary Fig. 9b), still agreeing with its C1s, Li 1s, and O 1s XPS results and further explaining the difference between its XPS spectra and those of **P5** (Fig. 4a–d and Supplementary Fig. 21b–d).

Consequently, combining the composition information reflected by the XPS results, these XRD results together with the SAED analyses reveal the Li concentration-dependent structure at each phase, such as the ARB structure at P5/P4, the mixed ARB and SP modes at P3, the theoretically assumed quasi-SP mode at the transient/transitional P2 stage, and the ARA mode at P1.

Figure R7. Normal-incidence SAED patterns of the **bilayered** P4 and P1 samples **after irradiation for some time**. Reproduced from Supplementary Figs 17a(ii) and 17d(viii). Similar revisions were made to Fig. 3b(i,iii) and Supplementary Figs 17d(ii,iv,v) and 17e(iv).

Result interpretation: In addition to the incommensurate structure in size, the beam irradiation also caused some rotation, leading the observed d_{002} value (3.42 Å) in **(a)** to be larger than the real one of the standard ARB-stacking $C_{42.7}LiC_{42.7}$ (~3.36 Å by the XRD results in Figure R6B) and the observed 3.50 Å in **(b)** to be smaller than the real one of the ARA-stacking C_6LiC_6 (~3.65 Å by the XRD results in Figure R6B).

Notes:

- [R1] A. Das, B. Chakraborty, A. K. Sood. Raman spectroscopy of graphene on different substrates and influence of defects. *Bull. Mater. Sci.* **31**, 579–584 (2008).
- [R2] A. C. Ferrari, *et al.* Raman spectrum of graphene and graphene layers. *Phys. Rev. Lett.* **97**, 187401 (2006).
- [R3] A. C. Ferrari. Raman spectroscopy of graphene and graphite: Disorder, electron–phonon coupling, doping and nonadiabatic effects. *Solid State Commun.* **143**, 47–57 (2007).
- [R4] F. Rosenburg, E. Ionescu, N. Nicoloso & R. Riedel. High-temperature Raman spectroscopy of nano-crystalline carbon in silicon oxycarbide. *Materials* **11**, 93 (2018).
- [R5] N. Ferralis. Probing mechanical properties of graphene with Raman spectroscopy. *J. Mater. Sci.* **45**, 5135–5149 (2010).
- [R6] M. H. Oliveira Jr., *et al.* Formation of high-quality quasi-free-standing bilayer graphene on SiC(0001) by oxygen intercalation upon annealing in air. *Carbon* **52**, 83–89 (2013).
- [R7] C. Trudeau, L.-I. Dion-Bertrand, S. Mukherjee, R. Martel & S. G. Cloutier.

Electrostatic deposition of large-surface graphene. *Materials* **11**, 116 (2018).

- [R8] H. Bukowska, *et al.* Raman spectra of graphene exfoliated on insulating crystalline substrates. *New J. Phys.* **13**, 063018 (2011).
- [R9] M. Z. Iqbal, M. F. Khan, M. W. Iqbal & J. Eom. Tuning the electrical properties of exfoliated graphene layers using deep ultraviolet irradiation. *J. Mater. Chem. C* **2**, 5404–5410 (2014).
- [R10] M. W. Iqbal, *et al.* Modification of the structural and electrical properties of graphene layers by Pt adsorbates. *Sci. Technol. Adv. Mater.* **15**, 055002 (2014).
- [R11] J.-U. Lee, *et al.* Polarization dependence of double resonant Raman scattering band in bilayer graphene. *Carbon* **72**, 257–263 (2014).
- [R12] R. Kato, *et al.* Bilayer graphene synthesis by plasma treatment of copper foils without using a carbon-containing gas. *Carbon* **77**, 823–828 (2014).
- [R13] B. R. Luo, *et al.* Chemical vapor deposition of bilayer graphene with layer-resolved growth through dynamic pressure control. *J. Mater. Chem. C* **4**, 7464–7471 (2016).
- [R14] L. Gong, *et al.* Reversible loss of Bernal stacking during the deformation of few-layer graphene in nanocomposites. *ACS Nano* **7**, 7287–7294 (2013).
- [R15] C.-W. Huang, *et al.* Probing 2D sub-bands of bi-layer graphene, *RSC Adv.* **4**, 51067–51071 (2014).
- [R16] L. M. Malard, M. A. Pimenta, G. Dresselhaus & M. S. Dresselhaus. Raman spectroscopy in graphene. *Phys. Rep.* **473**, 51–87 (2009).
- [R17] M. S. Dresselhaus, A. Jorio, M. Hofmann, G. Dresselhaus & R. Saito. Perspectives on Carbon Nanotubes and Graphene Raman Spectroscopy. *Nano Lett.* **10**, 751–758 (2010).
- [R18] T. M. Paronyan, A. K. Thapa, A. Sherehiy, J. B. Jasinski & J. S. D. Jangam. Incommensurate graphene foam as a high capacity lithium intercalation anode. *Sci. Report.* **7**, 39944 (2016).
- [R19] Z. P. Chen, *et al.* Three-dimensional flexible and conductive interconnected graphene networks grown by chemical vapour deposition. *Nat. Mater.* **10**, 424–428 (2011).
- [R20] G. Radhakrishnan, J. D. Cardema, P. M. Adams, H. I. Kim, & B. Foran, Fabrication and electrochemical characterization of single and multi-layer graphene anodes for lithium-ion batteries. *J. Electrochem. Soc.* **159**, A752–A761 (2012).
- [R21] Z. H. Wang, S. M. Selbach & T. Grande. Van der Waals density functional study of the energetics of alkali metal intercalation in graphite. *RSC Adv.* **4**, 4069–4079 (2014).
- [R22] R. L. Sacci, L. W. Gill, E. W. Hagaman & N. J. Dudney. Operando NMR and XRD study of chemically synthesized LiC_x oxidation in a dry room environment, *J. Power Sources* **287**, 253–260 (2015).

REVIEWERS' COMMENTS:

Reviewer #3 (Remarks to the Author):

Review Tracking #: NCOMMS-17-27749D (or 17-27749B)

Lithium intercalation into bilayer graphene

2 Kemeng Ji^{1,2,†,*}, Jiuhui Han^{1,2,†}, Akihiko Hirata¹, Takeshi Fujita¹, Yuhao Shen^{1,3}, Shoucong Ning⁴, Pan Liu¹, Hamzeh Kashani^{1,2}, Yuan Tian^{1,2}, Yoshikazu Ito^{5,6,*}, Jun-ichi Fujita⁵, Yutaka Oyama²,

After revision the Manuscript, significant changes have been made and new experimental data were added into previous version. New data analysis bring a better understanding of the mechanism proposed by Authors. I appreciate their efforts answering my previous concerns regarding the material characterization and clarifying it.

After reading Author's response and considering all the changes made in this version, I believe there is significant improvement in the manuscript.

Therefore, I would like to recommend the manuscript "Lithium intercalation into bilayer graphene" for publishing in Nature Communications due to importance of collected scientific new data. The study of Lithium intercalation mechanism into graphenic Carbon is very important for developing further efficient energy storages.

In addition, I would like bring some minor notes to your attention before the final Revision.

1. Line 91- please specify what is I(2D) and I(G) and based on identification check I(2D) and I(G) ratio values in Fig. 1(i). If I(2D) and I(G) are the intensity heights of the peaks, then the values in Fig. 1(i) don't look correct. Please, check these values and be sure it doesn't effect on statistical data of definition of Bernal Bilayer.
2. In my previous review I have mentioned about Raman 2D band of lithiated electrode, the Authors corrected it to G band? I would like confirm that G band may split 2 peaks or shift (even 4 by Lorentzian fit) during the Lithium interaction but it wouldn't disappear if sp² carbon exists. Although, some authors have noticed intensity decrease 2D band due to high intercalation for graphitic carbon.

According to your latest comments, our detailed response is provided below.

The revised parts, using the 'track changes' feature in the updated manuscript, are marked **in red** here.

Reviewer #3: *After revision the Manuscript, significant changes have been made and*

new experimental data were added into previous version. New data analysis bring a better understanding of the mechanism proposed by Authors. I appreciate their efforts answering my previous concerns regarding the material characterization and clarifying it.

After reading Author's response and considering all the changes made in this version, I believe there is significant improvement in the manuscript.

Therefore, I would like to recommend the manuscript "Lithium intercalation into bilayer graphene" for publishing in Nature Communications due to importance of collected scientific new data. The study of Lithium intercalation mechanism into graphenic Carbon is very important for developing further efficient energy storages.

In addition, I would like bring some minor notes to your attention

before the final Revision.

1. Line 91- please specify what is $I(2D)$ and $I(G)$ and based on identification check $I(2D)$ and $I(G)$ ratio values in Fig. e1(i). If $I(2D)$ and $I(G)$ are the intensity heights of the peaks, then the values in Fig. 1(i) don't look correct. Please, check these values and be sure it doesn't effect on statistical data of definition of Bernal Bilayer.

2. In my previous review I have mentioned about Raman 2D band of lithiated electrode, the Authors corrected it to G band? I would like confirm that G band may split 2 peaks or shift (even 4 by Lorentzian fit) during the Lithium interaction but it wouldn't disappear if sp^2 carbon exists. Although, some authors have noticed intensity decrease 2D band due to high intercalation for graphitic carbon.

Response: We really appreciate your recognition, positive comments, and constructive suggestions to help us improve this study constantly and significantly. And, thank you very much for your reminder about the two minor notes for our final manuscript.

Regarding to Note 1, the I_{2D} and I_G refer to the peak areas of 2D and G bands rather than their heights. Thus, you can see there is no problem about the calculation about the I_{2D}/I_G values in Fig. 1e(i) (as well as Supplementary Fig. 13b(i)). We have specified this point in this sentence [Line 91], as follows: “together with the matched FWHM and I_{2D}/I_G values (i.e., peak area ratios)²⁷”.

Regarding to Note 2, in fact we share the same opinion as you, namely that “G band may split 2 peaks or shift (even 4 by Lorentzian fit) during the Lithium

interaction but it wouldn't disappear if sp² carbon exists". This phenomenon should be associated with the generation mechanism of *G* band (namely the in-plane stretching vibration of sp² carbon atoms) and thus the possible factors affecting it (e.g., strain and charge carrier concentration), as we indicated at Result Interpretation of Figure R1 in the previous response letter. Meanwhile, our recorded *in-situ* Raman spectra (Supplementary Fig. 12c–e) seem to confirm the phenomenon at *2D* band reported by other researchers you mentioned here, namely that its intensity will decrease due to the interlayered Li intercalation. All in all, from our results it can be seen that, at the end of the Li intercalation the peak height of *2D* band decreased a little while the *G* band became rather weak but didn't disappear (refer to their intensity evolutions of the *G* and *2D* bands and their comparative slope values based on our initially rough statistical data, as shown below in the previous Figure R5c(ii)). Anyway, we sincerely apologize for misunderstanding your meaning by suspecting the *2D* peak in your previous comment possibly to be the *G* peak, in particular seeing you also mention *G* peak in the same paragraph: "*Graphene 2D peak (G peak?) should be disappear for fully intercalated sample due to inserted Lithium layers which seems Authors registered, however detailed 2D peak transformation by Lorentzian peaks would be essential to show. Also, Lorentzian analysis of Graphene G band also would show intercalation process.*" Luckily, this change doesn't influence the understanding of the Li intercalation phenomena and the existing discussions in this study.

In a word, we are much grateful for your very careful and professional reviews and guidance, which have benefited us and this study a lot.

Supplementary Fig. 12e

Figure R5c(ii)

Figure. In-situ Raman spectra with the CV rate at 0.02 mV s^{-1} (Supplementary Fig. 12e) and intensity evolutions of the recorded *G* and *2D* bands during the Li-intercalation process (Figure R5c(ii)). Due to the inevitable error from a single point, the slope values of the trend lines were applied to identify their intensity/height changes.